# Modeling Cell Dynamics and Interactions with Unbalanced Mean Field Schrödinger Bridge

**Zhenyi Zhang**[1,*]**, Zihan Wang**[2,*]**, Yuhao Sun**[3,*]**, Tiejun Li**[1,3,4,†] **and Peijie Zhou**[2,3,4,5,†]

[1]LMAM and School of Mathematical Sciences, Peking University.
[2]Center for Quantitative Biology, Peking University.
[3]Center for Machine Learning Research, Peking University.
[4]NELBDA, Peking University. [5]AI for Science Institute, Beijing.
Emails: {zhenyizhang,jackwzh,2501111524}@stu.pku.edu.cn,
{tieli,pjzhou}@pku.edu.cn

## Abstract

Modeling the dynamics from sparsely time-resolved snapshot data is crucial for understanding complex cellular processes and behavior. Existing methods leverage optimal transport, Schrödinger bridge theory, or their variants to simultaneously infer stochastic, unbalanced dynamics from snapshot data. However, these approaches remain limited in their ability to account for cell-cell interactions. This integration is essential in real-world scenarios since intercellular communications are fundamental life processes and can influence cell state-transition dynamics. To address this challenge, we formulate the Unbalanced Mean-Field Schrödinger Bridge (UMFSB) framework to model unbalanced stochastic interaction dynamics from snapshot data. Inspired by this framework, we further propose **CytoBridge**, a deep learning algorithm designed to approximate the UMFSB problem. By explicitly modeling cellular transitions, proliferation, and interactions through neural networks, CytoBridge offers the flexibility to learn these processes directly from data. The effectiveness of our method has been extensively validated using both synthetic gene regulatory data and real scRNA-seq datasets. Compared to existing methods, Cyto-Bridge identifies growth, transition, and interaction patterns, eliminates false transitions, and reconstructs the developmental landscape with greater accuracy. Code is available at: https://github.com/zhenyiizhang/CytoBridge-NeurIPS.

## 1 Introduction

Reconstructing dynamics from high-dimensional distribution samples is a central challenge in science and machine learning. In generative models, methods such as Variational Autoencoders (VAEs) (Kingma and Welling 2013), diffusion models (Ho et al. 2020; Sohl-Dickstein et al. 2015; Song et al. 2021), and flow matching (Lipman et al. 2023; Tong et al. 2024a) have achieved success in generating high-fidelity images by coupling high-dimensional distributions (Liu et al. 2023). Meanwhile, in biology, inferring dynamics (also known as trajectory inference problem) from several static snapshots of single-cell RNA sequencing (scRNA-seq) data (Ding et al. 2022) to build the continuous dynamics of an individual cell and construct the corresponding cell-fate landscapes has also attracted broad interests (Schiebinger et al. 2019; Klein et al. 2025; Zhang et al. 2025b).

To study the trajectory inference problem, optimal transport (OT) theory serves as a foundational tool (Bunne et al. 2024; Zhang et al. 2025b; Heitz et al. 2024; Zhang et al. 2025c). In particular, several works propose to infer continuous cellular dynamics over time by employing the Benamou–Brenier

---

*These authors contributed equally. †Corresponding authors.

formulation (Benamou and Brenier 2000). Given that cell growth and death are critical biological processes, modeling the coupling of underlying unnormalized distributions has led to the development of unbalanced dynamical OT by introducing Wasserstein–Fisher–Rao metric (Chizat et al. 2018a; Chizat et al. 2018b). To further account for the prevalent stochastic effects on single-cell level, methods inspired by the Schrödinger Bridge (SB) problem seek to identify the most likely stochastic transition path between two arbitrary distributions (Gentil et al. 2017; Léonard 2014). To tackle both stochastic and unbalanced effects simultaneously, methods have been developed to model unbalanced stochastic dynamics (Pariset et al. 2023; Lavenant et al. 2024), along with a recent deep learning method (Zhang et al. 2025a), which leverages the regularized unbalanced optimal transport (RUOT) framework to infer continuous unbalanced stochastic dynamics from samples without requiring prior knowledge.

Nevertheless, the majority of existing trajectory inference methods do not account for cell-cell interactions in cell-state transition dynamics, which involve important biological processes such as intercellular communications (Almet et al. 2024; Tejada-Lapuerta et al. 2025; Cang et al. 2023). Developing frameworks to infer unbalanced and stochastic continuous dynamics with particle interactions from multiple snapshot data remains a critical yet underexplored challenge.

To address this challenge, we propose the **Unbalanced Mean Field Schrödinger Bridge (UMFSB)**, a modeling framework based on the Mean Field Schrödinger Bridge that extends to unnormalized distributions. We further develop a new deep learning method (**CytoBridge**) to approximate the general UMFSB and learn continuous stochastic dynamics with cellular interactions from snapshot data with unbalanced distributions. Our primary contributions are summarized as follows:

- We formulate the UMFSB problem to model unbalanced stochastic dynamics of interactive particles from snapshot data. By reformulating UMFSB with a Fisher regularization form, we transform the original stochastic differential equation (SDE) constraints into computationally more tractable ordinary differential equation (ODE) constraints.

- We propose CytoBridge, a deep learning algorithm to approximate the UMFSB problem. By explicitly modeling cellular growth/death and interaction terms via neural networks, CytoBridge does not need prior knowledge of these functions.

- We validate the effectiveness of CytoBridge extensively on synthetic and real scRNA-seq datasets, demonstrating promising performance over existing trajectory inference methods.

## 2   Related Works

**Various Dynamical OT Extensions and Deep Learning Solvers**   Numerous efforts have been made to learn dynamics from snapshot data. To tackle the dynamical optimal transport problem, several methods have been proposed by leveraging neural ODEs or flow matching techniques (Tong et al. 2020; Huguet et al. 2022; Wan et al. 2023; Zhang et al. 2024a; Tong et al. 2024a; Albergo et al. 2023; Palma et al. 2025; Rohbeck et al. 2025; Petrović et al. 2025). To account for sink and source terms in unnormalized distributions, Peng et al. 2024; Sha et al. 2024; Tong et al. 2023; Eyring et al. 2024 developed the neural ODE-based solver for unbalanced dynamical OT. Wang et al. 2025 developed a flow matching approach to simultaneously learn velocity and growth. For the Schrödinger Bridge (SB) problem, approaches have been proposed based on its static or dynamic formulations (Shi et al. 2024; De Bortoli et al. 2021; Gu et al. 2025; Koshizuka and Sato 2023; Neklyudov et al. 2023; Neklyudov et al. 2024; Zhang et al. 2024b; Bunne et al. 2023; Chen et al. 2022b; Zhou et al. 2024a; Zhu et al. 2024; Yeo et al. 2021; Jiang and Wan 2024), with corresponding flow matching methods (Tong et al. 2024b). To further incorporate unbalanced effects in the SB framework, methods utilizing branching SDE theory (Lavenant et al. 2024; Ventre et al. 2023; Chizat et al. 2022), forward-backward SDE (Pariset et al. 2023), neural ODEs with Fisher information regularization (Zhang et al. 2025a), or with first-order optimality conditions (Sun et al. 2025) have been introduced. However, theoretical formulation, along with an effective deep learning solver to simultaneously account for **unbalanced stochastic effects** and **particle interactions** in the dynamical OT framework, remains largely lacking.

**Modeling Cellular Interactions in Trajectory Inference**   Several studies have explored the incorporation of cellular interaction effects into time-series scRNA-seq trajectory inference. For instance, Atanackovic et al. 2025 employs a graph convolutional network within a flow-matching framework

to model the impacts of neighborhood cells within the initial cell population. You et al. 2024 introduced a population-level regularization in the energy form. Yang 2025 formulated the topological Schrödinger Bridge problem on a discrete graph. Fu et al. 2025 improves the accuracy of pseudotime inference by integrating cellular communication pattern. However, **an explicit quantification of cell interaction dynamics** in scRNA-seq trajectory inference is yet to be explored.

**Mean-Field Control Problem** Several works have explored mean-field problem (Zhou et al. 2024b; Ruthotto et al. 2020; Lu et al. 2024; Yang et al. 2024; Huang et al. 2024; Han et al. 2024; Shen and Wang 2023; Li and Liu 2025; Li et al. 2023; Shen et al. 2022) and its variants, as well as the incorporation of particle interaction terms in the Schrödinger Bridge Problem. Backhoff et al. 2020; Hernández and Tangpi 2025 investigated theoretical properties of the Mean Field Schrödinger Bridge Problem. Rapakoulias et al. 2025 develop a deep learning solver for the MFSB problem. Liu et al. 2022b; Liu et al. 2024 proposed the generalized formulation of Schrödinger bridges that includes interacting terms. Yang et al. 2022 formulated the ensemble regression problem and developed a neural ODE-based approach to learn the dynamics of interacting particle systems from distribution data. However, these approaches either require **prior knowledge** to specify the interaction potential field or do not account for **unbalanced stochastic dynamics**.

## 3 Preliminaries and Backgrounds

In this section, we provide an overview of unbalanced stochastic effects and interaction forms within the dynamical OT framework. By integrating two perspectives of RUOT and MFSB described below, we motivate the formulation of the Unbalanced Mean Field Schrödinger Bridge (UMFSB) framework.

### 3.1 Regularized Unbalanced Optimal Transport

The regularized unbalanced optimal transport, also known as the unbalanced Schrödinger Bridge problem (Chen et al. 2022c), considers both the unbalanced stochastic effects in the dynamical OT framework (Baradat and Lavenant 2021; Zhang et al. 2025a):

**Definition 3.1** (Regularized Unbalanced Optimal Transport). *Consider*

$$\inf_{(\rho, \mathbf{b}, g)} \int_0^T \int_{\mathbb{R}^d} \frac{1}{2} \|\mathbf{b}(\mathbf{x}, t)\|_2^2 \rho(\mathbf{x}, t) \mathrm{d}\mathbf{x} \mathrm{d}t + \int_0^T \int_{\mathbb{R}^d} \alpha \Psi \left( g(\mathbf{x}, t) \right) \rho(\mathbf{x}, t) \mathrm{d}\mathbf{x} \mathrm{d}t,$$

*where $\Psi : \mathbb{R} \to [0, +\infty]$ corresponds to the growth penalty function, and $\alpha$ is the weight of the growth penalty. The infimum is taken over all pairs $(\rho, \mathbf{b}, g)$ such that $\rho(\cdot, 0) = \nu_0, \rho(\cdot, 1) = \nu_1, \rho(\mathbf{x}, t)$ absolutely continuous, and*

$$\partial_t \rho = -\nabla_{\mathbf{x}} \cdot (\rho \mathbf{b}) + \frac{1}{2} \sigma^2(t) \Delta \rho + g \rho$$

*with vanishing boundary condition:* $\lim_{|\mathbf{x}| \to \infty} \rho(\mathbf{x}, t) = 0$.

Here $\mathbf{b}(\mathbf{x}, t)$ is the velocity, $g(\mathbf{x}, t)$ is the growth function, and $\sigma(t)$ is the diffusion rate. Note that here $\nu_0$ and $\nu_1$ are not necessarily the normalized probability densities, but are generally unnormalized densities of masses.

### 3.2 Mean Field Schrödinger Bridge Problem

Schrödinger bridge problem aims to find the most probable path between a given initial distribution $\nu_0$ and a target distribution $\nu_1$, relative to a reference process. Formally, it can be stated as:

$$\min_{\mu_0^{\mathbf{X}} = \nu_0, \ \mu_1^{\mathbf{X}} = \nu_1} \mathcal{D}_{\mathrm{KL}} \left( \mu_{[0,1]}^{\mathbf{X}} \| \mu_{[0,1]}^{\mathbf{Y}} \right),$$

where $\mu_{[0,1]}^{\mathbf{X}}$ is the probability measure induced by $\mathbf{X}_t$ $(0 \leq t \leq 1)$ and the reference measure $\mu_{[0,1]}^{\mathbf{Y}}$. However, the classical Schrödinger bridge problem considers the independent particles. The mean field extends the SB problem to the *interacting particles* with given initial and final distributions. We consider the $\mathrm{d}\mathbf{Y}_t = \boldsymbol{\sigma}(\mathbf{Y}_t, t) \mathrm{d}\mathbf{W}_t$, where $\boldsymbol{\sigma}(\mathbf{Y}_t, t) \in \mathbb{R}^{d \times d}$ is the diffusion rate, $\mathbf{W}_t \in \mathbb{R}^d$ is

the standard multi-dimensional Brownian motion and it is called the diffusion Schrödinger bridge problem, where it has a dynamic formulation. So the mean field Schrödinger bridge problem can be stated through this dynamical formulation (Backhoff et al. 2020; Hernández and Tangpi 2025):

**Definition 3.2** (Mean Field Schrödinger Bridge Problem). *Consider*

$$\inf_{(\mathbf{b},\rho)} \int_{\mathbb{R}^d} \int_0^T \frac{1}{2} \|\mathbf{b}(\mathbf{x},t)\|_2^2 \rho(\mathbf{x},t) \mathrm{d}t \mathrm{d}\mathbf{x}, \tag{1}$$

*The infinium is taken over all* $(\mathbf{b}, \rho)$ *subject to* $\rho(\mathbf{x},0) = \nu_0, \rho(\mathbf{x},1) = \nu_1$, *and*

$$\frac{\partial \rho(\mathbf{x},t)}{\partial t} = -\nabla_{\mathbf{x}} \cdot \left[ \left( \mathbf{b}(\mathbf{x},t) - \int_{\mathbb{R}^d} k(\mathbf{x},\mathbf{y}) \nabla_{\mathbf{x}} \Phi(\mathbf{x}-\mathbf{y}) \rho(\mathbf{y},t) \, \mathrm{d}\mathbf{y} \right) \rho(\mathbf{x},t) \right] + \frac{\sigma^2(t)}{2} \Delta \rho(\mathbf{x},t), \tag{2}$$

*where* $\Phi$ *is the interaction potential and it is satisfied* $\Phi(-\mathbf{x}) = \Phi(\mathbf{x})$. *The* $k(\cdot,\cdot) : \mathbb{R}^d \times \mathbb{R}^d \to \mathbb{R}$ *is the interaction weight function.*

## 4 Unbalanced Mean Field Schrödinger Bridge

In this section, we introduce the unbalanced mean field Schrödinger Bridge problem. Inspired by regularized unbalanced optimal transport, the dynamical formulation Definition 3.2 suggests a natural way to relax the mass constraint by introducing a growth/death term $g\rho$ in (2). Meanwhile, we also introduce a loss function in (1) which considers both the growth and transition metric.

**Definition 4.1** (Unbalanced Mean Field Schrödinger Bridge). *Consider*

$$\inf_{(\mathbf{b},g,\rho,\Phi)} \int_{\mathbb{R}^d} \int_0^T \frac{1}{2} \|\mathbf{b}(\mathbf{x},t)\|_2^2 \rho(\mathbf{x},t) \mathrm{d}t \mathrm{d}\mathbf{x} + \int_{\mathbb{R}^d} \int_0^T \Psi(g(\mathbf{x},t)) \rho(\mathbf{x},t) \mathrm{d}t \mathrm{d}\mathbf{x},$$

*where* $\Psi(\cdot) : \mathbb{R} \to \mathbb{R}^+$ *is the growth cost function. The infinium is taken over all* $(\mathbf{b}, g, \rho, \Phi)$ *subject to* $\rho(\mathbf{x},0) = \nu_0, \rho(\mathbf{x},1) = \nu_1$, *and*

$$\frac{\partial \rho(\mathbf{x},t)}{\partial t} = -\nabla_{\mathbf{x}} \cdot \left[ \left( \mathbf{b}(\mathbf{x},t) - \int_{\mathbb{R}^d} k(\mathbf{x},\mathbf{y}) \nabla_{\mathbf{x}} \Phi(\mathbf{x}-\mathbf{y}) \rho(\mathbf{y},t) \, \mathrm{d}\mathbf{y} \right) \rho(\mathbf{x},t) \right] + \frac{\sigma^2(t)}{2} \Delta \rho(\mathbf{x},t) + g\rho,$$

*where* $\Phi$ *is the interaction potential and it is satisfied* $\Phi(-\mathbf{x}) = \Phi(\mathbf{x})$. *The* $k(\cdot,\cdot) : \mathbb{R}^d \times \mathbb{R}^d \to \mathbb{R}$ *is the interaction weight function.*

In Definition 4.1, if $k(\mathbf{x},\mathbf{y}) = 0$, which means there is no cell-cell interaction, it degenerates to the *regularized unbalanced optimal transport* problem. If the growth penalty is set such that $g(\mathbf{x},t)$ must be zero (i.e., by setting $\Psi(g(\mathbf{x},t)) = +\infty$ for $g(\mathbf{x},t) = 0$ and $\Psi(0) = 0$), the framework degenerates to simpler forms: if interactions are present ($k(\mathbf{x},\mathbf{y}) \neq 0$), the formulation reduces to the *Mean Field Schrödinger Bridge* problem; if interactions are absent ($k(\mathbf{x},\mathbf{y}) = 0$), it reduces to the *regularized optimal transport* problem. It becomes the *unbalanced dynamics optimal transport* when $k(\mathbf{x},\mathbf{y}) = \sigma(t) = 0$ and $\Psi(g) = g^2$. It becomes the *dynamics optimal transport* when growth, interaction and diffusion all goes to zero. We can reformulate Definition 4.1 with the following Fisher information regularization.

**Theorem 4.1.** *The unbalanced mean field Schrödinger Bridge Definition 4.1 is equivalent to*

$$\inf_{(\rho,\mathbf{b},g,\Phi)} \int_0^T \int_{\mathbb{R}^d} \left[ \frac{1}{2} \|\mathbf{v}\|_2^2 + \frac{\sigma^4(t)}{8} \|\nabla_{\mathbf{x}} \log \rho\|_2^2 + \frac{1}{2} \langle \mathbf{v}, \sigma^2(t) \nabla_{\mathbf{x}} \log \rho \rangle + \alpha \Psi(g) \right] \rho(\mathbf{x},t) \mathrm{d}\mathbf{x} \mathrm{d}t, \tag{3}$$

*where* $\Psi(\cdot) : \mathbb{R} \to \mathbb{R}$ *is the growth cost function, and* $\alpha$ *is the weight of the growth cost. The infinium is taken over all* $(\mathbf{b}, g, \rho, \Phi)$ *subject to* $\rho(\mathbf{x},0) = \nu_0, \rho(\mathbf{x},1) = \nu_1$, *and*

$$\frac{\partial \rho(\mathbf{x},t)}{\partial t} = -\nabla_{\mathbf{x}} \cdot \left[ \left( \mathbf{v}(\mathbf{x},t) - \int_{\mathbb{R}^d} k(\mathbf{x},\mathbf{y}) \nabla_{\mathbf{x}} \Phi(\mathbf{x}-\mathbf{y}) \rho(\mathbf{y},t) \, \mathrm{d}\mathbf{y} \right) \rho(\mathbf{x},t) \right] + g\rho, \tag{4}$$

*where* $\Phi$ *is the interaction potential and it is satisfied* $\Phi(-\mathbf{x}) = \Phi(\mathbf{x})$. *The* $k(\cdot,\cdot) : \mathbb{R}^d \times \mathbb{R}^d \to \mathbb{R}$ *is the interaction weight function.*

Here $\mathbf{v}(\mathbf{x}, t)$ is a new vector field function. This Fisher information form transforms the original SDE problem into the ODE problem which is computationally more tractable. Next, we will focus on solving this problem. The proof is simple and we left it in Appendix D.1 for reference.

**Remark 4.1.** *From the proof of Theorem 4.1, the relation between the new vector filed $\mathbf{v}(\mathbf{x}, t)$ and $\mathbf{b}(\mathbf{x}, t)$ is $\mathbf{v}(\mathbf{x}, t) = \mathbf{b}(\mathbf{x}, t) - \frac{1}{2}\sigma^2(t)\nabla_{\mathbf{x}}\log\rho(\mathbf{x}, t)$. The new $\mathbf{v}(\mathbf{x}, t)$ is also known as probability flow ODE and $\nabla_{\mathbf{x}}\log\rho(\mathbf{x}, t)$ is the score function. Conversely, if the probability flow ODE $\mathbf{v}(\mathbf{x}, t)$ and the score function $\nabla_{\mathbf{x}}\log\rho(\mathbf{x}, t)$ are known, then we can recover the original drift term.*

## 5 Learning Cell Dynamics and Interactions through Neural Networks

Assume that we collect scRNA-seq samples from unnormalized distributions $\mathbf{X}_t \in \mathbb{R}^{n_i \times d}$ ($t = 1, 2, \cdots, T$) at $T$ time points, where $n_i$ is the number of cells at time $i$ and $d$ is the number of genes, here we propose the **CytoBridge** algorithm to approximate UMFSB through neural networks. We parameterize transition velocity $\mathbf{v}(\mathbf{x}, t)$, cell growth rate $g(\mathbf{x}, t)$, log density function $\frac{1}{2}\sigma^2(t)\log\rho(\mathbf{x}, t)$ and cellular interaction potential $\Phi(\mathbf{x}, t)$ using neural networks $\mathbf{v}_\theta$, $g_\theta$, $s_\theta$ and $\Phi_\theta$ respectively, as shown in Fig. 1. To effectively approximate the loss function in Theorem 4.1, we model the evolution of the mass densities through a number of weighted interacting particles, which is supported by the following proposition.

**Proposition 5.1.** *Consider a system of $N$ weighted particles in $\mathbb{R}^d$, where each particle $i$ has a position $\mathbf{X}_t^i \in \mathbb{R}^d$ and a positive weight $w_i(t) > 0$. The weight $w_i(t)$ evolves according to the ordinary differential equation (ODE) $\frac{\mathrm{d}w_i}{\mathrm{d}t} = g(\mathbf{X}_t^i, t)w_i(t)$, where*

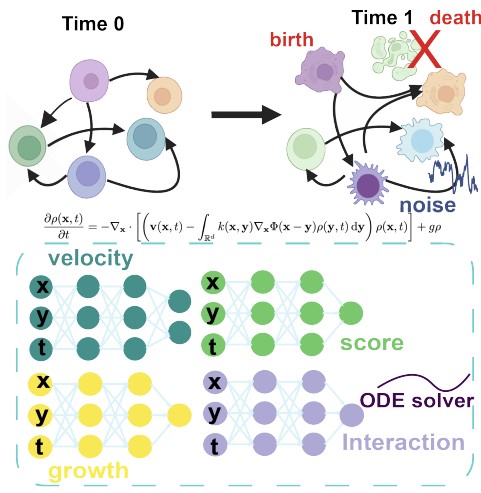

Figure 1: Overview of CytoBridge.

$g: \mathbb{R}^d \times [0, T] \to \mathbb{R}$ *is a given growth rate function. The position $\mathbf{X}_t^i$ evolves according to the stochastic differential equation (SDE)*

$$\mathrm{d}\mathbf{X}_t^i = \mathbf{b}(\mathbf{X}_t^i, t)\,\mathrm{d}t - \frac{1}{N-1}\sum_{j \neq i} k(\mathbf{X}_t^i, \mathbf{X}_t^j)w_j(t)\nabla_x\Phi(\mathbf{X}_t^i - \mathbf{X}_t^j)\,\mathrm{d}t + \sigma(t)\,\mathrm{d}\mathbf{W}_t^i.$$

*Under assumptions stated in D.1, in the limit of $N \to \infty$, the weighted empirical measure $\mu_t^N = \frac{1}{N}\sum_{i=1}^N w_i(t)\delta_{\mathbf{X}_t^i}$ converges weakly to a deterministic measure $\rho(\mathbf{x}, t)\,\mathrm{d}\mathbf{x}$, where $\rho(\mathbf{x}, t)$ is a weak solution to the partial differential equation (PDE)*

$$\frac{\partial\rho}{\partial t} = -\nabla_x\cdot\left[\left(\mathbf{b}(\mathbf{x}, t) - \int_{\mathbb{R}^d} k(\mathbf{x}, \mathbf{y})\nabla_x\Phi(\mathbf{x} - \mathbf{y})\rho(\mathbf{y}, t)\,\mathrm{d}\mathbf{y}\right)\rho(\mathbf{x}, t)\right] + \frac{\sigma^2(t)}{2}\Delta\rho(\mathbf{x}, t) + g(\mathbf{x}, t)\rho(\mathbf{x}, t),$$

*with the initial condition $\rho(\mathbf{x}, 0) = \rho_0(\mathbf{x})$.*

The derivation is left in Appendix D.2. By combining the results in Proposition 5.1 and Theorem 4.1, the ODE constraints we simulate is indeed $\mathrm{d}\mathbf{X}_t^i = \mathbf{v}(\mathbf{X}_t^i, t)\mathrm{d}t - \frac{1}{N-1}\sum_{j \neq i, j=1}^N k_{i,j}w_j\nabla_{\mathbf{x}}\Phi(\mathbf{X}_t^i - \mathbf{X}_t^j)\mathrm{d}t$, where $k_{i,j} = k(\mathbf{X}_t^i, \mathbf{X}_t^j)$ is the cell-cell interaction strength and $w_j = w(\mathbf{X}_t^j)$ is the particle weight.

### 5.1 Simulating ODEs: Random Batch Methods

In ODE simulation, the computation complexity is $\mathcal{O}(N^2)$ due to the cellular interaction term. To speed up simulation, we adopt the Random Batch Methods (RBMs) (Jin et al. 2020; Jin et al. 2021; Jin and Li 2022), which transforms the interaction among all particles into interaction with particles within random grouping only. The algorithm reduces the computational complexity to $\mathcal{O}((p-1)N)$, where $p$ is the number in the batch $\mathcal{C}_p$. Assuming $\mathbf{v}(\mathbf{x}, t)$ and $\Phi$ satisfy certain conditions, it has the convergence result such that $\mathcal{W}_2\left(\tilde{\mu}_N^{(1)}(t), \mu_N^{(1)}(t)\right) \leq C\sqrt{\tau}$ (Jin et al. 2020; Jin et al. 2021), where $\tau$ is the step size. The $\tilde{\mu}_N^{(1)}(t)$ is the empirical distribution produced by Algorithm 1 and $\mu_N^{(1)}(t)$ is the distribution by simulating the original ODEs.

**Algorithm 1** Simulating ODEs: RBM
___
1: **for** $m \in 1 : [T/\tau]$ **do**
   Randomly divide $\{1, 2, \cdots, N = np\}$ into $n$ batches
2:      **for** each batch **do**
        Update particles with $\mathrm{d}\mathbf{X}_t^i = \mathbf{v}(\mathbf{X}_t^i, t)\mathrm{d}t - \frac{1}{p-1} \sum\limits_{j \neq i, j \in \mathcal{C}_p}^{N} k_{i,j} w_j \nabla_{\mathbf{x}} \Phi(\mathbf{X}_t^i - \mathbf{X}_t^j)\mathrm{d}t$
___

## 5.2    Reformulating the Loss with Weighted Particle Simulation

The total loss to solve Theorem 4.1 is composed of three parts, e.g., the energy loss, the reconstruction loss, and the Fokker-Planck constraint such that $\mathcal{L} = \mathcal{L}_{\text{Energy}} + \lambda_r \mathcal{L}_{\text{Recons}} + \lambda_f \mathcal{L}_{\text{FP}}$. Here $\mathcal{L}_{\text{Energy}}$ loss promotes the least action principle of transition energy, $\mathcal{L}_{\text{Recons}}$ promotes the matching loss. i.e., $\rho(\mathbf{x}, 1) = \nu_1$, and $\mathcal{L}_{\text{FP}}$ promotes the three neural networks that satisfy the Fokker-Planck equation constraint. We reformulate the loss terms through weighted particle representation and an RBM-based Neural ODE solver.

**Energy Loss**    Generalizing the idea in (Sha et al. 2024), the energy loss in CytoBridge is equivalent to $\mathbb{E}_{\mathbf{x_o} \sim \rho_0} \int_0^T \left[ \frac{1}{2} \|\mathbf{v}_\theta(\mathbf{x}(t), t)\|_2^2 + \frac{1}{2} \|\nabla_{\mathbf{x}} s_\theta\|_2^2 + \langle \mathbf{v}_\theta(\mathbf{x}(t), t), s_\theta \rangle + \alpha \Psi(g_\theta) \right] w_\theta(t)\mathrm{d}t$, where $w_\theta(t) = \exp\left( \int_0^t g_\theta(\mathbf{x}(t), s)\mathrm{d}s \right)$ and $\mathbf{x}(t)$ satisfies $\mathrm{d}\mathbf{x}^i/\mathrm{d}t = \mathbf{v}(\mathbf{x}^i, t) - \frac{1}{N-1} \sum_{j \neq i} k(\mathbf{x}^i, \mathbf{x}^j) w_j \nabla_{\mathbf{x}^i} \Phi(\mathbf{x}^i - \mathbf{x}^j)$. However, direct optimization of this term is challenging due to the involvement of the inner product $\langle \mathbf{v}_\theta(\mathbf{x}(t), t), s_\theta \rangle$ which introduces mutual dependencies between the optimization of $\mathbf{v}$ and $s_\theta$. To address this issue and simplify the computation, we adopt an upper bound of the energy for training purposes (Appendix A.5).

**Reconstruction Loss**    The reconstruction loss aims to align the final generated density to the true data density. Here we need to consider the unbalanced effect, so we use the unbalanced optimal transport to align it. $\mathcal{L}_{\text{Recons}} = \lambda_m \mathcal{L}_{\text{Mass}} + \lambda_d \mathcal{L}_{\text{OT}}$. The $\mathcal{L}_{\text{Mass}}$ is used to obtain the cell weights and align the cell number/ mass in the datasets. We then use the weights to normalize the distribution and apply $\mathcal{L}_{\text{OT}}$ to match the distribution. In this work, we employ the local mass matching strategy from (Zhang et al. 2025a). Specifically, the trajectory mapping function $\phi_\theta^{\mathbf{v}}$ predicts particle coordinates $\widehat{A}_1, \ldots, \widehat{A}_{T-1}$ from an initial set $A_0$ over time indices $\mathcal{T}$, governed by $\mathrm{d}\mathbf{x}/\mathrm{d}t = \mathbf{v}_\theta$ if no interaction is considered, or with a modified velocity $\widetilde{\mathbf{v}}$ incorporating the interaction potential $\Phi$: $\widetilde{\mathbf{v}}(\mathbf{x}^i, t) = \mathbf{v}(\mathbf{x}^i, t) - \frac{1}{N-1} \sum_{j \neq i} k(\mathbf{x}^i, \mathbf{x}^j) \nabla_{\mathbf{x}^i} \Phi(\mathbf{x}^i - \mathbf{x}^j)$. Additionally, a weight mapping function $\phi_\theta^g$ models particle weights via $\mathrm{d} \log w_i(t)/\mathrm{d}t = g_\theta(\mathbf{x}_i(t), t)$, starting from initial weights $w_i(0) = 1/N$ where $N$ represents batch size. Mathematically, we use the empirical measure $\mu_0^N = \frac{1}{N} \sum_{i=1}^N \delta_{x_i}$ to approximate the true distribution $\mu_0$, hence the uniform weights. This convergence to the true distribution is guaranteed when $N \to \infty$. The mass matching loss $\mathcal{L}_{\text{Mass}}$ is composed of two terms. The first term is defined as the local mass matching loss $\mathcal{L}_{\text{Local Mass}} = \sum_{k=1}^{T-1} M_k$, where $M_k$ quantifies the error between predicted weights $w_i(t_k)$ and target weights based on the cardinality of mapped points. As detailed in Appendix A.4, the target weights, derived from the number of closest real data points, encourages the growth network to assign higher weights to particles moving into denser state space regions, thus provides fine-grained guidance on the growth network. Besides, the second term is defined as the global mass matching loss $\mathcal{L}_{\text{Global Mass}} = \sum_{k=1}^{T-1} G_k$, where $G_k$ is used to align the change of weights in total. An optimal transport loss $\mathcal{L}_{\text{OT}} = \sum_{k=1}^{T-1} \mathcal{W}_2(\widehat{\mathbf{w}}^k, \mathbf{w}(t_k))$ further aligns the predicted and observed distributions. Details can be found in Appendix A.4.

**Fokker-Planck Constraint**    To enforce the physical relationships among the four neural networks, it is essential to introduce a physics-informed loss (PINN-loss), which incorporates the Fokker-Planck constraint as a guiding principle. $\mathcal{L}_{\text{FP}} = \|\partial_t \rho_\theta + \nabla_{\mathbf{x}} \cdot (\rho_\theta \widetilde{\mathbf{v}}_\theta) - g_\theta \rho_\theta\| + \lambda_w \|\rho_\theta(\mathbf{x}, 0) - p_0\|$, where $\rho_\theta = \exp \frac{2}{\sigma^2} s_\theta$.

**Training**    CytoBridge aims to train four neural networks to model cell dynamics and interactions. To stabilize the training procedure, we leverage a two-phase training strategy. In the pre-training

phase, we seek to provide a suitable initialization of these four neural networks. We first initialize $g_\theta$ and $\mathbf{v}_\theta$ without the interaction term to provide an approximated matching. Then we train $\Phi_\theta$ with fixed $g_\theta$, leading to refined dynamics. The score network $s_\theta$ is trained based on conditional flow matching. In the training phase, these initialized networks are further refined by minimizing the proposed total loss. We summarize the training procedure in Algorithm 2 and Appendix A. We conduct ablation studies on different components of our training procedure in Appendix B.7. The selection of loss weighting is discussed in Appendix C.2.

---

**Algorithm 2** Training CytoBridge

---

**Require:** Datasets $A_0, \ldots, A_{T-1}$, batch size $N$, ODE iteration $n_{\text{ode}}$, log density iteration $n_{\text{log-density}}$
**Ensure:** Trained neural ODE $\mathbf{v}_\theta$, growth function $g_\theta$, score network $s_\theta$ and interaction $\Phi_\theta$.

1: **Pre-Training Phase:**
2: **for** $i = 1$ to $n_{\text{ode}}$ **do**  ▷ *1. Initialize growth $g_\theta$ and velocity $v_\theta$*
3:     **for** $t = 0$ to $T - 2$ **do**
4:         $\hat{A}_{t+1} \leftarrow \phi_\theta^{\mathbf{v}}(\hat{A}_t, t+1)$, $w(\hat{A}_{t+1}) \leftarrow \phi_\theta^g(w(\hat{A}_t), t+1)$
5:         $\mathcal{L}_{\text{Recons}} \leftarrow \lambda_m M_t + \lambda_d \mathcal{W}_2(\hat{\mathbf{w}}^t, \mathbf{w}(t))$; update $\mathbf{v}_\theta, g_\theta$ w.r.t. $\mathcal{L}_{\text{Recons}}$
6: **for** $i = 1$ to $n_{\text{ode}}$ **do**  ▷ *2. Initialize interaction potential $\Phi_\theta$*
7:     **for** $t = 0$ to $T - 2$ **do**
8:         $\hat{A}_{t+1} \leftarrow \phi_\theta^{\widetilde{\mathbf{v}}}(\hat{A}_t, t+1)$, $w(\hat{A}_{t+1}) \leftarrow \phi_\theta^g(w(\hat{A}_t), t+1)$
9:         $\mathcal{L}_{\text{Recons}} \leftarrow \mathcal{W}_2(\hat{\mathbf{w}}^t, \mathbf{w}(t))$; update $\mathbf{v}_\theta, \Phi_\theta$ w.r.t. $\mathcal{L}_{\text{Recons}}$
10: **for** $t = 0$ to $T - 2$ **do** $\hat{A}_{t+1} \leftarrow \phi_\theta^{\mathbf{v}}(\hat{A}_t, t+1)$  ▷ *Generate datasets $\hat{A}_t$.*
11: **for** $i = 1$ to $n_{\text{log-density}}$ **do**  ▷ *4. Initialize the score network $s_\theta$*
12:     $(\mathbf{x}_0, \mathbf{x}_1) \sim q(\mathbf{x}_0, \mathbf{x}_1)$, $t \sim \mathcal{U}(0, 1)$, $\mathbf{x} \sim p(\mathbf{x}, t \mid \mathbf{x}_0, \mathbf{x}_1)$ using $\hat{A}_0, \ldots, \hat{A}_{T-1}$
13:     $\mathcal{L}_{\text{score}} \leftarrow \|\lambda_s \nabla_{\mathbf{x}} s_\theta(\mathbf{x}, t) + \boldsymbol{\epsilon}_1\|_2^2$; update $s_\theta$ w.r.t. $\mathcal{L}_{\text{score}}$

14: **Training Phase:**
15: Estimate initial distribution $p_0(\mathbf{x})$ from $A_0$ using Gaussian Mixture Model (GMM).
16: **for** $i = 1$ to $n_{\text{ode}}$ **do**
17:     **for** $t = 0$ to $T - 2$ **do**
18:         $\hat{A}_{t+1} \leftarrow \phi_\theta^{\widetilde{\mathbf{v}}}(\hat{A}_t, t+1)$, $w(\hat{A}_{t+1}) \leftarrow \phi_\theta^g(w(\hat{A}_t), t+1)$
19:         $\mathcal{L}_{\text{Energy}} \leftarrow \mathbb{E}_{\mathbf{x}_\tau \sim \rho_\tau} \int_{\tau=t}^{\tau=t+1} \left[ \frac{1}{2} \|\mathbf{v}_\theta\|_2^2 + \frac{1}{2} \|\nabla_{\mathbf{x}} s_\theta\|_2^2 + \|\mathbf{v}_\theta\|_2 \|s_\theta\|_2 + \alpha \Psi(g_\theta) \right] w_\theta(\tau) \mathrm{d}\tau$
20:         $\mathcal{L}_{\text{Recons}} \leftarrow \lambda_m (M_t + G_t) + \lambda_d \mathcal{W}_2(\hat{\mathbf{w}}^t, \mathbf{w}(t))$
21:         $\mathcal{L}_{\text{FP}} \leftarrow \|\partial_\tau \rho_\theta(\mathbf{x}, \tau) + \nabla_{\mathbf{x}} \cdot (\rho_\theta(\mathbf{x}, \tau) \widetilde{\mathbf{v}}_\theta(\mathbf{x}, \tau)) - g_\theta(\tau) \rho_\theta(\mathbf{x}, \tau)\| + \lambda_w \|\rho_\theta(\mathbf{x}, 0) - p_0(\mathbf{x})\|$
22:         $\mathcal{L}_{\text{Total}} \leftarrow \mathcal{L}_{\text{Energy}} + \lambda_r \mathcal{L}_{\text{Recons}} + \mathcal{L}_{\text{FP}}$; update $\mathbf{v}_\theta, g_\theta, s_\theta, \Phi_\theta$ w.r.t. $\mathcal{L}_{\text{Total}}$

---

## 6 Results

Next, we evaluate CytoBridge's ability to simultaneously learn cell dynamics and cell-cell interactions. In computations below, we take $\Psi(g(\mathbf{x}, t)) = g^2(\mathbf{x}, t)$ and $\sigma(t)$ is constant. We also assume the interaction term is dependent on the distance between cells in gene expression space and we use radial basis functions (RBFs) to approximate it (Appendix A.3).

**Synthetic Gene Regulatory Network** In order to examine CytoBridge's capabilities of learning cell dynamics as well as their underlying interactions simultaneously, we conducted experiments on the three-gene simulation model following (Zhang et al. 2025a). The dynamics of the original three-gene model are governed by stochastic ordinary differential equations, incorporating self-activation, mutual inhibition, and external activation (Appendix B.1), as shown in Fig. 2 (a). Additionally, we incorporated interactions into the simulation process. We consider the following types of interactions: (1) attractive interactions. (2) Lennard-Jones-like potential (both attractive and repulsive) (3) no interactions. We aim to test whether CytoBridge can recover interaction in each case. For attractive interactions, the cells with similar gene expressions tend to converge toward similar levels. The interaction potential $\Phi$ is defined as: $\Phi(\mathbf{x} - \mathbf{y}) = \|\mathbf{x} - \mathbf{y}\|^2$ As shown in Fig. 2(b), the dynamics of the three-gene model exhibit a quiescent area as well as an area with notable transition and

increasing cell numbers. Moreover, the attractive potential results in the reduction in variance of the observed data at different time points. We compared CytoBridge with other methods across all time points using the Wasserstein distance ($\mathcal{W}_1$) and the Total Mass Variation (TMV) metric, defined in Appendix C. We summarized the results in Table 1. It is shown that CytoBridge achieves the best performance in both distribution matching and mass matching. Balanced Schrödinger bridge (e.g., (Tong et al. 2024b)), which neglects both growth and interaction terms, results in false transition and variance patterns (Fig. 2(c)). DeepRUOT (Zhang et al. 2025a), which is an unbalanced Schrödinger Bridge solver, exhibits correct transition patterns but fails to capture the reduction in variance as it neglects the cell-cell interactions (Fig. 2(d)). By leveraging the UMFSB framework, as shown in Fig. 2(e), CytoBridge correctly models both the transition patterns and the reduction in variance as the correct interaction potential (Fig. 2(f)) and growth rate (Fig. 2(g)) can be directly learned by CytoBridge. Also, CytoBridge is capable of constructing the underlying Waddington landscape (Fig. 2(h)). The low-lying regions on this landscape correspond to areas of high probability density, representing stable states. Other than the attractive interaction potential, we also incorporated the Lennard-Jones-like potential, and no interaction case. Results can be found in Appendix B.1, Figs. 4 and 5 and we find CytoBridge can correctly identify both the LJ potential and no interaction in these cases. Overall, both quantitative results and qualitative analysis indicate the necessity of incorporating both growth and interaction terms.

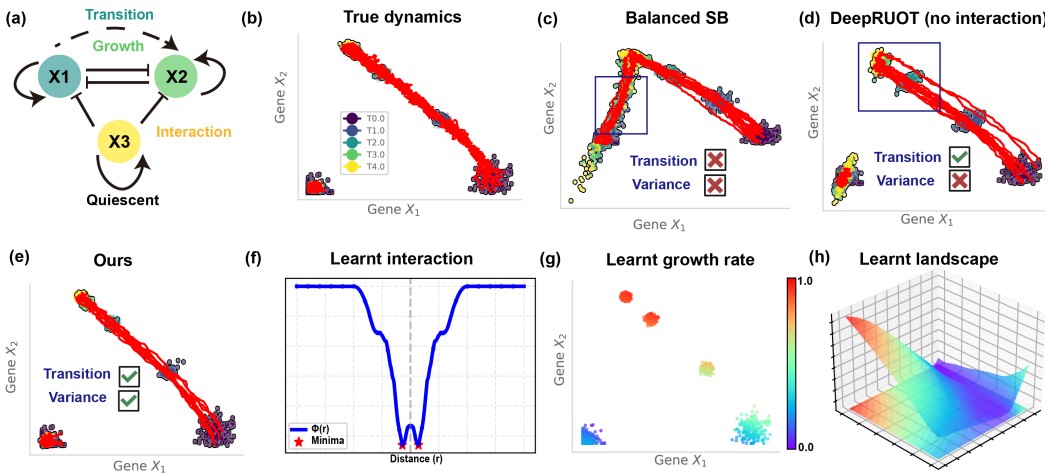

Figure 2: (a) Illustration of the synthetic gene regulatory dynamics. (b) The ground truth cellular dynamics project on $(X_1, X_2)$. The red lines indicate the ground truth trajectories of cells in (b), or inferred trajectories of cells in (c) to (e). (c) The dynamics learned by balanced Schrödinger bridge SF2M (Tong et al. 2024b). (d) The dynamics learned by DeepRUOT. (e) The dynamics learned by CytoBridge. (f) The learned interaction potential. (g) The growth rates inferred by CytoBridge. (h) The constructed landscape at $t = 4$. The z-axis represents the density of cells.

Table 1: Wasserstein distance ($\mathcal{W}_1$) and Total Mass Variation (TMV) of predictions at different time points across five runs on synthetic gene regulatory data with attractive interactions ($\sigma = 0.05$). We show the mean value with one standard deviation, where **bold** indicates the best among all algorithms.

| Model | $t = 1$ | | $t = 2$ | | $t = 3$ | | $t = 4$ | |
| --- | --- | --- | --- | --- | --- | --- | --- | --- |
| | $\mathcal{W}_1$ | TMV | $\mathcal{W}_1$ | TMV | $\mathcal{W}_1$ | TMV | $\mathcal{W}_1$ | TMV |
| SF2M (Tong et al. 2024b) | $0.146_{\pm 0.002}$ | $0.080_{\pm 0.000}$ | $0.320_{\pm 0.004}$ | $0.250_{\pm 0.000}$ | $0.447_{\pm 0.005}$ | $0.515_{\pm 0.000}$ | $0.554_{\pm 0.005}$ | $0.930_{\pm 0.000}$ |
| Meta FM (Atanackovic et al. 2025) | $0.149_{\pm 0.000}$ | $0.080_{\pm 0.000}$ | $0.241_{\pm 0.000}$ | $0.250_{\pm 0.000}$ | $0.288_{\pm 0.000}$ | $0.515_{\pm 0.000}$ | $0.404_{\pm 0.000}$ | $0.930_{\pm 0.000}$ |
| MMFM (Rohbeck et al. 2025) | $0.101_{\pm 0.000}$ | $0.080_{\pm 0.000}$ | $0.223_{\pm 0.000}$ | $0.250_{\pm 0.000}$ | $0.438_{\pm 0.000}$ | $0.515_{\pm 0.000}$ | $0.366_{\pm 0.000}$ | $0.930_{\pm 0.000}$ |
| Metric FM (Kapusniak et al. 2024) | $0.319_{\pm 0.000}$ | $0.080_{\pm 0.000}$ | $0.751_{\pm 0.000}$ | $0.250_{\pm 0.000}$ | $0.690_{\pm 0.000}$ | $0.515_{\pm 0.000}$ | $0.614_{\pm 0.000}$ | $0.930_{\pm 0.000}$ |
| UOT-FM (Eyring et al. 2024) | $0.051_{\pm 0.000}$ | $\mathbf{0.010}_{\pm 0.000}$ | $0.058_{\pm 0.000}$ | $0.036_{\pm 0.000}$ | $0.060_{\pm 0.000}$ | $0.044_{\pm 0.000}$ | $0.054_{\pm 0.000}$ | $0.095_{\pm 0.000}$ |
| MIOFlow (Huguet et al. 2022) | $0.315_{\pm 0.000}$ | $0.080_{\pm 0.000}$ | $0.387_{\pm 0.000}$ | $0.250_{\pm 0.000}$ | $0.483_{\pm 0.000}$ | $0.515_{\pm 0.000}$ | $0.518_{\pm 0.000}$ | $0.930_{\pm 0.000}$ |
| uAM (Neklyudov et al. 2023) | $0.489_{\pm 0.000}$ | $0.081_{\pm 0.000}$ | $0.995_{\pm 0.000}$ | $0.033_{\pm 0.000}$ | $1.402_{\pm 0.000}$ | $0.459_{\pm 0.000}$ | $1.655_{\pm 0.000}$ | $1.516_{\pm 0.000}$ |
| UDSB (Pariset et al. 2023) | $1.131_{\pm 0.009}$ | $0.018_{\pm 0.006}$ | $1.489_{\pm 0.018}$ | $0.135_{\pm 0.014}$ | $1.455_{\pm 0.022}$ | $0.447_{\pm 0.011}$ | $0.543_{\pm 0.015}$ | $1.018_{\pm 0.035}$ |
| TIGON (Sha et al. 2024) | $0.169_{\pm 0.000}$ | $0.097_{\pm 0.000}$ | $0.184_{\pm 0.000}$ | $0.165_{\pm 0.000}$ | $0.167_{\pm 0.000}$ | $0.210_{\pm 0.000}$ | $0.179_{\pm 0.000}$ | $0.384_{\pm 0.000}$ |
| DeepRUOT (Zhang et al. 2025a) | $0.044_{\pm 0.002}$ | $0.014_{\pm 0.007}$ | $0.045_{\pm 0.002}$ | $0.026_{\pm 0.018}$ | $0.053_{\pm 0.002}$ | $0.059_{\pm 0.032}$ | $0.057_{\pm 0.003}$ | $0.075_{\pm 0.044}$ |
| CytoBridge (Ours) | $\mathbf{0.015}_{\pm 0.001}$ | $0.013_{\pm 0.009}$ | $\mathbf{0.014}_{\pm 0.001}$ | $\mathbf{0.021}_{\pm 0.024}$ | $\mathbf{0.018}_{\pm 0.002}$ | $\mathbf{0.043}_{\pm 0.041}$ | $\mathbf{0.038}_{\pm 0.003}$ | $\mathbf{0.058}_{\pm 0.061}$ |

**Mouse Blood Hematopoiesis**  To demonstrate the scalability of CytoBridge to high-dimensional data, we adopt the mouse hematopoiesis dataset (Weinreb et al. 2020) which includes 49,302 cells with lineage tracing data collected at three time points. We use PCA to reduce the dimensions to 50 and serve as the input of CytoBridge. The dataset comprises diverse cell states and demonstrates pronounced cell division. Consequently, accurately modeling both cellular dynamics and growth rates is critical for reliable inference of cell fate. As shown in Table 2, CytoBridge outperforms other state-of-the-art methods in distribution matching, highlighting CytoBridge's capabilities of capturing transition patterns (Fig. 3(a)). Besides, evidenced by the TMV metric, CytoBridge is able to recover the increase in cell numbers by learning the growth rate (Fig. 3(b)). The regions with high learned growth rates correspond to the hematopoietic stem cell populations. This is biologically consistent with the lineage tracing barcode results (Sha et al. 2024). The score function learned by CytoBridge indicates the existence of multiple attractors, which may lead to different cell fates (Fig. 3(c)). To further demonstrate the impact of cell-cell interactions on the transition of cells, we computed the correlation of each cell's drift and interacting force. As shown in Fig. 3(d), the learned cell-cell interactions may promote early-stage cell differentiation and inhibit later-stage cell differentiation. Some additional results can be found in Appendix B.2.

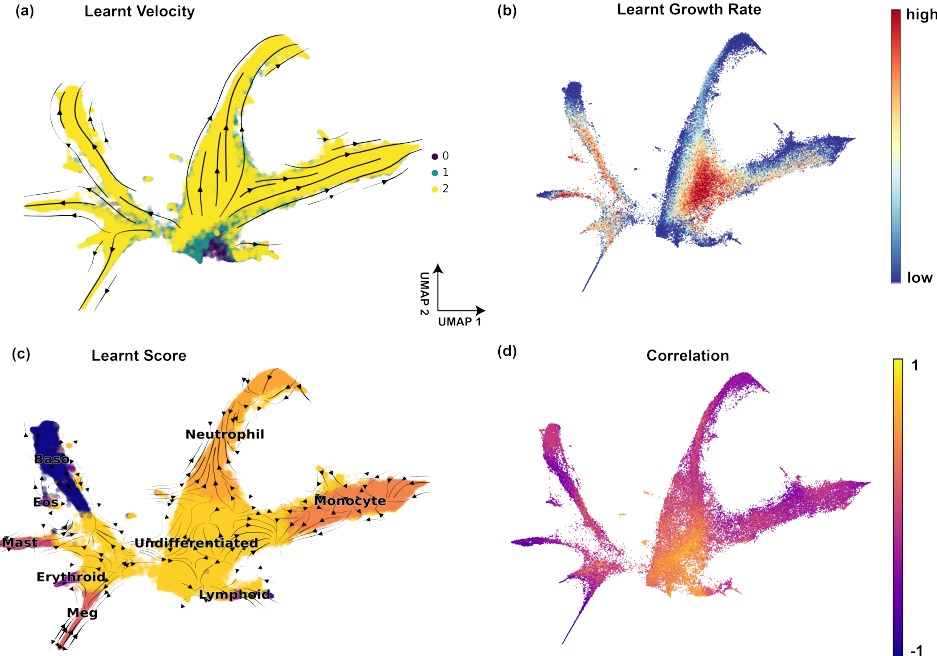

Figure 3: Application in mouse blood hematopoiesis data ($\sigma = 0.1$), visualized in UMAP space. (a) The overall velocity learned by CytoBridge. (b) The growth rates learned by CytoBridge. (c) The score function learned by CytoBridge at $t = 2$. (d) The correlation of velocity and interacting forces.

**Embryoid Body, Pancreatic $\beta$-cell differentiation and A549 EMT**  We further evaluated Cyto-Bridge on the Embryoid Body single-cell data which consists of 16,819 cells collected at five time points (Moon et al. 2019). We used PCA to reduce the dimensions to 50 and serve as the input of CytoBridge. As shown in Table 9, CytoBridge generally outperforms other methods in distribution matching and maintains mass-matching results comparable to other unbalanced algorithms. We plotted the learned velocity and score in Fig. 6, indicating the different cell fates learned by CytoBridge. The correlation of each cell's drift and interacting force is shown in Fig. 6(d) indicates that the learned cell-cell interactions may promote cell differentiation while inhibiting cell differentiation of some outliers. We also consider the 3D-cultured in vitro pancreatic $\beta$-cell differentiation dataset (Veres et al. 2019) and A549 lung cancer cell line epithelial-mesenchymal transition (EMT) dataset induced by TGFB1 (Cook and Vanderhyden 2020). Interestingly, we find that the interaction in the A549 cell line EMT process may be very weak. The detailed results can be found in Appendices B.3 to B.5.

Table 2: Wasserstein distance ($\mathcal{W}_1$) and Total Mass Variation (TMV) of predictions at different time points across five runs on mouse hematopoiesis data ($\sigma = 0.1$). We show the mean value with one standard deviation, where **bold** indicates our algorithm as the best among all algorithms.

| Model | $t = 1$ | | $t = 2$ | |
| --- | --- | --- | --- | --- |
| | $\mathcal{W}_1$ | TMV | $\mathcal{W}_1$ | TMV |
| SF2M (Tong et al. 2024b) | $8.217_{\pm 0.001}$ | $2.231_{\pm 0.000}$ | $11.086_{\pm 0.002}$ | $5.399_{\pm 0.000}$ |
| Meta FM (Atanackovic et al. 2025) | $8.545_{\pm 0.000}$ | $2.231_{\pm 0.000}$ | $10.313_{\pm 0.000}$ | $5.399_{\pm 0.000}$ |
| MMFM (Rohbeck et al. 2025) | $7.647_{\pm 0.000}$ | $2.231_{\pm 0.000}$ | $10.156_{\pm 0.000}$ | $5.399_{\pm 0.000}$ |
| Metric FM (Kapusniak et al. 2024) | $7.788_{\pm 0.000}$ | $2.231_{\pm 0.000}$ | $11.449_{\pm 0.000}$ | $5.399_{\pm 0.000}$ |
| UOT-FM (Eyring et al. 2024) | $8.114_{\pm 0.000}$ | $\mathbf{0.100}_{\pm 0.000}$ | $9.170_{\pm 0.000}$ | $0.118_{\pm 0.000}$ |
| MIOFlow (Huguet et al. 2022) | $6.313_{\pm 0.000}$ | $2.231_{\pm 0.000}$ | $6.746_{\pm 0.000}$ | $5.399_{\pm 0.000}$ |
| uAM (Neklyudov et al. 2023) | $7.537_{\pm 0.000}$ | $2.875_{\pm 0.000}$ | $9.762_{\pm 0.000}$ | $5.670_{\pm 0.000}$ |
| UDSB (Pariset et al. 2023) | $10.687_{\pm 0.058}$ | $0.282_{\pm 0.146}$ | $13.477_{\pm 0.053}$ | $3.010_{\pm 0.225}$ |
| TIGON (Sha et al. 2024) | $6.140_{\pm 0.000}$ | $1.234_{\pm 0.000}$ | $6.973_{\pm 0.000}$ | $2.083_{\pm 0.000}$ |
| DeepRUOT (Zhang et al. 2025a) | $6.052_{\pm 0.002}$ | $0.200_{\pm 0.001}$ | $6.757_{\pm 0.006}$ | $0.260_{\pm 0.007}$ |
| CytoBridge (Ours) | $\mathbf{6.013}_{\pm 0.002}$ | $0.208_{\pm 0.001}$ | $\mathbf{6.644}_{\pm 0.011}$ | $\mathbf{0.078}_{\pm 0.013}$ |

**Extension to Spatiotemporal Transcriptomics**    To demonstrate CytoBridge's applicability in modeling cellular interactions with explicit physical proximity, we applied CytoBridge to a zebrafish spatiotemporal transcriptomics dataset (Liu et al. 2022a), using slices from 5.25 hpf and 10 hpf as input. We evaluated the performance of CytoBridge on the task of reconstructing the dynamics of cell spatial migration, as well as gene expression, with a separate velocity and interactions for physical space and gene expression space respectively. As shown in Table 13, CytoBridge outperforms other methods in both tasks. The results are visualized in Fig. 9. Furthermore, downstream interpretability analysis of the learned interactions identified biologically relevant pathways crucial for zebrafish development. Detailed results can be found in Appendix B.6. The preliminary application of CytoBridge to spatiotemporal transcriptomics demonstrates our framework's potential in modeling spatially resolved data.

# 7    Conclusion

We have introduced CytoBridge for learning unbalanced stochastic mean-field dynamics from time-series snapshot data. To tackle the interacting particle system inference from temporal snapshots, CytoBridge transforms the SDE constraints into ODE leveraging Fisher regularization for more efficient simulation in training. We have demonstrated the effectiveness of our method on both synthetic gene regulatory networks and single-cell RNA-seq data, showing its promising performance. Overall, CytoBridge provides a unified framework for generative modeling of time-series transcriptomics data, enabling more robust and realistic inference of underlying biological dynamics.

**Limitations and Further Directions** While CytoBridge offers valuable insights into incorporating cell-cell interaction with unbalanced stochastic dynamics, several aspects could benefit from further exploration. Firstly, CytoBridge minimizes the upper bound of the energy term in the UMFSB problem. Directly optimizing the original UMFSB formulation remains an important question. Potential solutions may involve leveraging neural SDEs or the integration-by-parts strategy proposed by (Zhang et al. 2025a) as viable approaches. Secondly, the current neural network parameterization of cellular interaction terms is based on techniques used to reduce degrees of freedom, such as RBF expansion. Future work could explore sparse representation methods to replace it for improved expressive power. Thirdly, the training of CytoBridge involves multiple stages. Simplifying the training process by incorporating optimality conditions (e.g., HJB equations) presents a promising research direction (Sun et al. 2025). Furthermore, the current modeling of cellular interactions have not incorporated certain biological priors such as ligand-receptor information. We plan to explore this direction in future work to further refine our approach. Finally, extending the concept of flow matching to this context and developing simulation-free training methods for stochastic, growth/death, and interaction dynamics could also advance the CytoBridge's applicability.

## Acknowledgments and Disclosure of Funding

We thank Dr. Zhenfu Wang (PKU), Prof. Chao Tang (PKU) and Prof. Qing Nie (UCI) for their insightful discussions. This work was supported by the National Key R&D Program of China (No. 2021YFA1003301 to T.L.), National Natural Science Foundation of China (NSFC No. 12288101 to T.L. & P.Z., and 8206100646, T2321001 to P.Z.) and The Fundamental Research Funds for the Central Universities, Peking University. We acknowledge the support from the High-performance Computing Platform of Peking University for computation.

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

# A    Training Details

## A.1    Training CytoBridge

The training of CytoBridge involved training $\mathbf{v}_\theta$, $g_\theta$, $s_\theta$ and $\Phi_\theta$. These networks were initialized after the pre-traing phase, and the overall training process involved minimizing the following loss:

$$\mathcal{L} = \mathcal{L}_{\text{Energy}} + \lambda_r \mathcal{L}_{\text{Recons}} + \lambda_f \mathcal{L}_{\text{FP}}.$$

The calculation of each loss component involved the numerical solution of temporal integrals and ODEs, which was performed using the Neural ODE solver (Chen et al. 2018). The gradients of the loss function with respect to the parameters for $\mathbf{v}_\theta$, $g_\theta$, $s_\theta$ and $\Phi_\theta$ were derived through neural ODE computations, and these networks were optimized using Pytorch (Paszke et al. 2017). To calculate the reconstruction loss during training, we utilized the implementation of Sinkhorn algorithm provided by (Feydy et al. 2019).

## A.2    Training Initial Log Density Function through Score Matching

Conditional Flow Matching (CFM) is used to learn an initial log density. First, sample pairs $(\mathbf{x_0}, \mathbf{x_1})$ are chosen from an optimal transport plan $q(\mathbf{x_0}, \mathbf{x_1})$, and Brownian bridges are constructed between them. We initially assume $\sigma(t)$ to be constant.

The log density is matched with these bridges, where $p(\mathbf{x}, t \mid (\mathbf{x_0}, \mathbf{x_1})) = N(\mathbf{x}; t\mathbf{x_1} + (1 - t)\mathbf{x_0}, \sigma^2 t(1 - t))$ and its gradient is $\nabla_\mathbf{x} \log p(\mathbf{x}, t \mid (\mathbf{x_0}, \mathbf{x_1})) = \frac{t\mathbf{x_1} + (1-t)\mathbf{x_0} - \mathbf{x}}{\sigma^2 t(1-t)}$, for $t \in [0, 1]$. The neural network $s_\theta(\mathbf{x}, t)$ is used to approximate $\frac{1}{2}\sigma^2 \log p(\mathbf{x}, t)$ with a weighting function $\lambda_s$. The unsupervised loss $\mathcal{L}_{\text{us}}$ is:

$$\mathcal{L}_{\text{us}} = \lambda_s^2 \|\nabla_\mathbf{x} s_\theta(\mathbf{x}, t) - \frac{1}{2}\sigma^2 \nabla_\mathbf{x} \log p(\mathbf{x}, t)\|_2^2.$$

The corresponding CFM loss, $\mathcal{L}_{\text{score}}$, is:

$$\mathcal{L}_{\text{score}} = \mathbb{E}_{Q'} \lambda_s^2 \|\nabla_\mathbf{x} s_\theta(\mathbf{x}, t) - \frac{1}{2}\sigma^2 \nabla_\mathbf{x} \log p(\mathbf{x}, t \mid (\mathbf{x_0}, \mathbf{x_1}))\|_2^2,$$

where $Q' = (t \sim \mathcal{U}(0, 1)) \otimes q(\mathbf{x_0}, \mathbf{x_1}) \otimes p(\mathbf{x}, t \mid (\mathbf{x_0}, \mathbf{x_1}))$. By taking the weighting function as:

$$\lambda_s(t) = \frac{2\sqrt{t(1-t)}}{\sigma}.$$

The CFM loss can be converted to:

$$\mathcal{L}_{\text{score}} = \|\lambda_s(t)\nabla_\mathbf{x} s_\theta(\mathbf{x}, t) + \boldsymbol{\epsilon}_1\|_2^2,$$

where $\boldsymbol{\epsilon}_1 \sim \mathcal{N}(0, \mathbf{I})$. This formulation is computationally more tractable.

## A.3    Modeling Interaction Potential

Inspired by the design of machine learning force fields in physics (Schütt et al. 2017; Batzner et al. 2022; Wang et al. 2024), we model the cell interaction network by expanding distances $d_{ij}$ between cells $i$ and $j$ using radial basis functions (RBFs). An exponential transformation is first applied to the raw distance $d_{ij}$. A set of $K$ RBF features, $e_k(d_{ij})$, are then computed based on this exponentially scaled distance:

$$e_k(d_{ij}) = \exp(-\beta_k(\exp(-d_{ij}) - \mu_k)^2)$$

The RBF centers $\mu_k$ are initialized by uniformly discretizing the exponentially transformed distance interval that corresponds to original distances from $0$ to a predefined cutoff $d_{\text{cutoff}}$. The width parameters $\beta_k$ are initialized as the inverse square of a term proportional to the average spread of these centers in the transformed exponential scale. The RBF expansion allows our model to learn interactions between different cells by encoding interaction distances across multiple scales, rather than enforcing interactions only between similar cells.

These expanded features, forming a vector $\mathbf{e}(d_{ij}) = [e_1(d_{ij}), \ldots, e_K(d_{ij})]^T$, are subsequently fed into a multi-layer neural network (NN) with its own set of trainable parameters, $\theta_{NN}$, to predict the interaction potential $\Phi$. This relationship can be expressed as:

$$\Phi(\mathbf{x}_i, \mathbf{x_j}) = \text{NN}(\mathbf{e}(d_{ij}); \theta_{NN})$$

The interaction weight function $k(\mathbf{x}_i, \mathbf{x}_j)$ is defined as:

$$k(\mathbf{x}_i, \mathbf{x}_j) = \begin{cases} 1 & \text{if } d_{ij} < d_{\text{cutoff}} \\ 0 & \text{if } d_{ij} \geq d_{\text{cutoff}} \end{cases}$$

## A.4 Reconstruction Loss Function

The local mass matching loss $\mathcal{L}_{\text{Local Mass}}$ is designed to ensure that the distribution of weights among the sampled particles at each time step aligns with the local density of the real data. Suppose we sample particles at different time points with batch size $N$, which is denoted as $A_i$. At a given time step $t_k$, the local mass matching error $M_k$ is computed for the $N$ sampled particles. This error is defined as:

$$M_k = \sum_{i=1}^{N} \left\| w_i(t_k) - \frac{1}{N} \text{card} \left( h_k^{-1}(\mathbf{x}_i(t_k)) \right) \frac{n_k}{n_0} \right\|_2^2.$$

Here, $w_i(t_k)$ is the weight of the $i$-th sampled particle at time $t_k$, $n_k$ denotes the number of cells in the original dataset at time $t_k$. The term $\text{card} \left( h_k^{-1}(\mathbf{x}_i(t_k)) \right)$ represents the number of sampled data points from $A_k$ that are mapped to the $i$-th sampled particle $\mathbf{x}_i(t_k)$ via the mapping $h_k$. This mapping $h_k$ assigns each real data point in $A_k$ to its closest sampled particle in $\hat{A}_k$. Essentially, the formula measures the squared difference between the particle's current weight and the proportion of real data points it represents. Thus, it provides a fine-grained guidance on the weights of particles. Moreover, under the condition that $M_k = 0$, it follows that $\sum_{i=1}^{N} w_i(t_k) = \frac{1}{N} \sum_{i=1}^{N} \text{card} \left( h_k^{-1}(\mathbf{x}_i(t_k)) \right) \frac{n_k}{n_0} = n_k/n_0$. Therefore the local matching loss also encourages the alignment of the total mass. The local mass matching loss $\mathcal{L}_{\text{Local Mass}}$ is then calculated by summing these errors over all time steps from $k = 1$ to $T - 1$: $\mathcal{L}_{\text{Local Mass}} = \sum_{k=1}^{T-1} M_k$.

Besides, a global mass matching loss is further adopted to align the total number of cells at different time points during the training phase. Specifically, the global mass error term $G_k$ is defined as:

$$G_k = \left| \sum_i w_i(t_k) - \frac{n_k}{n_0} \right|^2,$$

Here, The global mass matching loss $\mathcal{L}_{\text{Global Mass}}$ is then calculated by summing these errors over all time steps from $k = 1$ to $T - 1$: $\mathcal{L}_{\text{Global Mass}} = \sum_{k=1}^{T-1} G_k$.

The optimal transport loss is computed as $\mathcal{L}_{\text{OT}} = \sum_{k=1}^{T-1} \mathcal{W}_2(\hat{\mathbf{w}}^k, \mathbf{w}(t_k))$. Here $\hat{\mathbf{w}}^k = (1/N, 1/N, \ldots, 1/N)$ denotes the uniform distribution of sampled points $A_k$ at time $t_k$ with batch size $N$, $\mathbf{w}(t_k) = (w_1(t_k), w_2(t_k), \ldots, w_N(t_k))/\sum_{i=1}^{N} w_i(t_k)$ denotes the predicted weight distribution of particles at time $t_k$.

## A.5 Energy Loss Function

To simplify the computation, we adopt an upper bound of the energy for training purposes:

$$\mathcal{L}_{\text{energy}} = \mathbb{E}_{\mathbf{x_0} \sim \rho_0} \int_0^T \left[ \frac{1}{2} \|\mathbf{v}_\theta(\mathbf{x}(t), t)\|_2^2 + \frac{1}{2} \|\nabla_\mathbf{x} s_\theta\|_2^2 + \|\mathbf{v}_\theta(\mathbf{x}(t), t)\|_2 \|s_\theta\|_2 + \alpha \Psi(g_\theta) \right] w_\theta(t) \text{d}t,$$

# B Additional Results

## B.1 Synthetic Gene Regulatory Network

**Dynamics** By combining these interactions with the three-gene regulatory network, the system dynamics is defined as follows:

$$\frac{\mathrm{d}X_1^i}{\mathrm{d}t} = \frac{\alpha_1(X_1^i)^2 + \beta}{1 + \gamma_1(X_1^i)^2 + \alpha_2(X_2^i)^2 + \gamma_3(X_3^i)^2 + \beta} - \delta_1 X_1^i + \eta_1\xi_t - \frac{1}{N-1}\sum_{j\neq i,j=1}^{N} k_{i,j}w_j\nabla_1\Phi_{i,j},$$

$$\frac{\mathrm{d}X_2^i}{\mathrm{d}t} = \frac{\alpha_2(X_2^i)^2 + \beta}{1 + \gamma_1(X_1^i)^2 + \alpha_2(X_2^i)^2 + \gamma_3(X_3^i)^2 + \beta} - \delta_2 X_2^i + \eta_2\xi_t - \frac{1}{N-1}\sum_{j\neq i,j=1}^{N} k_{i,j}w_j\nabla_2\Phi_{i,j},$$

$$\frac{\mathrm{d}X_3^i}{\mathrm{d}t} = \frac{\alpha_3(X_3^i)^2}{1 + \alpha_3(X_3^i)^2} - \delta_3 X_3^i + \eta_3\xi_t - \frac{1}{N-1}\sum_{j\neq i,j=1}^{N} k_{i,j}w_j\nabla_3\Phi_{i,j}.$$

Here, $\mathbf{X}^i(t)$ denotes the gene expressions of the $i$ th cell at time $t$, $\Phi_{i,j}$ represents $\Phi(\mathbf{X}^i - \mathbf{X}^j)$, while $\alpha_i, \gamma_i$ and $\beta$ represent the strengths of self-activation, inhibition, and external stimulus respectively. The interaction weight function is defined based on a pre-defined threshold $d_{\text{cutoff}}$, where $k_{i,j}$ is set to 1 if $d_{i,j}$ is within the threshold, otherwise $k_{i,j} = 0$. The parameters $\delta_i$ describe the rates of gene degradation, and $\eta_i\xi_t$ represents stochastic influences via additive white noise. The probability of cell division is associated with $X_2$ expression, given by the formula $g = \alpha_g \frac{X_2^2}{1+X_2^2}$. When a cell divides, new cells are generated with independent random perturbations $\eta_d N(0,1)$ for each gene around the gene expression profile $(X_1(t), X_2(t), X_3(t))$ of the parent cell. Hyper-parameters are detailed in Table 3. The initial cell population is drawn independently from two normal distributions, $\mathcal{N}([2, 0.2, 0], 0.1)$ and $\mathcal{N}([0, 0, 2], 0.1)$. At each time step, any negative expression values are set to 0.

Table 3: Simulation parameters on gene regulatory network.

| Parameter | Value | Description |
|---|---|---|
| $\alpha_1$ | 0.5 | Strength of self-activation for $X_1$ |
| $\gamma_1$ | 0.5 | Strength of inhibition by $X_3$ on $X_1$ |
| $\alpha_2$ | 1 | Strength of self-activation for $X_2$ |
| $\gamma_2$ | 1 | Strength of inhibition by $X_3$ on $X_2$ |
| $\alpha_3$ | 1 | Strength of self-activation for $X_3$ |
| $\gamma_3$ | 10 | Half-saturation constant for inhibition terms |
| $\delta_1$ | 0.4 | Degradation rate for $X_1$ |
| $\delta_2$ | 0.4 | Degradation rate for $X_2$ |
| $\delta_3$ | 0.4 | Degradation rate for $X_3$ |
| $\eta_1$ | 0.05 | Noise intensity for $X_1$ |
| $\eta_2$ | 0.05 | Noise intensity for $X_2$ |
| $\eta_3$ | 0.01 | Noise intensity for $X_3$ |
| $\eta_d$ | 0.014 | Noise intensity for cell perturbations |
| $\beta$ | 1 | External signal activating $X_1$ and $X_2$ |
| $d_{\text{cutoff}}$ | 0.5 | Threshold of interaction weight function |
| $d_e$ | 0.1 | Equilibrium distance of the Lennard-Jones potential |
| $F_{max}$ | 100 | The upper bound of the magnitude of forces |
| $dt$ | 1 | Time step size |
| Time Points | [0, 8, 16, 24, 32] | Time points at which data is recorded |

**Hold-one-out Evaluations**    To evaluate CytoBridge's capability of recovering distributions at unseen time points, we conducted hold-one-out experiments on synthetic gene regulatory data with attractive interactions. Specifically, intermediate time points were individually left out during training and subsequently recovered during evaluation. Note that the default setting of UDSB only supports data with three time points as inputs, we only compare with the performance of other methods. As shown in Table 4, CytoBridge performed best on all held-out time points, indicating its ability to infer reasonable trajectories from snapshots.

**Lennard-Jones Interaction Potential**    The Lennard-Jones-like interaction potential lets cells tend to prevent others from being too similar while attracting cells with distinct gene expressions. The $\Phi$

Table 4: Wasserstein distance ($\mathcal{W}_1$) of predictions at held-out time points across five runs on synthetic gene regulatory data with attractive interactions ($\sigma = 0.05$). We show the mean value with one standard deviation, where **bold** indicates the best among all algorithms.

| | $t = 1$ | $t = 2$ | $t = 3$ |
|---|---|---|---|
| Model | $\mathcal{W}_1$ | $\mathcal{W}_1$ | $\mathcal{W}_1$ |
| SF2M (Tong et al. 2024b) | $0.184_{\pm 0.002}$ | $0.241_{\pm 0.010}$ | $0.507_{\pm 0.007}$ |
| Meta FM (Atanackovic et al. 2025) | $0.272_{\pm 0.000}$ | $0.293_{\pm 0.000}$ | $0.344_{\pm 0.000}$ |
| MMFM (Rohbeck et al. 2025) | $0.476_{\pm 0.000}$ | $0.466_{\pm 0.000}$ | $1.215_{\pm 0.000}$ |
| Metric FM (Kapusniak et al. 2024) | $0.188_{\pm 0.000}$ | $0.498_{\pm 0.000}$ | $0.630_{\pm 0.000}$ |
| UOT-FM (Eyring et al. 2024) | $0.204_{\pm 0.000}$ | $0.155_{\pm 0.000}$ | $0.136_{\pm 0.000}$ |
| MIOFlow (Huguet et al. 2022) | $0.225_{\pm 0.000}$ | $0.270_{\pm 0.000}$ | $0.234_{\pm 0.000}$ |
| uAM (Neklyudov et al. 2023) | $0.600_{\pm 0.000}$ | $0.975_{\pm 0.000}$ | $1.243_{\pm 0.000}$ |
| TIGON (Sha et al. 2024) | $0.254_{\pm 0.000}$ | $0.214_{\pm 0.000}$ | $0.178_{\pm 0.000}$ |
| DeepRUOT (Zhang et al. 2025a) | $0.184_{\pm 0.001}$ | $0.086_{\pm 0.004}$ | $0.079_{\pm 0.003}$ |
| CytoBridge (Ours) | $\mathbf{0.182}_{\pm 0.001}$ | $\mathbf{0.064}_{\pm 0.004}$ | $\mathbf{0.043}_{\pm 0.002}$ |

is defined as:

$$\Phi(\mathbf{x} - \mathbf{y}) = 4\left[\left(\frac{d_e}{\|\mathbf{x} - \mathbf{y}\|}\right)^{12} - \left(\frac{d_e}{\|\mathbf{x} - \mathbf{y}\|}\right)^6\right]$$

where $d_e$ represents the distance at which the repulsive and attractive forces balance each other. Moreover, to avoid the singularity of the potential function, we clipped the magnitude of forces to a pre-defined value $F_{max}$. As shown in Fig. 4, CytoBridge is able to reproduce the Lennard-Jones interaction potential with only information from snapshots provided, leading to correct transition and change of variance. We also conducted quantitative evaluations compared with DeepRUOT, in which the cellular interactions are neglected. As shown in Table 5, CytoBridge consistently outperforms DeepRUOT across all time points, underscoring the importance of explicitly modeling cellular interactions.

Table 5: Wasserstein distance ($\mathcal{W}_1$) of predictions for DeepRUOT and CytoBridge (Ours) on synthetic gene regulatory data with Lennard-Jones Interaction Potential ($\sigma = 0.05$).

| | $t = 1$ | $t = 2$ | $t = 3$ | $t = 4$ |
|---|---|---|---|---|
| Model | $\mathcal{W}_1$ | $\mathcal{W}_1$ | $\mathcal{W}_1$ | $\mathcal{W}_1$ |
| DeepRUOT | $0.036_{\pm 0.000}$ | $0.042_{\pm 0.001}$ | $0.049_{\pm 0.002}$ | $0.054_{\pm 0.002}$ |
| CytoBridge (Ours) | $\mathbf{0.030}_{\pm 0.001}$ | $\mathbf{0.030}_{\pm 0.001}$ | $\mathbf{0.027}_{\pm 0.001}$ | $\mathbf{0.031}_{\pm 0.003}$ |

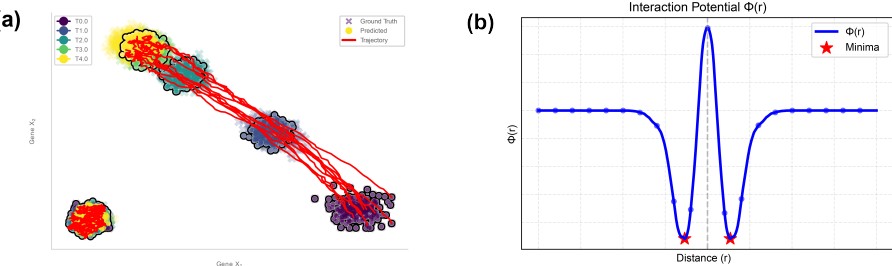

Figure 4: Results of synthetic gene regulatory data with Lennard-Jones Interaction Potential. (a) The dynamics learned by CytoBridge. (b) The learned interaction potential.

**No Interaction** Despite the good performance of CytoBridge on the synthetic gene model with different types of interactions, it yet remains unknown whether our method can help identify the

cases where the ground truth dynamics itself does not involve cellular interactions. Therefore, we further conduct experiments on the synthetic gene model without interactions involved by setting $\Phi(\mathbf{x} - \mathbf{y}) = 0$. As shown in Table 6, explicitly incorporating cellular interactions does not result in the improvement of performance in distribution matching at most time points. The main reason for this phenomenon is that it may be hard for a neural network to directly infer all-zero outputs. The numerical error resulting from our interaction network may lead to perturbations at certain time points and thereby exhibit poorer performance than methods that neglect the interaction term. We further analyze the learned interaction forces acting on each cell. As shown in Fig. 5, compared with forces learned from the dynamics with explicit interactions, forces learned from dynamics with no interaction exhibit notably smaller magnitudes. We compute the correlation of learned velocity and forces of each cell to examine whether the learned forces from data without interaction may exhibit certain patterns related to the transitions of cells. Moran's I, which is a statistic that measures spatial autocorrelation, indicating the degree to which nearby locations have similar values, is utilized to identify whether these patterns exist. Moran's I calculated from data with ground truth interactions is 0.514, while that calculated from data without interactions is 0.281, which is notably smaller. This may indicate that instead of explainable correlated patterns between transition and interaction, interacting forces learned from data without interaction exhibit random patterns to some extent. Biologically, it remains challenging to precisely know whether and how cells interact with each other. Based on this difficulty, we hope that our preceding observations can help explain the performance of our method when applied to real-world biological datasets. Furthermore, we aim for this analysis to provide valuable biological insight into the presence and nature of cell-cell interactions.

Table 6: Wasserstein distance ($\mathcal{W}_1$) of predictions for DeepRUOT and CytoBridge (Ours) on synthetic gene regulatory data **without** interaction potential ($\sigma = 0.05$).

|  | $t = 1$ | $t = 2$ | $t = 3$ | $t = 4$ |
|---|---|---|---|---|
| Model | $\mathcal{W}_1$ | $\mathcal{W}_1$ | $\mathcal{W}_1$ | $\mathcal{W}_1$ |
| DeepRUOT | $\mathbf{0.035}_{\pm 0.001}$ | $0.054_{\pm 0.001}$ | $\mathbf{0.042}_{\pm 0.001}$ | $\mathbf{0.046}_{\pm 0.002}$ |
| CytoBridge (Ours) | $0.036_{\pm 0.001}$ | $\mathbf{0.053}_{\pm 0.001}$ | $0.044_{\pm 0.001}$ | $0.052_{\pm 0.003}$ |

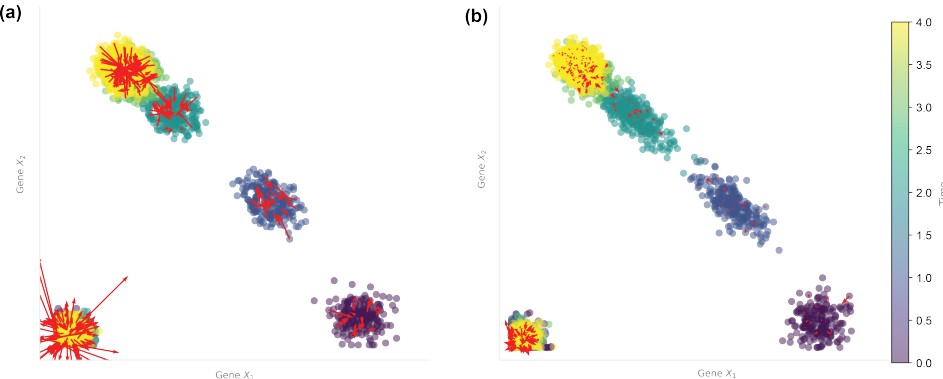

Figure 5: Comparisons of interacting forces learned from (a) dynamics with Lennard-Jones interaction potential, (b) dynamics without interaction potential.

## B.2 Mouse Blood Hematopoiesis

We obtained the dataset from (Weinreb et al. 2020). We used the cells profiled from in vitro cultured mouse hematopoietic progenitors, grown in conditions supporting multi-lineage differentiation. While not transplanted, the system retains extensive intercellular signaling, making it suitable for studying interaction-driven transition dynamics. The original gene expression space was projected to a 50-dimensional subspace using PCA. We summarized the results of hold-one-out evaluations in Table 7. The improvements observed in both Table 2 and Table 7 indicate that incorporating interaction terms indeed helps to recover the trajectories of cells. Moreover, the learned interacting forces exhibit significant correlation with the learned velocity, as quantified by a Moran's I of

0.757. Based on our preceding analysis, this observed pattern and the improvement in performance strongly suggest the presence of genuine interaction forces within this realworld data. To enhance interpretability, we identified top 200 genes influenced most significantly by cell-cell interactions. Subsequent enrichment analysis of these genes revealed key pathways associated with the biological processes, as shown in Table 8. Applying this to the mouse hematopoiesis dataset, we identified pathways related to positive regulation of leukocyte activation and cell-cell adhesion which are closely linked to hematopoiesis, interactions, and differentiation. We also incorporated Trajectory Inference with Cell–Cell Interactions (TICCI) (Fu et al. 2025) as a reference method. By applying TICCI to the mouse hematopoietic dataset, it identified ligand-recptor pair Lgals9-Cd45, which is closely related to the regulation of T cell activation, further provides biological interpretation for enriched pathways. Moreover, we calculated the gradients of growth network with respect to genes to identify key genes that contribute to growth dynamics, including Meis1, Nfkb1, and Rap1b. Meis1 regulates hematopoietic stem cell proliferation and self-renewal, Nfkb1 promotes cell cycle entry and survival as a signaling hub, and Rap1b drives cell division via pathways like MAPK, consistent with established biological knowledge. These findings demonstrate CytoBridge's ability to uncover biologically relevant mechanisms.

Table 7: Wasserstein distance ($\mathcal{W}_1$) of predictions at held-out time point across five runs on mouse hematopoiesis data ($\sigma = 0.1$). We show the mean value with one standard deviation, where **bold** indicates the best among all algorithms.

| Model | $\mathcal{W}_1$ |
|---|---|
| SF2M (Tong et al. 2024b) | $8.646_{\pm 0.001}$ |
| Meta FM (Atanackovic et al. 2025) | $10.821_{\pm 0.000}$ |
| MMFM (Rohbeck et al. 2025) | $8.263_{\pm 0.000}$ |
| Metric FM (Kapusniak et al. 2024) | $7.753_{\pm 0.000}$ |
| UOT-FM (Eyring et al. 2024) | $9.332_{\pm 0.000}$ |
| MIOFlow (Huguet et al. 2022) | $7.779_{\pm 0.000}$ |
| uAM (Neklyudov et al. 2023) | $9.157_{\pm 0.000}$ |
| TIGON (Sha et al. 2024) | $8.402_{\pm 0.000}$ |
| DeepRUOT (Zhang et al. 2025a) | $6.868_{\pm 0.003}$ |
| CytoBridge (Ours) | $\mathbf{6.847}_{\pm 0.003}$ |

Table 8: Enriched pathways from interactions of mouse hematopoiesis data, showing the adjusted p-value and gene count for each term.

| Pathway | p.adjust | Count |
|---|---|---|
| regulation of T cell activation | $3.38 \times 10^{-12}$ | 23 |
| positive regulation of cell activation | $1.42 \times 10^{-11}$ | 23 |
| positive regulation of cell-cell adhesion | $5.81 \times 10^{-11}$ | 20 |
| positive regulation of leukocyte activation | $1.90 \times 10^{-10}$ | 21 |
| leukocyte cell-cell adhesion | $3.18 \times 10^{-10}$ | 21 |
| lymphocyte differentiation | $1.04 \times 10^{-9}$ | 21 |

## B.3 Embryoid Body

We use the same dataset in (Moon et al. 2019; Tong et al. 2020), which consists of 16,819 cells collected at five time points. The cells are from human embryoid bodies (EBs), formed in 3D culture by spontaneous aggregation of stem cells over a 27-day differentiation time course. The experimental setup mimics early developmental conditions, where diverse cell types emerge and are expected to interact through signaling and spatial organization. We projected the original gene expression space to 50 dimensions using PCA. Hold-out evaluations were performed at four distinct time points, and our method consistently yielded the best results in these tests (Table 10). However, the observed performance gain, while positive, was less pronounced compared to the results on the mouse hematopoiesis data. Correspondingly, the learned interacting forces for this dataset exhibit a correlation pattern in the cell-state landscape with a Moran's I of 0.450. This moderate positive

Moran's I suggests the presence of cell-state transition-related interaction effects within this data, though potentially weaker or less organized than those inferred for the lineage-constrained systems such as mouse hematopoiesis data.

Table 9: Wasserstein distance ($\mathcal{W}_1$) and Total Mass Variation (TMV) of predictions at different time points across five runs on embryoid body data ($\sigma = 0.1$). We show the mean value with one standard deviation, where **bold** indicates the best among all algorithms.

| Model | $t = 1$ | | $t = 2$ | | $t = 3$ | | $t = 4$ | |
|---|---|---|---|---|---|---|---|---|
| | $\mathcal{W}_1$ | TMV | $\mathcal{W}_1$ | TMV | $\mathcal{W}_1$ | TMV | $\mathcal{W}_1$ | TMV |
| SF2M (Tong et al. 2024b) | $9.146_{\pm 0.001}$ | $0.748_{\pm 0.000}$ | $10.882_{\pm 0.002}$ | $0.377_{\pm 0.000}$ | $11.650_{\pm 0.004}$ | $0.539_{\pm 0.000}$ | $12.154_{\pm 0.007}$ | $0.399_{\pm 0.000}$ |
| Meta FM (Atanackovic et al. 2025) | $9.497_{\pm 0.000}$ | $0.748_{\pm 0.000}$ | $11.054_{\pm 0.000}$ | $0.377_{\pm 0.000}$ | $11.567_{\pm 0.000}$ | $0.539_{\pm 0.000}$ | $11.487_{\pm 0.000}$ | $0.399_{\pm 0.000}$ |
| MMFM (Rohbeck et al. 2025) | $9.124_{\pm 0.000}$ | $0.748_{\pm 0.000}$ | $10.474_{\pm 0.000}$ | $0.377_{\pm 0.000}$ | $11.022_{\pm 0.000}$ | $0.539_{\pm 0.000}$ | $11.480_{\pm 0.000}$ | $0.399_{\pm 0.000}$ |
| Metric FM (Kapusniak et al. 2024) | $8.506_{\pm 0.000}$ | $0.748_{\pm 0.000}$ | $9.795_{\pm 0.000}$ | $0.377_{\pm 0.000}$ | $10.621_{\pm 0.000}$ | $0.539_{\pm 0.000}$ | $12.042_{\pm 0.000}$ | $0.399_{\pm 0.000}$ |
| UOT-FM (Eyring et al. 2024) | $9.000_{\pm 0.000}$ | $0.054_{\pm 0.000}$ | $10.982_{\pm 0.000}$ | $0.078_{\pm 0.000}$ | $11.584_{\pm 0.000}$ | $0.014_{\pm 0.000}$ | $12.824_{\pm 0.000}$ | $0.092_{\pm 0.000}$ |
| MIOFlow (Huguet et al. 2022) | $8.447_{\pm 0.000}$ | $0.748_{\pm 0.000}$ | $9.229_{\pm 0.000}$ | $0.377_{\pm 0.000}$ | $9.436_{\pm 0.000}$ | $0.539_{\pm 0.000}$ | $10.123_{\pm 0.000}$ | $0.399_{\pm 0.000}$ |
| uAM (Neklyudov et al. 2023) | $12.315_{\pm 0.000}$ | $1.486_{\pm 0.000}$ | $14.996_{\pm 0.000}$ | $1.323_{\pm 0.000}$ | $15.685_{\pm 0.000}$ | $1.526_{\pm 0.000}$ | $18.407_{\pm 0.000}$ | $1.396_{\pm 0.000}$ |
| UDSB (Bunne et al. 2023) | $11.983_{\pm 0.022}$ | $0.429_{\pm 0.008}$ | $14.009_{\pm 0.011}$ | $\mathbf{0.005}_{\pm 0.004}$ | $14.656_{\pm 0.018}$ | $0.166_{\pm 0.006}$ | $15.884_{\pm 0.012}$ | $0.029_{\pm 0.007}$ |
| TIGON (Sha et al. 2024) | $8.433_{\pm 0.000}$ | $0.118_{\pm 0.000}$ | $9.275_{\pm 0.000}$ | $0.030_{\pm 0.000}$ | $9.802_{\pm 0.000}$ | $0.276_{\pm 0.000}$ | $10.148_{\pm 0.000}$ | $0.141_{\pm 0.000}$ |
| DeepRUOT (Zhang et al. 2025a) | $8.159_{\pm 0.002}$ | $0.050_{\pm 0.001}$ | $9.034_{\pm 0.003}$ | $0.161_{\pm 0.002}$ | $9.369_{\pm 0.003}$ | $\mathbf{0.005}_{\pm 0.005}$ | $9.773_{\pm 0.007}$ | $0.262_{\pm 0.007}$ |
| CytoBridge (Ours) | $\mathbf{8.159}_{\pm 0.002}$ | $\mathbf{0.002}_{\pm 0.001}$ | $\mathbf{9.027}_{\pm 0.003}$ | $0.057_{\pm 0.002}$ | $\mathbf{9.351}_{\pm 0.005}$ | $0.175_{\pm 0.007}$ | $\mathbf{9.750}_{\pm 0.006}$ | $\mathbf{0.022}_{\pm 0.006}$ |

Table 10: Wasserstein distance ($\mathcal{W}_1$) of predictions at held-out time points across five runs on embryoid body data ($\sigma = 0.1$). We show the mean value with one standard deviation, where **bold** indicates the best among all algorithms.

| Model | $t = 1$ | $t = 2$ | $t = 3$ |
|---|---|---|---|
| | $\mathcal{W}_1$ | $\mathcal{W}_1$ | $\mathcal{W}_1$ |
| SF2M (Tong et al. 2024b) | $10.302_{\pm 0.001}$ | $11.276_{\pm 0.002}$ | $11.380_{\pm 0.001}$ |
| Meta FM (Atanackovic et al. 2025) | $10.504_{\pm 0.000}$ | $11.478_{\pm 0.000}$ | $11.660_{\pm 0.000}$ |
| MMFM (Rohbeck et al. 2025) | $10.239_{\pm 0.000}$ | $11.469_{\pm 0.000}$ | $11.930_{\pm 0.000}$ |
| Metric FM (Kapusniak et al. 2024) | $9.672_{\pm 0.000}$ | $11.041_{\pm 0.000}$ | $11.466_{\pm 0.000}$ |
| UOT-FM (Eyring et al. 2024) | $10.366_{\pm 0.000}$ | $13.583_{\pm 0.000}$ | $15.858_{\pm 0.000}$ |
| MIOFlow (Huguet et al. 2022) | $10.684_{\pm 0.000}$ | $11.755_{\pm 0.000}$ | $10.440_{\pm 0.000}$ |
| uAM (Neklyudov et al. 2023) | $12.857_{\pm 0.000}$ | $15.743_{\pm 0.000}$ | $17.433_{\pm 0.000}$ |
| TIGON (Sha et al. 2024) | $11.199_{\pm 0.000}$ | $11.207_{\pm 0.000}$ | $10.833_{\pm 0.000}$ |
| DeepRUOT (Zhang et al. 2025a) | $9.628_{\pm 0.001}$ | $10.382_{\pm 0.004}$ | $10.215_{\pm 0.007}$ |
| CytoBridge (Ours) | $\mathbf{9.626}_{\pm 0.001}$ | $\mathbf{10.333}_{\pm 0.004}$ | $\mathbf{10.201}_{\pm 0.007}$ |

## B.4 Pancreatic $\beta$-cell Differentiation Data

We evaluated our method on the dataset from (Veres et al. 2019), which contains 51,274 cells collected across eight time points from human pluripotent stem cells differentiating toward pancreatic $\beta$-like cells in 3D suspension culture. The original gene expression space was projected to 30 dimensions using PCA. As shown in Fig. 7, CytoBridge is able to infer the velocity and growth rate, while identifying attractors. When comparing our approach, which explicitly models cellular interactions, to DeepRUOT, an algorithm that does not, we observed improved performance in 5 out of the 7 tested time points. Furthermore, the learned interacting forces for this dataset exhibited a moderate correlation distribution pattern with inferred state-transition velocity, quantified by a Moran's I of 0.590. Overall, the performance gains across most tested time points, coupled with the observed moderate patterns in the inferred forces, may suggest the presence of interaction forces within this dataset. The correlation between forces and velocities indicates that cells may tend to prevent others from differentiation at early stages while promoting differentiation at later time points.

## B.5 EMT Data

We use the dataset from (Sha et al. 2024; Cook and Vanderhyden 2020), derived from A549 cancer cells undergoing TGFB1-induced epithelial-mesenchymal transition (EMT). This dataset comprises four distinct time points. The cells were cultured in standard 2D monolayers and treated with TGFB1 over several days. Although EMT is a coordinated process in vivo involving paracrine and

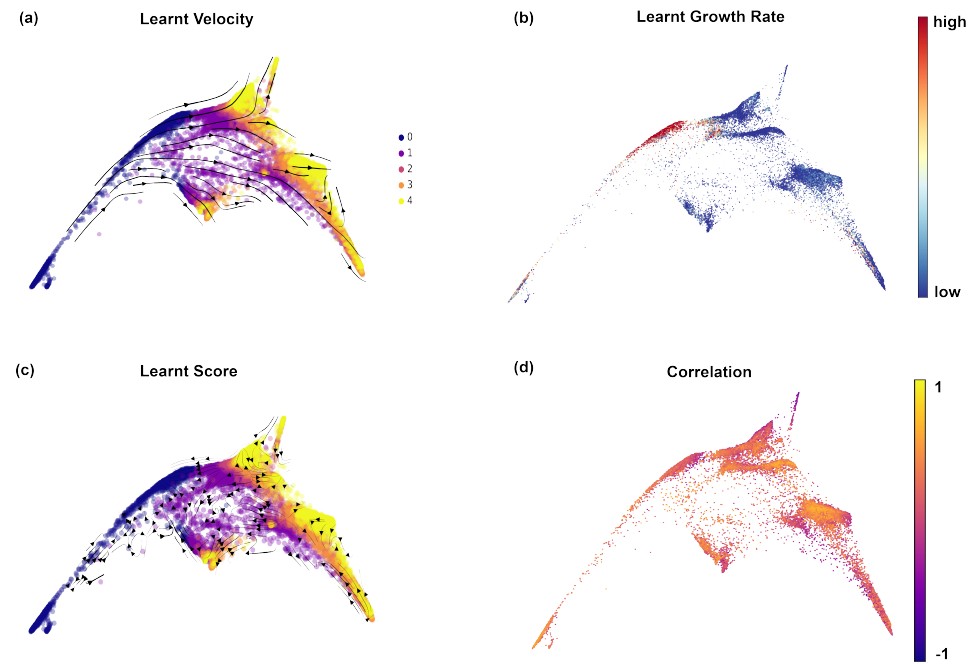

Figure 6: Application in embryoid body data ($\sigma = 0.1$), visualized in PHATE space. (a) The overall velocity learned by CytoBridge. (b) The growth rates learned by CytoBridge. (c) The score function learned by CytoBridge at $t = 4$. (d) The correlation of velocity and interacting forces.

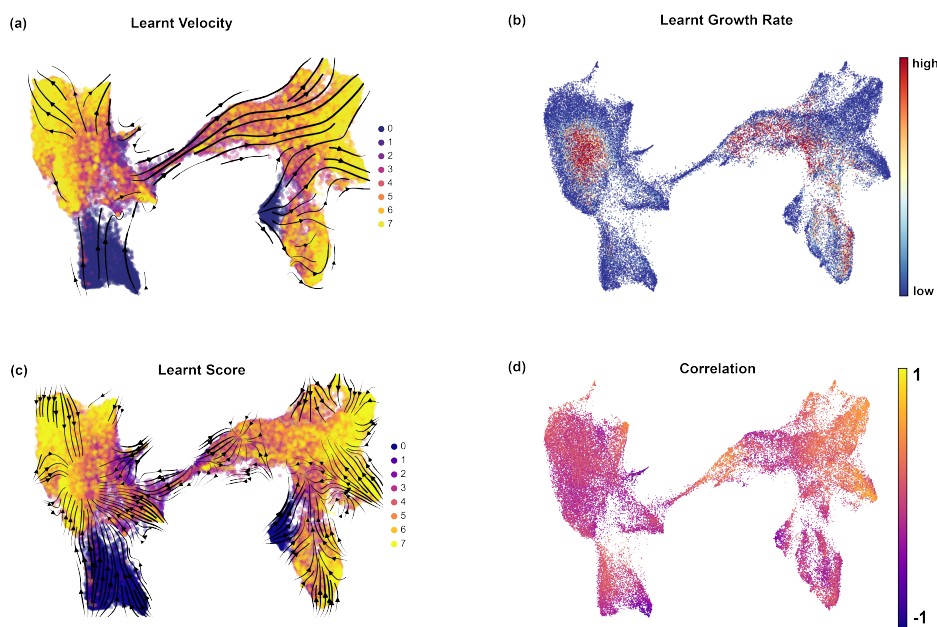

Figure 7: Application in pancreatic $\beta$-cell differentiation data ($\sigma = 0.1$), visualized in UMAP space. (a) The overall velocity learned by CytoBridge. (b) The growth rates learned by CytoBridge. (c) The score function learned by CytoBridge at $t = 7$. (d) The correlation of velocity and interacting forces.

Table 11: Wasserstein distance ($\mathcal{W}_1$) of predictions for DeepRUOT and CytoBridge (Ours) at different time points on pancreatic $\beta$-cell differentiation data ($\sigma = 0.1$).

| Model | $t = 1$ $\mathcal{W}_1$ | $t = 2$ $\mathcal{W}_1$ | $t = 3$ $\mathcal{W}_1$ | $t = 4$ $\mathcal{W}_1$ | $t = 5$ $\mathcal{W}_1$ | $t = 6$ $\mathcal{W}_1$ | $t = 7$ $\mathcal{W}_1$ |
|---|---|---|---|---|---|---|---|
| DeepRUOT | **8.0447**$_{\pm 0.0005}$ | 8.0773$_{\pm 0.0021}$ | 7.6301$_{\pm 0.0032}$ | **8.0064**$_{\pm 0.0042}$ | 7.9018$_{\pm 0.0117}$ | 8.3977$_{\pm 0.0102}$ | 7.8346$_{\pm 0.0109}$ |
| CytoBridge (Ours) | 8.0448$_{\pm 0.0005}$ | **8.0771**$_{\pm 0.0021}$ | **7.6299**$_{\pm 0.0032}$ | 8.0066$_{\pm 0.0043}$ | **7.9018**$_{\pm 0.0117}$ | **8.3974**$_{\pm 0.0102}$ | **7.8343**$_{\pm 0.0109}$ |

contact-dependent signaling, this in vitro system mimics EMT as a largely cell-autonomous response to an external stimulus. Despite the fact that CytoBridge still is able to infer the velocities, the growth of cells, and different cell fates (Fig. 8). When comparing our method, which explicitly models interaction terms, to approaches like DeepRUOT that do not, we observed no improvement in performance on this dataset. Furthermore, the distribution pattern of the learned interacting forces for this dataset was very disorganized, yielding a Moran's I of 0.040. This indicates that the inferred forces are largely random compared to the transition velocity direction, suggesting a lack of significant or organized intercellular interactions for state-transition. The absence of both performance gain and significant correlation in inferred forces suggests that transitions in this setting are primarily driven by direct transcriptional responses rather than intercellular signaling, which is consistent with the biological experiment setup.

Table 12: Wasserstein distance ($\mathcal{W}_1$) of predictions for DeepRUOT and CytoBridge (Ours) at different time points on EMT data ($\sigma = 0.05$).

| Model | $t = 1$ $\mathcal{W}_1$ | $t = 2$ $\mathcal{W}_1$ | $t = 3$ $\mathcal{W}_1$ |
|---|---|---|---|
| DeepRUOT | 0.239 | 0.253 | 0.261 |
| CytoBridge (Ours) | 0.240 | 0.259 | 0.269 |

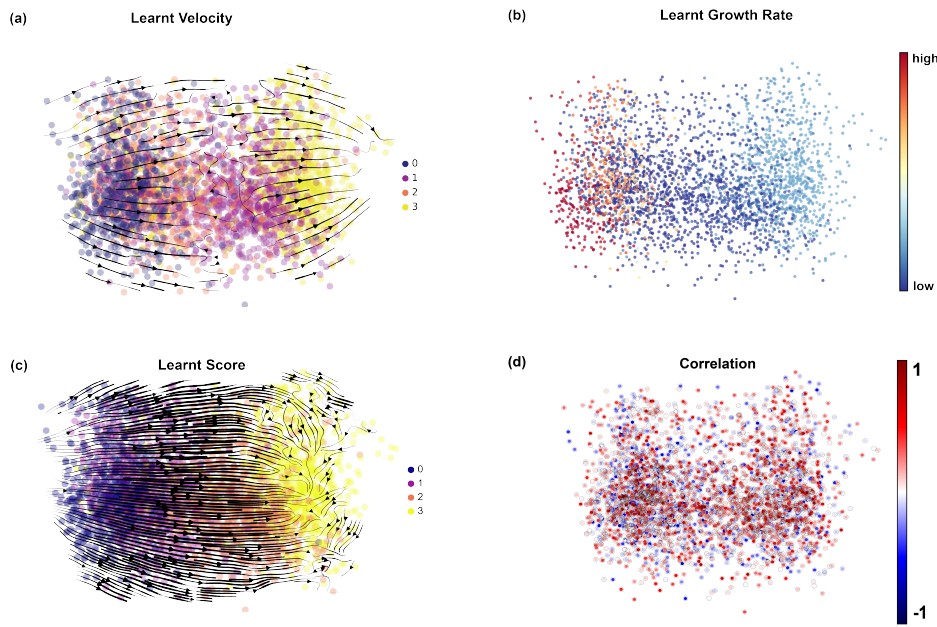

Figure 8: Application in EMT data ($\sigma = 0.05$), visualized in PCA space. (a) The overall velocity learned by CytoBridge. (b) The growth rates learned by CytoBridge. (c) The score function learned by CytoBridge at $t = 4$. (d) The correlation of velocity and interacting forces.

### B.6 Zebrafish Spatiotemporal Data

We adopt the Zebrafish Embryogenesis Spatiotemporal Transcriptomic Atlas (ZESTA) (Liu et al. 2022a), created using the Stereo-seq spatial transcriptomics technology (Chen et al. 2022a). The dataset provides a high-resolution map of gene expression in zebrafish embryos across six critical time points within the first 24 hours of development. Among the six time points, we selected 5.25 hpf and 10 hpf as input. To align spatial coordinates between these two time points, we adopt the rigid body transformation invariant optimal transport. Then, we projected the original gene expression space to 50 dimensions using PCA. To model cellular interactions in both physical space and gene expression space, we transformed both the distances in physical and gene expression space into separate RBF features, and concatenated them as inputs to the interaction potential. Thus, the effects of cellular interactions are achieved by calculating the gradients of interaction potential with respect to distances in physical and gene expression space respectively. The reconstruction loss is calculated in both physical space and gene expression space. We compared the $\mathcal{W}_1$ distances in the physical space and gene expression space between CytoBridge and other methods. As shown in Table 13, CytoBridge achieves better performance over other methods in both physical space and gene expression space. We also visualize the predicted cell states in physical space and gene expression space in Fig. 9. To further interpret the biological effects of learned cellular interactions on the development process of zebrafish, we identified top 200 genes influenced most significantly by interactions. We identified pathways related to somite development which are critical to zebrafish embryonic development based on the enrichment analysis, as shown in Table 14. These findings align with known biological processes, showing the framework's potential in spatially resolved data.

Table 13: Wasserstein distance ($\mathcal{W}_1$) of predictions on zebrafish embryogenesis data. We report metrics in physical space (denoted as 'Space') and gene expression space (denoted as 'Gene'). **Bold** indicates the best result.

| Model | Space | Gene |
|---|---|---|
| SF2M (Tong et al. 2024b) | 0.265 | 5.423 |
| Meta FM (Atanackovic et al. 2025) | 0.268 | 5.413 |
| MMFM (Rohbeck et al. 2025) | 0.247 | 5.208 |
| Metric FM (Kapusniak et al. 2024) | 0.273 | 5.366 |
| UOT-FM (Eyring et al. 2024) | 0.227 | 5.173 |
| MIOflow (Huguet et al. 2022) | 0.263 | 4.720 |
| uAM (Neklyudov et al. 2023) | 0.177 | 6.499 |
| TIGON (Sha et al. 2024) | 0.352 | 4.979 |
| DeepRUOT (Zhang et al. 2025a) | 0.261 | 4.745 |
| CytoBridge (Ours) | **0.035** | **4.712** |

Table 14: Enriched pathways from zebrafish embryogenesis data, showing the adjusted p-value and gene count for each term.

| Pathway | p.adjust | Count |
|---|---|---|
| nucleolus | $5.76 \times 10^{-5}$ | 12 |
| somite development | $1.82 \times 10^{-4}$ | 10 |
| lipid transport | $1.82 \times 10^{-4}$ | 12 |
| gastrulation | $4.15 \times 10^{-4}$ | 12 |
| somitogenesis | $4.15 \times 10^{-4}$ | 8 |

### B.7 Ablation Studies

We conducted ablation studies on the synthetic gene regulatory data with attractive interactions to demonstrate the effectiveness of CytoBridge's different components. We first note that without the interaction term, our method reduces to the framework of DeepRUOT. Compared with DeepRUOT, CytoBridge performs better in all these metrics, underscoring the effectiveness of explicitly considering cell-cell interactions. We then examine the impact of growth term on our algorithm. First, we set the growth term $g$ to zero, and observe that omitting the growth term will lead to poorer performance

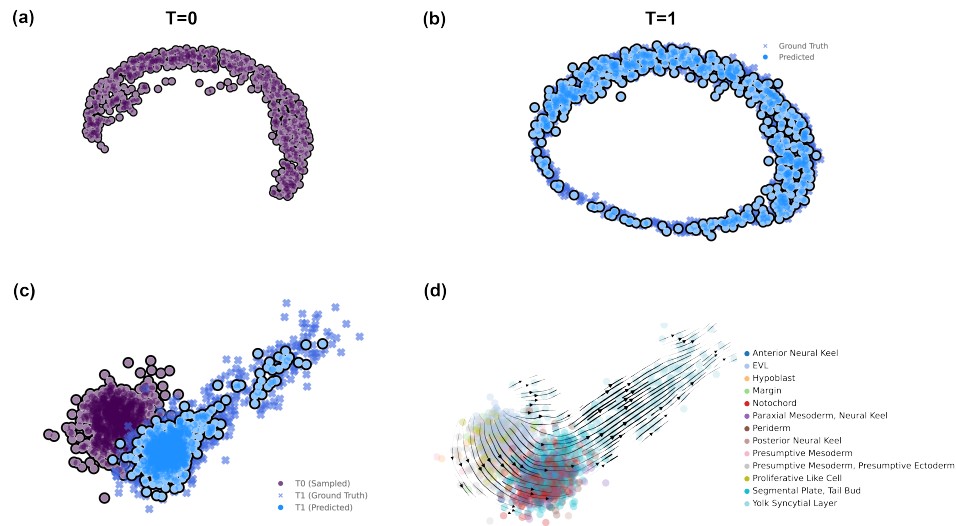

Figure 9: Application in zebrafish embryogenesis data ($\sigma = 0.05$). (a) The spatial coordinates of cells sampled at $T = 0$. (b) The predicted spatial coordinates of cells at $T = 1$ by CytoBridge (c) The predicted gene expression by CytoBridge in the PCA space. (d) The overall velocity learned by CytoBridge in gene expression space.

in distribution matching, which may result from the false transition of balanced Schrödinger Bridge solvers. We then evaluate the impact of $\mathcal{L}_{\text{Mass}}$. By setting the weight of mass loss to zero, we observe that although taking the growth term into account indeed helps eliminate the false transition compared to the results without growth, it still falls short in capturing the changes in total mass, evidenced by the TMV metric. Therefore, it is important to incorporate the $\mathcal{L}_{\text{Mass}}$ term to match the mass changes. Furthermore, we examine the impact of Fokker-Planck Constraint. By setting the weight of $\mathcal{L}_{\text{FP}}$ to zero, we also observed an overall drop in performance. As the Fokker-Planck constraint is necessary to restrain the relationship of our different networks, omitting it will prevent the score function from correctly reflecting the distributions of cells.

Next, we investigate the role of pretraining. Without the pretraining phase, CytoBridge exhibits a significant drop in performance across all time points, with considerably higher distribution matching metrics and less accurate mass matching at later time points. This indicates that pretraining is crucial for initializing the networks effectively and achieving stable overall training. Omitting the score matching phase used to initialize the score function also leads to poorer performance in distribution matching. This suggests that the score matching phase is vital for effectively training the score network. Finally, analyzing the model without the main end-to-end training (relying solely on pretraining, denoted as "w/o training"), we observe performance drop in distribution matching across all time points compared to the full CytoBridge model, and large discrepancies in TMV at later stages. This confirms that while the pretraining phase provides a beneficial initialization, the complete end-to-end training procedure is indispensable for achieving CytoBridge's superior results.

Table 15: Wasserstein distance ($\mathcal{W}_1$) and Total Mass Variation (TMV) of predictions with different settings at different time points across five runs on synthetic gene regulatory data with attractive interactions ($\sigma = 0.05$). We show the mean value with one standard deviation.

| | $t = 1$ | | $t = 2$ | | $t = 3$ | | $t = 4$ | |
| Model | $\mathcal{W}_1$ | TMV | $\mathcal{W}_1$ | TMV | $\mathcal{W}_1$ | TMV | $\mathcal{W}_1$ | TMV |
|---|---|---|---|---|---|---|---|---|
| CytoBridge w/o interaction (DeepRUOT) | $0.044_{\pm 0.002}$ | $0.014_{\pm 0.007}$ | $0.045_{\pm 0.002}$ | $0.026_{\pm 0.018}$ | $0.053_{\pm 0.002}$ | $0.059_{\pm 0.032}$ | $0.057_{\pm 0.003}$ | $0.075_{\pm 0.044}$ |
| CytoBridge w/o growth | $0.068_{\pm 0.001}$ | $0.080_{\pm 0.000}$ | $0.175_{\pm 0.003}$ | $0.250_{\pm 0.000}$ | $0.343_{\pm 0.010}$ | $0.515_{\pm 0.000}$ | $0.480_{\pm 0.018}$ | $0.930_{\pm 0.000}$ |
| CytoBridge w/o $\mathcal{L}_{\text{mass}}$ | $0.016_{\pm 0.001}$ | $0.042_{\pm 0.014}$ | $0.037_{\pm 0.004}$ | $0.165_{\pm 0.030}$ | $0.046_{\pm 0.007}$ | $0.352_{\pm 0.041}$ | $0.047_{\pm 0.005}$ | $0.689_{\pm 0.046}$ |
| CytoBridge w/o $\mathcal{L}_{\text{FP}}$ | $\mathbf{0.014}_{\pm 0.000}$ | $0.013_{\pm 0.012}$ | $0.016_{\pm 0.001}$ | $0.030_{\pm 0.019}$ | $0.022_{\pm 0.003}$ | $0.048_{\pm 0.038}$ | $0.039_{\pm 0.004}$ | $0.063_{\pm 0.058}$ |
| CytoBridge w/o pretraining | $0.426_{\pm 0.003}$ | $0.016_{\pm 0.013}$ | $0.915_{\pm 0.003}$ | $0.035_{\pm 0.016}$ | $1.195_{\pm 0.003}$ | $0.044_{\pm 0.040}$ | $1.451_{\pm 0.004}$ | $0.168_{\pm 0.081}$ |
| CytoBridge w/o score matching | $0.025_{\pm 0.001}$ | $\mathbf{0.011}_{\pm 0.011}$ | $0.025_{\pm 0.002}$ | $0.024_{\pm 0.028}$ | $0.034_{\pm 0.003}$ | $0.049_{\pm 0.039}$ | $0.039_{\pm 0.003}$ | $0.084_{\pm 0.051}$ |
| CytoBridge w/o training | $0.036_{\pm 0.002}$ | $0.012_{\pm 0.006}$ | $0.029_{\pm 0.003}$ | $0.021_{\pm 0.025}$ | $0.067_{\pm 0.003}$ | $0.050_{\pm 0.025}$ | $0.108_{\pm 0.002}$ | $0.174_{\pm 0.076}$ |
| CytoBridge (Ours) | $0.015_{\pm 0.001}$ | $0.013_{\pm 0.009}$ | $\mathbf{0.014}_{\pm 0.001}$ | $\mathbf{0.021}_{\pm 0.024}$ | $\mathbf{0.018}_{\pm 0.002}$ | $\mathbf{0.043}_{\pm 0.041}$ | $\mathbf{0.038}_{\pm 0.003}$ | $\mathbf{0.058}_{\pm 0.061}$ |

## C Experimental Details

### C.1 Evaluation Metrics

We evaluate the 1-Wasserstein distance ($\mathcal{W}_1$), which is used to evaluate the similarity between inferred distributions and true distributions, and Total Mass Variation (TMV), which is used to evaluate whether the inferred dynamics is able to reflect the growth of cells, on the synthetic gene data and real-world single-cell data. The metrics are defined as follows:

$$\mathcal{W}_1(p, q) = \left( \min_{\pi \in \Pi(p,q)} \int \|x - y\|_2 d\pi(x, y) \right),$$

where $p$ and $q$ represent empirical distributions. The $\mathcal{W}_1$ is calculated using the Python Optimal Transport library (POT) (Flamary et al. 2021).

$$\text{TMV}(t_k) = \left| \sum_i w_i(t_k) - \frac{n_k}{n_0} \right|,$$

where $w_i(t_k)$ represents the weight of particle $i$ at time $t_k$, and $n_k$ denotes the number of cells in the original dataset at time $t_k$.

For all datasets, models were trained using all available time points and were evaluated using $\mathcal{W}_1$ and TMV. For synthetic gene regulatory networks with attractive interactions, mouse hematopoiesis, and embryoid body dataset, additional experiments with one-time point held out were conducted. $\mathcal{W}_1$ is used to evaluate the performance at held-out time points. We evaluate our method by applying the learned dynamics to all initial data points to generate trajectory and their weights for subsequent time points starting from initial weights $w_i(0) = 1/n_0$. Next, we compute the weighted $\mathcal{W}_1$ and TMV between the generated data and the real data based on the inferred weights. We ran the simulation five times to calculate the mean value and the standard deviation. To evaluate the performance of DeepRUOT (Zhang et al. 2025a), we deploy the same procedure as ours but without interaction. To evaluate the performance of TIGON (Sha et al. 2024), we reimplemented their method to avoid certain instabilities. As for uAM (Neklyudov et al. 2023), we deploy its default parameter settings. TIGON and uAM are evaluated by simulating the dynamics of weighted particles.

To ensure a fair comparison with other methods, we used their default settings for datasets featured in their original papers. For all other datasets, we adjusted the models' network sizes to ensure comparable parameter counts and tuned their training epochs and learning rates accordingly. To evaluate the performance of SF2M (Tong et al. 2024b), we keep the diffusion coefficient $\sigma$ the same as ours while maintaining other parameters as defaults for fair comparison. The weights of the inferred particles are set to uniform distribution to evaluate as the growth term is not considered. To evaluate the performance of UDSB (Pariset et al. 2023), as its default setting only involves three-time points, we use samples at $t = 0, 2, 4$ from synthetic gene data and embryoid body data as inputs.

### C.2 Hyperparameters Selection and Loss Weighting

The experiments were performed on a shared high-performance computing cluster with NVIDIA A100 GPU and 128 CPU cores. As we aim to make our algorithm universally applicable to different types of biological data, most of the hyperparameters are kept identical across different datasets, while those varying are mainly related to the scale of datasets. Specifically, for the modeling of interaction potential, we set the number of RBF kernels to 8 across different datasets. For real-world scRNA-seq data, the threshold $d_{\text{cutoff}}$ is set no lower than the largest distance between cells in specific datasets so that all pairs of cells are involved in interacting with each other. For the zebrafish data, we set the threshold $d_{\text{cutoff}}$ in the physical space, where cells interact with neighboring cells in physical space. We conducted experiments on the effect of different values of $d_{\text{cutoff}}$ in Table 16.

As shown, we found that the performance was quite robust with respect to different choices of $d_{\text{cutoff}}$ unless the cutoff is set too small, which may potentially lead to inadequate modeling of cellular interactions. To simulate ODEs with random batch methods, we also examined the choices of number of particles $p$ within one group in Table 17. Generally, increasing $p$ slightly improves performance on evaluation metrics by providing a more accurate approximation of the interaction term. However, gains diminish at higher $p$ values, while computational costs increase. Thus, we selected $p = 16$ as the default, balancing robust performance with computational efficiency.

Table 16: Wasserstein distance ($\mathcal{W}_1$) and Total Mass Variation (TMV) of predictions with different $d_{\text{cutoff}}$ values at different time points. The table shows the mean value with one standard deviation.

| $d_{\text{cutoff}}$ | $t = 1$ | | $t = 2$ | | $t = 3$ | | $t = 4$ | |
|---|---|---|---|---|---|---|---|---|
| | $\mathcal{W}_1$ | TMV | $\mathcal{W}_1$ | TMV | $\mathcal{W}_1$ | TMV | $\mathcal{W}_1$ | TMV |
| 0.1 | $0.080_{\pm 0.001}$ | $0.010_{\pm 0.008}$ | $0.119_{\pm 0.003}$ | $0.032_{\pm 0.016}$ | $0.141_{\pm 0.003}$ | $0.050_{\pm 0.042}$ | $0.167_{\pm 0.003}$ | $0.077_{\pm 0.043}$ |
| 0.3 | $0.015_{\pm 0.001}$ | $0.013_{\pm 0.011}$ | $0.016_{\pm 0.001}$ | $0.024_{\pm 0.023}$ | $0.026_{\pm 0.003}$ | $0.038_{\pm 0.044}$ | $0.042_{\pm 0.004}$ | $0.052_{\pm 0.063}$ |
| 0.5 | $0.015_{\pm 0.001}$ | $0.013_{\pm 0.009}$ | $0.014_{\pm 0.001}$ | $0.021_{\pm 0.024}$ | $0.018_{\pm 0.002}$ | $0.043_{\pm 0.041}$ | $0.038_{\pm 0.003}$ | $0.058_{\pm 0.061}$ |
| 0.7 | $0.013_{\pm 0.001}$ | $0.013_{\pm 0.007}$ | $0.013_{\pm 0.001}$ | $0.024_{\pm 0.023}$ | $0.024_{\pm 0.003}$ | $0.047_{\pm 0.031}$ | $0.037_{\pm 0.003}$ | $0.069_{\pm 0.046}$ |
| 1.0 | $0.014_{\pm 0.001}$ | $0.012_{\pm 0.009}$ | $0.014_{\pm 0.001}$ | $0.025_{\pm 0.023}$ | $0.026_{\pm 0.004}$ | $0.039_{\pm 0.041}$ | $0.043_{\pm 0.004}$ | $0.057_{\pm 0.056}$ |
| 2.0 | $0.018_{\pm 0.001}$ | $0.014_{\pm 0.011}$ | $0.025_{\pm 0.002}$ | $0.023_{\pm 0.021}$ | $0.042_{\pm 0.003}$ | $0.046_{\pm 0.031}$ | $0.049_{\pm 0.005}$ | $0.072_{\pm 0.037}$ |

Table 17: Wasserstein distance ($\mathcal{W}_1$) across different time points for varying values of parameter $p$.

| $p$ | $t = 1$ | $t = 2$ | $t = 3$ | $t = 4$ |
|---|---|---|---|---|
| 2 | 0.025 | 0.033 | 0.048 | 0.094 |
| 4 | 0.017 | 0.023 | 0.025 | 0.043 |
| 8 | 0.016 | 0.019 | 0.022 | 0.035 |
| 16 | 0.015 | 0.014 | 0.018 | 0.038 |
| 32 | 0.015 | 0.016 | 0.020 | 0.030 |
| 64 | 0.014 | 0.014 | 0.024 | 0.030 |

Moreover, we also compared the results with and without the random batch method in Table 18. Results show that the full model (without RBM) and the RBM model yield comparable performance, with the full model slightly better at certain time points. However, RBM significantly enhances computational efficiency, reducing the interaction term's complexity from $O(N^2)$ to $O(pN)$. The tables demonstrate substantial reductions in memory and inference time with RBM. Given its comparable accuracy and scalability for large-scale single-cell datasets, RBM is a justified and essential component of our framework.

Table 18: Performance comparison on simulation data with and without RBM.

| | $t = 1$ | $t = 2$ | $t = 3$ | $t = 4$ | Time (s) | Memory (GB) |
|---|---|---|---|---|---|---|
| Method | $\mathcal{W}_1$ | $\mathcal{W}_1$ | $\mathcal{W}_1$ | $\mathcal{W}_1$ | | |
| w/o RBM | 0.015 | 0.013 | 0.023 | 0.029 | 0.568 | 22.1 |
| $p = 16$ | 0.015 | 0.014 | 0.018 | 0.038 | 0.115 | 1.7 |

For our training procedure, the parameters differ only in the number of training epochs. As real-world scRNA-seq data mainly involves large numbers of cells, it typically will require more iterations for our model to converge. Increasing the number of training epochs has few adverse effect, allowing users to use a default of (500, 100, 500) epochs for larger datasets. For other hyper-parameters, we will then provide a guideline on their choices. In the pre-training phase, we first need to provide a suitable initialization for the velocity network and the growth network, which involves selecting the parameters $\lambda_m$ and $\lambda_d$ in $\mathcal{L}_{\text{Recons}} = \lambda_m \mathcal{L}_{\text{Mass}} + \lambda_d \mathcal{L}_{\text{OT}}$. We set $\lambda_d$ to 1 and $\lambda_m$ to 0.01 in order to encourage the transition to match the observed distribution while maintaining the unbalanced effect. We empirically found that lowering the mass loss weighting during pre-training improves distribution matching performance. Here we only adopt the local mass matching loss without restricting the exact number of cells in order to learn general growth patterns. Subsequently, we set $\lambda_m$ to zero to initialize the interaction network. By doing so, the interaction network can be stably trained in order to refine the variance of distributions. The parameter selections of pre-training procedure is summarized in Table 19, where the arrow notation ($\rightarrow$) represents the adjustment of hyperparameters during two stages of our pre-training phase.

During the training phase, as the four networks have been reasonably initialized, these networks can be trained together stably. We set $\lambda_m$ and $\lambda_d$ to 1, while adding the global mass matching term to encourage our growth network to match the exact number of cells at different time points. $\lambda_r$ in $\mathcal{L} = \mathcal{L}_{\text{Energy}} + \lambda_r \mathcal{L}_{\text{Recons}} + \lambda_f \mathcal{L}_{\text{FP}}$ is set to better match the distributions. $\lambda_f$ and $\lambda_w$ in $\mathcal{L}_{\text{FP}}$ are

set to align the score network to match the density. The specific choices of these parameters are summarized in Table 19. We utilize the set of parameters consistently across different datasets.

Table 19: Parameter Settings for Different Datasets Across Two Training Stages (Synthetic gene, mouse hematopoiesis, embryoid body, pancreatic $\beta$-cell differentiation, EMT, Zebrafish).

| Parameter | Synthetic gene | mouse hematopoiesis | embryoid body | pancreatic $\beta$-cell differentiation | EMT | Zebrafish |
|---|---|---|---|---|---|---|
| **Pre-Training Phase** | | | | | | |
| $(\lambda_m, \lambda_d, \text{Epochs})$ | $(0.01, 1, 200)$ $\rightarrow (0.0, 1, 100)$ | $(0.01, 1, 500)$ $\rightarrow (0.0, 1, 100)$ | $(0.01, 1, 500)$ $\rightarrow (0.0, 1, 100)$ | $(0.01, 1, 500)$ $\rightarrow (0.0, 1, 100)$ | $(0.01, 1, 100)$ $\rightarrow (0.0, 1, 100)$ | $(0.01, 1, 500)$ $\rightarrow (0.0, 1, 100)$ |
| **Training Phase** | | | | | | |
| $\lambda_m$ | 1 | 1 | 1 | 1 | 1 | 1 |
| $\lambda_d$ | 1 | 1 | 1 | 1 | 1 | 1 |
| $\lambda_r$ | $1 \times 10^3$ | $1 \times 10^3$ | $1 \times 10^3$ | $1 \times 10^3$ | $1 \times 10^3$ | $1 \times 10^3$ |
| $\lambda_f$ | $1 \times 10^4$ | $1 \times 10^4$ | $1 \times 10^4$ | $1 \times 10^4$ | $1 \times 10^4$ | $1 \times 10^4$ |
| $\lambda_w$ | 10 | 10 | 10 | 10 | 10 | 10 |
| $d_{\text{cutoff}}$ | 0.5 | 300 | 100 | 100 | 2 | 0.3 |
| Epochs | 200 | 1000 | 500 | 500 | 200 | 500 |

## C.3  Scalability and Computational Efficiency

We conducted an evaluation of the scalability and computational efficiency of CytoBridge on the embryoid body data by extending the input from 50 to 100 PCs. As shown in Table 20, CytoBridge achieves the lowest $\mathcal{W}_1$ distance across all time points, demonstrating that CytoBridge remains effective at higher dimensionality. Regarding the computational efficiency, CytoBridge maintains comparable to other neural ODE-based methods with a training time of 11 minutes and a peak GPU memory usage of 6.3 GB.

Table 20: Performance on 100D embryoid body data. We show the Wasserstein distance ($\mathcal{W}_1$) at four time points, alongside total runtime and peak memory usage. **Bold** indicates the best result.

| Model | $t = 1$ $\mathcal{W}_1$ | $t = 2$ $\mathcal{W}_1$ | $t = 3$ $\mathcal{W}_1$ | $t = 4$ $\mathcal{W}_1$ | Time | Memory (GB) |
|---|---|---|---|---|---|---|
| SF2M (Tong et al. 2024b) | 11.333 | 12.982 | 13.718 | 14.945 | 4min 43s | 0.7 |
| Meta FM (Atanackovic et al. 2025) | 11.699 | 13.398 | 14.037 | 14.727 | 5min 50s | 2.2 |
| MMFM (Rohbeck et al. 2025) | 13.150 | 14.135 | 14.441 | 14.907 | 4min 38s | 0.6 |
| Metric FM (Kapusniak et al. 2024) | 10.806 | 12.348 | 13.622 | 16.801 | 2min 16s | 4.2 |
| UOT-FM (Eyring et al. 2024) | 10.757 | 12.799 | 13.761 | 15.657 | 3min 27s | 0.6 |
| uAM (Neklyudov et al. 2023) | 13.628 | 18.315 | 20.309 | 22.973 | 38s | 1.5 |
| MIOflow (Huguet et al. 2022) | 11.387 | 12.331 | 11.905 | 12.908 | 6min 19s | 0.7 |
| TIGON (Sha et al. 2024) | 10.547 | 12.926 | 13.897 | 14.535 | 7min 33s | 0.7 |
| DeepRUOT (Zhang et al. 2025a) | 10.226 | 11.110 | 11.544 | 12.424 | 10min 17s | 3.7 |
| CytoBridge (Ours) | **10.217** | **11.070** | **11.505** | **12.368** | 11min 26s | 6.3 |

## C.4  Visualization

For the Mouse hematopoiesis and pancreatic $\beta$-cell differentiation data, we project them to 2 dimension using UMAP (McInnes et al. 2018) for visualization. For embryoid body data, we project them to 2 dimension using PHATE (Moon et al. 2019). For EMT data, we use the first two PCs for visualization. The learned velocity, score and interaction are visualized using scVelo (Bergen et al. 2020). Here, the velocity stands for the combination of the drift $\mathbf{b}(\mathbf{x}, t)$ of corresponding SDE and interacting forces, which drives the process of the transition of cells. Specifically, scVelo projects high-dimensional vectors into a lower-dimensional space by first computing a transition matrix reflecting cell-to-cell transition probabilities; these probabilities are based on the cosine similarity between the target high-dimensional vector and the displacement vector to its neighbors. Using this matrix and the cells' positions in a chosen embedding, scVelo then estimates the embedded vector for each cell as the expected displacement, effectively summarizing the likely future state of the cell in the low-dimensional representation.

# D  Mathematical Details

## D.1  Proof of Theorem 4.1

*Proof.* From Definition 4.1 we obtain

$$\frac{\partial \rho}{\partial t} = -\nabla_x \cdot \left[ \left( \mathbf{b}(\mathbf{x},t) - \int_{\mathbb{R}^d} k(\mathbf{x},\mathbf{y})\nabla_x \Phi(\mathbf{x}-\mathbf{y})\rho(\mathbf{y},t)\,\mathrm{d}\mathbf{y} \right) \rho(\mathbf{x},t) \right] + \frac{\sigma^2(t)}{2}\Delta\rho(\mathbf{x},t) + g\rho,$$

$$= -\nabla_{\mathbf{x}} \cdot \left( \left( \mathbf{b} - \frac{1}{2}\sigma^2(t)\nabla_{\mathbf{x}}\log\rho(\mathbf{x},t) - \int_{\mathbb{R}^d} k(\mathbf{x},\mathbf{y})\nabla_x \Phi(\mathbf{x}-\mathbf{y})\rho(\mathbf{y},t)\,\mathrm{d}\mathbf{y} \right) \rho(\mathbf{x},t) \right) + g\rho.$$

Using the change of variable $\mathbf{v}(\mathbf{x},t) = \mathbf{b}(\mathbf{x},t) - \frac{1}{2}\sigma^2(t)\nabla_{\mathbf{x}}\log\rho(\mathbf{x},t)$, we see that it is equivalent to

$$\frac{\partial\rho(\mathbf{x},t)}{\partial t} = -\nabla_{\mathbf{x}} \cdot \left[ \left( \mathbf{v}(\mathbf{x},t) - \int_{\mathbb{R}^d} k(\mathbf{x},\mathbf{y})\nabla_{\mathbf{x}}\Phi(\mathbf{x}-\mathbf{y})\rho(\mathbf{y},t)\,\mathrm{d}\mathbf{y} \right) \rho(\mathbf{x},t) \right] + g\rho.$$

Correspondingly, the integrand in the objective functional becomes

$$\inf_{(\rho,\mathbf{b},g,\Phi)} \int_0^T \int_{\mathbb{R}^d} \left[ \frac{1}{2}\|\mathbf{v}\|_2^2 + \frac{\sigma^4(t)}{8}\|\nabla_{\mathbf{x}}\log\rho\|_2^2 + \frac{1}{2}\langle \mathbf{v}, \sigma^2(t)\nabla_{\mathbf{x}}\log\rho\rangle + \alpha\Psi(g) \right] \rho(\mathbf{x},t)\mathrm{d}\mathbf{x}\mathrm{d}t. \tag{5}$$

$\square$

## D.2  Derivation of Proposition 5.1

We assume the following conditions hold.

**Assumption D.1.** *The initial positions $\mathbf{X}_0^i$ are independently and identically distributed (i.i.d.) with a common density $\rho_0(\mathbf{x}) \in L^1(\mathbb{R}^d) \cap L^\infty(\mathbb{R}^d)$, and the initial weights are set as $w_i(0) = 1$ for all $i = 1,\ldots,N$. The functions $\mathbf{b}$, $g$, $\phi$, and $\Phi$ are Lipschitz continuous and bounded: specifically, $\mathbf{b}$ and $g$ are Lipschitz in $\mathbf{x}$ uniformly in $t$, $k(\mathbf{x},\mathbf{y})$ is Lipschitz in both arguments and bounded, and $\nabla_x\Phi$ is Lipschitz continuous and bounded. The diffusion coefficient $\sigma(t)$ is continuous and bounded on $[0,T]$. The empirical measure $\mu_t^N = \frac{1}{N}\sum_{i=1}^N w_i(t)\delta_{\mathbf{X}_t^i}$ converges to a deterministic limit $\rho(\mathbf{x},t)$. The system in Proposition 5.1 satisfies argument as $N \to \infty$, i.e., $\frac{1}{N-1}\sum_{j\neq i} w_j(t)f(\mathbf{X}_t^j,t) \to \int_{\mathbb{R}^d} f(\mathbf{x},t)\rho(\mathbf{x},t)$, where $f$ is an arbitrary test function.*

**Remark D.1.** *The argument we assume here is related to the notions of "chaos" and "propagation of chaos" and it has a rich theory in mathematics (Jabin and Wang 2017; Chaintron and Diez 2021). To rigorously prove such an argument, we refer to some related works (Fournier et al. 2014; Feng and Wang 2024; Duteil 2022; Ben-Porat et al. 2024; Ayi and Duteil 2021).*

We derive the macroscopic continuity equation from the microscopic particle dynamics using the weak formulation, employing test functions and taking the mean-field limit.

Define the weighted empirical measure as

$$\mu_t^N = \frac{1}{N}\sum_{i=1}^N w_i(t)\delta_{\mathbf{X}_t^i}.$$

This measure encapsulates the distribution of particles along with their weights. Our goal is to show that, as $N \to \infty$, $\mu_t^N$ converges weakly to $\rho(\mathbf{x},t)\,\mathrm{d}\mathbf{x}$, where $\rho$ satisfies the stated PDE.

Consider a smooth test function $\varphi : \mathbb{R}^d \to \mathbb{R}$ in $C_c^\infty(\mathbb{R}^d)$ (i.e., infinitely differentiable with compact support). We examine the time evolution of the pairing

$$\int_{\mathbb{R}^d} \varphi(\mathbf{x})\,\mu_t^N(\mathrm{d}\mathbf{x}) = \frac{1}{N}\sum_{i=1}^N w_i(t)\varphi(\mathbf{X}_t^i).$$

To handle the stochasticity, we take the expectation and compute

$$\frac{\mathrm{d}}{\mathrm{d}t}\mathbb{E}\left[ \frac{1}{N}\sum_{i=1}^N w_i(t)\varphi(\mathbf{X}_t^i) \right],$$

and then pass to the limit as $N \to \infty$ to recover the weak form of the PDE.

For each particle $i$, the quantity $w_i(t)\varphi(\mathbf{X}_t^i)$ evolves due to both the deterministic weight dynamics and the stochastic position dynamics. Since $w_i(t)$ follows an ODE and $\mathbf{X}_t^i$ follows an SDE, we apply Itô's product rule:

$$\mathrm{d}\left[w_i(t)\varphi(\mathbf{X}_t^i)\right] = \varphi(\mathbf{X}_t^i)\,\mathrm{d}w_i(t) + w_i(t)\,\mathrm{d}\varphi(\mathbf{X}_t^i) + \mathrm{d}w_i(t) \cdot \mathrm{d}\varphi(\mathbf{X}_t^i).$$

The weight evolution term is given by

$$\mathrm{d}w_i(t) = g(\mathbf{X}_t^i, t)w_i(t)\,\mathrm{d}t,$$

so

$$\varphi(\mathbf{X}_t^i)\,\mathrm{d}w_i(t) = g(\mathbf{X}_t^i, t)w_i(t)\varphi(\mathbf{X}_t^i)\,\mathrm{d}t.$$

For the position evolution, apply Itô's formula to $\varphi(\mathbf{X}_t^i)$:

$$\mathrm{d}\varphi(\mathbf{X}_t^i) = \nabla\varphi(\mathbf{X}_t^i) \cdot \mathrm{d}\mathbf{X}_t^i + \frac{1}{2}\sum_{k=1}^{d}\frac{\partial^2\varphi}{\partial x_k^2}(\mathbf{X}_t^i) \cdot (\sigma(t))^2\,\mathrm{d}t,$$

where the diffusion term arises from the Wiener process (with variance $\sigma^2(t)$). Substituting the SDE

$$\mathrm{d}\mathbf{X}_t^i = \left[\mathbf{b}(\mathbf{X}_t^i, t) - \frac{1}{N-1}\sum_{j\neq i}k_{i,j}w_j(t)\nabla_x\Phi(\mathbf{X}_t^i - \mathbf{X}_t^j)\right]\mathrm{d}t + \sigma(t)\,\mathrm{d}\mathbf{W}_t^i,$$

we obtain

$$\mathrm{d}\varphi(\mathbf{X}_t^i) = \nabla\varphi(\mathbf{X}_t^i) \cdot \left[\mathbf{b} - \frac{1}{N-1}\sum_{j\neq i}k_{i,j}w_j(t)\nabla_x\Phi(\mathbf{X}_t^i - \mathbf{X}_t^j)\right]\mathrm{d}t + \sigma(t)\nabla\varphi(\mathbf{X}_t^i) \cdot \mathrm{d}\mathbf{W}_t^i + \frac{\sigma^2(t)}{2}\Delta\varphi(\mathbf{X}_t^i)\,\mathrm{d}t.$$

Thus,

$$w_i(t)\,\mathrm{d}\varphi(\mathbf{X}_t^i) = w_i(t)\left\{\nabla\varphi(\mathbf{X}_t^i) \cdot \left[\mathbf{b} - \frac{1}{N-1}\sum_{j\neq i}k_{i,j}w_j(t)\nabla_x\Phi(\mathbf{X}_t^i - \mathbf{X}_t^j)\right]\mathrm{d}t + \frac{\sigma^2(t)}{2}\Delta\varphi(\mathbf{X}_t^i)\,\mathrm{d}t\right\}$$
$$+ w_i(t)\sigma(t)\nabla\varphi(\mathbf{X}_t^i) \cdot \mathrm{d}\mathbf{W}_t^i.$$

Since $\mathrm{d}w_i(t) = g(\mathbf{X}_t^i, t)w_i(t)\,\mathrm{d}t$ is deterministic (with no stochastic component), the cross variation $\mathrm{d}w_i(t) \cdot \mathrm{d}\varphi(\mathbf{X}_t^i) = 0$. Combining these, the total differential is

$$\mathrm{d}\left[w_i(t)\varphi(\mathbf{X}_t^i)\right] = g(\mathbf{X}_t^i, t)w_i(t)\varphi(\mathbf{X}_t^i)\mathrm{d}t + w_i(t)\nabla\varphi(\mathbf{X}_t^i) \cdot \left(\mathbf{b} - \frac{1}{N-1}\sum_{j\neq i}k_{i,j}w_j(t)\nabla_x\Phi(\mathbf{X}_t^i - \mathbf{X}_t^j)\right)\mathrm{d}t$$

$$+ w_i(t)\frac{\sigma^2(t)}{2}\Delta\varphi(\mathbf{X}_t^i)\mathrm{d}t + w_i(t)\sigma(t)\nabla\varphi(\mathbf{X}_t^i) \cdot \mathrm{d}\mathbf{W}_t^i.$$

Summing over all $N$ particles and dividing by $N$, we have

$$\frac{\mathrm{d}}{\mathrm{d}t}\left(\frac{1}{N}\sum_{i=1}^{N}w_i(t)\varphi(\mathbf{X}_t^i)\right) = \frac{1}{N}\sum_{i=1}^{N}g(\mathbf{X}_t^i, t)w_i(t)\varphi(\mathbf{X}_t^i)$$

$$+ w_i(t)\nabla\varphi(\mathbf{X}_t^i) \cdot \left(\mathbf{b} - \frac{1}{N-1}\sum_{j\neq i}k_{i,j}w_j(t)\nabla_x\Phi(\mathbf{X}_t^i - \mathbf{X}_t^j)\right)$$

$$+ w_i(t)\frac{\sigma^2(t)}{2}\Delta\varphi(\mathbf{X}_t^i) + \frac{1}{N}\sum_{i=1}^{N}w_i(t)\sigma(t)\nabla\varphi(\mathbf{X}_t^i) \cdot \frac{\mathrm{d}\mathbf{W}_t^i}{\mathrm{d}t}.$$

Taking the expectation, the stochastic term vanishes since $\mathbb{E}[\nabla\varphi(\mathbf{X}_t^i) \cdot \mathrm{d}\mathbf{W}_t^i] = 0$, yielding

$$\mathbb{E}\left[\frac{\mathrm{d}}{\mathrm{d}t}\left(\frac{1}{N}\sum_{i=1}^{N}w_i(t)\varphi(\mathbf{X}_t^i)\right)\right] = \mathbb{E}\left[\frac{1}{N}\sum_{i=1}^{N}\left(g(\mathbf{X}_t^i, t)w_i(t)\varphi(\mathbf{X}_t^i)\right.\right.$$

$$+ w_i(t)\nabla\varphi(\mathbf{X}_t^i) \cdot \left[\mathbf{b} - \frac{1}{N-1}\sum_{j\neq i}k_{i,j}w_j(t)\nabla_x\Phi(\mathbf{X}_t^i - \mathbf{X}_t^j)\right]$$

$$\left.\left.+ w_i(t)\frac{\sigma^2(t)}{2}\Delta\varphi(\mathbf{X}_t^i)\right)\right].$$

As $N \to \infty$, we invoke the assumption. The empirical measure $\mu_t^N$ converges to $\rho(\mathbf{x}, t)\,\mathrm{d}\mathbf{x}$. The interaction sum

$$\frac{1}{N-1} \sum_{j \neq i} k_{i,j} w_j(t) \nabla_x \Phi(\mathbf{X}_t^i - \mathbf{X}_t^j)$$

approximates the mean-field interaction

$$\int_{\mathbb{R}^d} k(\mathbf{X}_t^i, \mathbf{y}) \nabla_x \Phi(\mathbf{X}_t^i - \mathbf{y}) \rho(\mathbf{y}, t)\,\mathrm{d}\mathbf{y}.$$

By the regularity of $\phi$ and $\nabla_x \Phi$, this approximation holds in expectation as $N \to \infty$. Thus, in the limit,

$$\frac{\mathrm{d}}{\mathrm{d}t} \int_{\mathbb{R}^d} \varphi(\mathbf{x}) \rho(\mathbf{x}, t)\,\mathrm{d}\mathbf{x} = \int_{\mathbb{R}^d} \Big[ g\rho\varphi + \rho(\mathbf{x}, t)\nabla\varphi(\mathbf{x}) \cdot \Big( \mathbf{b} - \int_{\mathbb{R}^d} k(\mathbf{x}, \mathbf{y})\nabla_x\Phi(\mathbf{x} - \mathbf{y})\rho(\mathbf{y}, t)\,\mathrm{d}\mathbf{y} \Big)$$
$$+ \frac{\sigma^2(t)}{2} \rho(\mathbf{x}, t)\Delta\varphi(\mathbf{x}) \Big]\mathrm{d}\mathbf{x}.$$

Rewrite the right-hand side using integration by parts, noting that $\varphi$ has compact support (so boundary terms vanish). The drift term becomes

$$\int_{\mathbb{R}^d} \rho\nabla\varphi \cdot \mathbf{b}\,\mathrm{d}\mathbf{x} = - \int_{\mathbb{R}^d} \varphi\nabla \cdot (\mathbf{b}\rho)\,\mathrm{d}\mathbf{x}.$$

The interaction term is

$$- \int_{\mathbb{R}^d} \rho\nabla\varphi \cdot \Big( \int_{\mathbb{R}^d} k(\mathbf{x}, \mathbf{y})\nabla_x\Phi(\mathbf{x} - \mathbf{y})\rho(\mathbf{y}, t)\,\mathrm{d}\mathbf{y} \Big)\mathrm{d}\mathbf{x}$$
$$= \int_{\mathbb{R}^d} \varphi\nabla \cdot \Big[ \rho \int_{\mathbb{R}^d} k(\mathbf{x}, \mathbf{y})\nabla_x\Phi(\mathbf{x} - \mathbf{y})\rho(\mathbf{y}, t)\,\mathrm{d}\mathbf{y} \Big]\mathrm{d}\mathbf{x}.$$

The diffusion term is

$$\int_{\mathbb{R}^d} \rho\Delta\varphi\,\mathrm{d}\mathbf{x} = \int_{\mathbb{R}^d} \varphi\Delta\rho\,\mathrm{d}\mathbf{x},$$

and the growth term is

$$\int_{\mathbb{R}^d} g\rho\varphi\,\mathrm{d}\mathbf{x}.$$

Thus,

$$\frac{\mathrm{d}}{\mathrm{d}t} \int_{\mathbb{R}^d} \rho\varphi\,\mathrm{d}\mathbf{x} = \int_{\mathbb{R}^d} \varphi\Big[ -\nabla \cdot (\mathbf{b}\rho) + \nabla \cdot \Big( \rho \int_{\mathbb{R}^d} k(\mathbf{x}, \mathbf{y})\nabla_x\Phi(\mathbf{x} - \mathbf{y})\rho(\mathbf{y}, t)\,\mathrm{d}\mathbf{y} \Big) + \frac{\sigma^2(t)}{2}\Delta\rho + g\rho \Big]\mathrm{d}\mathbf{x}.$$

Since this equality holds for all $\varphi \in C_c^\infty(\mathbb{R}^d)$, the fundamental lemma of calculus of variations implies

$$\frac{\partial\rho}{\partial t} = -\nabla \cdot \Big[ \mathbf{b}\rho - \rho \int_{\mathbb{R}^d} k(\mathbf{x}, \mathbf{y})\nabla_x\Phi(\mathbf{x} - \mathbf{y})\rho(\mathbf{y}, t)\,\mathrm{d}\mathbf{y} \Big] + \frac{\sigma^2(t)}{2}\Delta\rho + g\rho,$$

which is the desired continuity equation.

# E  More Background on Trajectory Inference

In this section, we provide more background on the trajectory inference task. In single-cell transcriptomics, methods vary depending on the data type:

**Single Time-Point Snapshot Data:** In this context, trajectory inference methods are broadly categorized into two groups. The first type is pseudotime-based methods (Trapnell et al. 2017; Street et al. 2018; Wolf et al. 2019), which infer trajectories by ordering cells along a pseudotime axis but often require specifying a starting point, limiting their flexibility. The second type is RNA velocity-based methods (Bergen et al. 2020; Qiao and Huang 2021; Gayoso et al. 2024), which leverage spliced and

unspliced counts to estimate dynamics but are constrained by the need for such data. Both approaches infer dynamics from existing data without generating new cell states.

**Multi Time-Point Snapshot Data:** Our work addresses a distinct scenario involving single-cell sequencing data across multiple time points. Due to the destructive nature of technology, this can be framed as a generative modeling task, where dynamics are inferred from distributions at different time points and could be interpolated at unseen time points. Once learned, these dynamics enable the generation of intermediate cell states. Methods like optimal transport, which have gained significant attention in computational systems biology and machine learning, are well-suited for this task (Zhang et al. 2025c).

Regarding evaluation and comparability, our generative approach, unlike pseudotime or RNA velocity methods, evaluates performance using metrics for generated distributions, which traditional methods cannot produce. This fundamental difference in data and modeling objectives, along with our hold-out experiments, renders direct comparisons with traditional trajectory inference methods infeasible. Zhang et al. 2025b discusses modeling approaches for different data types. We summarizes these distinctions in Table 21.

Table 21: Comparison of Trajectory Inference Methodologies

| Method | Data | Key | Generative | Evaluation |
|---|---|---|---|---|
| Pseudotime | Single snapshot or merged snapshots without using time labels | Infers trajectories by ordering cells; requires starting point | No | Pseudotime accuracy with true time labels |
| RNA Velocity | Single snapshots with spliced/unspliced counts | Estimates dynamics using RNA splicing kinetics | No | Velocity consistency |
| Optimal Transport | Multi-time-point snapshots | Infers dynamics and generates new cell states | Yes | Generated distribution metrics |

# F  Broader Impacts

Our work presents a new step forward in data-driven modeling of complex, dynamic biological systems by introducing the UMFSB framework and an associated deep learning methodology Cyto-Bridge, capable of explicitly accounting for cell-cell interactions alongside stochastic and unbalanced population effects. The enhanced ability to dissect intercellular communication within evolving cellular landscapes offers the potential for deeper mechanistic insights into fundamental biological processes such as organismal development, disease pathogenesis, and tissue regeneration. Moreover, our approach could enable more accurate predictions of individual therapeutic responses, aid in the design of optimized combination therapies or cell-based treatments. By providing a more accurate representation of these systems, our approach may accelerate the discovery of novel therapeutic targets, helpful for more effective healthcare interventions.

Although the prospective benefits for scientific understanding and biomedical application are considerable, the use of such predictive models also requires careful consideration of potential social impacts and risks. As with any data-driven approach, biases present in training datasets could be amplified by the model, potentially leading to biased outcomes if applied without a rigorous check. Consequently, a thorough validation across diverse conditions and a systematic assessment of potential biases are needed.

