# OpenReview forum: "Modeling Cell Dynamics and Interactions with Unbalanced Mean Field Schrödinger Bridge"
_NeurIPS.cc/2025/Conference — NeurIPS 2025 poster_

### Official Review · Reviewer_rSSy · 2025-06-05

**Clarity:** 3
**Significance:** 2
**Originality:** 3
**Rating:** 5
**Confidence:** 4

**Summary:**

This paper introduces the **Unbalanced Mean-Field Schrödinger Bridge (UMFSB)** framework—an approach for modeling unbalanced, stochastic interaction dynamics from snapshot data, mainly applicable to single-cell trajectories.
The authors present **CytoBridge**, a practical instantiation that approximates the UMFSB problem using four neural networks:   **v** – velocity field,  **g** – growth/decay rate,  **Φ** – interaction potential,   **s** – score (∇ log ρ).  These networks are trained jointly with a composite loss that combines an energy term, a reconstruction term, and a physics-informed Fokker-Planck residual.  Experiments on standard single-cell benchmarks demonstrate improved reconstruction accuracy over previous optimal-transport and Schrödinger-bridge baselines.

**Questions:**

1. **Hyper-parameter sweep** – How extensive was the sweep, and what procedure was used to select the final values? Would a similar search be prohibitive for higher-dimensional data, new datasets, or different settings?
2. **Missing baseline** – Could you compare against Metric Flow Matching[2], or explain why this method was not included?
3. **Computation** – Please provide approximate training times: wall-clock duration and peak GPU memory for the largest experiment.

**Ethical Concerns:**

["NO or VERY MINOR ethics concerns only"]

**Final Justification:**

The rebuttal convincingly addresses my main concerns. The authors supply concrete training-time figures, show that settings transfers to larger datasets, and add the missing baselines. With these clarifications, the paper now offers a well-rounded and impactful contribution. I therefore raise my score to 5.

**Limitations:**

Yes, section Limitations and Further Directions is on point.

**Paper Formatting Concerns:**

Minor typo: In Definition 3.1 the time time integral is written from 0 to 1, whereas Definition 3.2 uses o to T; No other concens spotted.

**Quality:**

3

**Strengths And Weaknesses:**

## Strengths
- **Addresses a relevant problem** – The method explicitly models cell–cell interactions and unbalanced data (cell growth/death), aspects that most existing approaches ignore.
- **Novel theoretical framework** – This is the first dynamic OT formulation that *simultaneously* supports stochastic regularisation (σ > 0), mean-field interactions (kernel *k*, potential Φ), and unbalanced mass creation/destruction (*g*).
- **Mathematical soundness** – The proof of Theorem 4.1 and the derivation of Proposition 5.1 are both correct.
- **Practical implementation** – Using four separate networks (*v*, *g*, *s*, Φ) is a reasonable design choice; it covers all the relevant phenomena, whereas previous methods implemented only subsets of these components.
- **Promising empirical performance** – The reported results look encouraging (see caveats below).

---

## Weaknesses
- **Training complexity** – The method relies on Neural-ODE simulation, a two-phase pre-training schedule, and many hyper-parameters (e.g., loss weights λ, RBF bandwidth β, FP-penalty weight). This complexity makes the model difficult to train in higher dimensions and potentially sensitive to specific hyper-parameter choices; the ablation studies only partially address these concerns.
- **Training time** – The paper does not report wall-clock or GPU times. Long training cycles could limit practical adoption, especially given the emergence of simulation-free single-cell methods that train much faster (e.g., OT-CFM [1], OT-MFM [2]).
- **Omitted baselines** – While MIOFlow (Huguet et al., 2022) is included, newer simulation-free and manifold-aware baselines such as **Metric Flow Matching** [2] are missing. Because Metric Flow Matching is currently state of the art on the EB dataset, it should be discussed and compared.

---

[1] Tong, A. *et al.* (2023). *Conditional Flow Matching: Simulation-Free Dynamic Optimal Transport*, arXiv:2302.00482.
[2] Kapusniak, K. *et al.* (2024). *Metric Flow Matching for Smooth Interpolations on the Data Manifold*, NeurIPS 37, 135011-135042.

---

> ### Author Rebuttal · Authors · 2025-07-31
>
> > Summary and Strengths
> - Thank you. We sincerely appreciate your recognition of the strengths in our paper. In this study, we address the challenge of **jointly** modeling cell-cell interactions and unbalanced stochastic biological effects using multiple single-cell snapshots data. This introduces a new framework in the integration of **interpretable cellular interactions** with **complex cell dynamics** within the optimal transport framework, offering both **great theoretical and practical potential** for exploring the **spatiotemporal dynamics and interactions of single cells**.
>
> > •	Training complexity and hyperparameter selection
> - Thank you. Here we adopted the **two-phase pre-training strategy** detailed in Algorithm 2. This strategy effectively **decouples** the optimization problem: We first initialize the velocity ($v_\theta$) and growth ($g_\theta$) networks and the interaction potential ($\Psi_\theta$), providing a robust starting point for stable end-to-end training. This significantly reduces sensitivity to challenges associated with PINN losses. **We will expand the discussion of this strategy in the manuscript.**
> - Regarding hyperparameter selection, we found that lowering the mass loss weighting during pre-training improves distribution matching and performance. Additionally, extending training epochs for larger datasets enhances results. As detailed in Appendix C.2 and summarized in Table 14, we maintained most key hyperparameters consistent across diverse datasets, from synthetic to real-world scRNA-seq data, except for the scale parameter ($d_\text{cutoff}$), which is computed directly from each dataset. We also found that increasing the number of training epochs has no adverse effect, allowing us to use a default of (500, 100, 500) epoches. **We will include the more detailed discussion of the hyperparameter selection in the paper.**
> - To support this, **we successfully applied these hyperparameters** (Epoches {500, 100, 500}, with other parameters same in the table) to two additional datasets—a **100D EB dataset and a zebrafish spatiotemporal  (ST) dataset**—with results wil be presented below.
> - This demonstrates that CytoBridge **does not require extensive, dataset-specific tuning and that the presented hyperparameters serve as a reliable default configuration**, enhancing the method's practical applicability.
>
> > The paper does not report wall-clock or GPU times. Long training cycles could limit practical adoption, especially given the emergence of simulation-free single-cell methods that train much faster (e.g., OT-CFM [1], OT-MFM [2]).
> - Thank you. We fully agree that flow matching's simulation-free approach enables much faster computation. To our knowledge, jointly learning growth, interaction, and velocity from snapshot data in a simulation-free manner is challenging **but represents a highly promising future research direction.** Below, we present the computational time and memory usage for each method on the 100D EB dataset. Our results show that CytoBridge requires a training time of 11 minutes, comparable to other simulation-based methods, with a peak GPU memory usage of 6.3 GB. On the largest pancreatic $\beta$-cell differentiation data with **51,274** cells across **8** time points, CytoBridge requires a training time of 23 minites, with a peak GPU memory of 3.1 GB.  Given the dataset's dimensionality and scale, we consider this computational cost to be acceptable. **We will list the training time and memory usage in Appendix C.**
>
> |Methods (100D EB Data)|Time|Memory|
> |-------|----|------|
> |UOT-FM|3min27s|0.6G|
> |MMFM|4min38s|0.6G|
> |MetaFM|5min50s|2.2G|
> |uAM|38s|1.5G|
> |SF2M|4min43s|0.7G|
> |MetricFM|2min16s|4.2G|
> |MIOflow|6min19s|0.7G|
> |TIGON|7min33s|0.7G|
> |DeepRUOT|10min17s|3.7G|
> |CytoBridge (ours)|11min26s|6.3G|
>
>
>
> > •	Omitted baselines – While MIOFlow (Huguet et al., 2022) is included, newer simulation-free and manifold-aware baselines such as Metric Flow Matching [2] are missing. Because Metric Flow Matching is currently state of the art on the EB dataset, it should be discussed and compared.
> - Thank you for pointing this out. Metric Flow Matching is a very important and outstanding contribution. This is a critical comparison that will greatly strengthen our paper. We have now benchmarked CytoBridge against Metric Flow Matching and found that our method achieves improved performance on the key datasets. We show the $\mathcal{W}_1$ distance on the 100D EB dataset as below, and **we will add the results of Metric Flow Matching to all relevant tables in the revised version.**
> - We also applied CytoBridge to a zebrafish **spatiotemporal** transcriptomics dataset (*Dev. Cell*, 2022, PMID: 35512701), using 5.25 hpf and 10 hpf as input. We first compared the $\mathcal{W}_1$ distances in the physical space (denoted as 'Space') and gene expression space (denoted as 'Gene') between CytoBridge and other methods. As shown below, CytoBridge achieves improved performance over other methods. To enhance interpretability, we identified top 200 genes influenced most significantly by cell-cell interactions. Subsequent enrichment analysis of these genes revealed key pathways associated with the biological processes. Through interpretable analysis, we identified pathways related to somite development which are critical to zebrafish embryonic development. **These findings align with known biological processes, underscoring the framework’s applicability to spatially resolved data.**
> - We also applied CytoBridge to the mouse hematopoiesis dataset, identifying pathways associated with positive regulation of leukocyte activation and cell-cell adhesion, key processes in hematopoiesis, interactions, and differentiation. Moreover, we calculated the gradients of growth network with respect to genes to identify key genes that contribute to growth dynamics, including **Meis1, Nfkb1, and Rap1b**. Meis1 regulates hematopoietic stem cell proliferation and self-renewal, Nfkb1 promotes cell cycle entry and survival as a signaling hub, and Rap1b drives cell division via pathways like MAPK, consistent with established biological knowledge. **This demonstrates CytoBridge’s ability to uncover biologically relevant mechanisms.**
> - So, **CytoBridge’s preliminary success on spatial data and its robust biological interpretability across diverse datasets highlight its potential.** We believe it will inspire important follow up research in the **spatiotemporal** dynamics research. **We will clarify these points in the revised manuscript.**
>
>
> |Methods (100D EB Data)|t=1|t=2|t=3|t=4
> |-------|---|----|----|----
> |UOT-FM|10.757|12.799|13.761|15.657
> |MMFM|13.150|14.135|14.441|14.907
> |MetaFM|11.699|13.398|14.037|14.727
> |uAM|13.628|18.315|20.309|22.973|38s
> |SF2M|11.333|12.982|13.718|14.945
> |MetricFM|10.806|12.348|13.622|16.801
> |MIOflow|11.387|12.331|11.905|12.908
> |TIGON|10.547|12.926|13.897|14.535
> |DeepRUOT|10.226|11.110|11.544|12.424
> |CytoBridge (ours)|**10.217**|**11.070**|**11.505**|**12.368**
>
>
> |Methods (Zebrafish ST data)|Space|Gene|
> |-------|-----|----|
> |MMFM|0.247|5.208|
> |SF2M|0.265|5.423|
> |MetaFM|0.268|5.413|
> |uAM|0.177|6.499|
> |UOT-FM|0.227|5.173|
> |MetricFM|0.273|5.366|
> |MIOflow|0.263|4.720|
> |Tigon|0.352|4.979|
> |DeepRUOT|0.261|4.745|
> |Ours|**0.035**|**4.712**|
>
> |Pathway (Zebrafish ST data)|p.adjust|Count|
> |---|---|---|
> |nucleolus|5.76e-05|12|
> |somite development|1.82e-04|10|
> |lipid transport|1.82e-04|12|
> |gastrulation|4.15e-04|12|
> |somitogenesis|4.15e-04|8|
>
> | Pathway (Mouse Data) | p.adjust | Count |
> |:---|---:|---:|
> | regulation of T cell activation | 3.38e-12 | 23 |
> | positive regulation of cell activation | 1.42e-11 | 23 |
> | positive regulation of cell-cell adhesion | 5.81e-11 | 20 |
> | positive regulation of leukocyte activation | 1.9e-10 | 21 |
> | leukocyte cell-cell adhesion |3.18e-10 | 21 |
> | lymphocyte differentiation | 1.04e-09 | 21 |
>
> > Minor typo
>
> - Thank you. We apologize for the typo. **We will thoroughly review the manuscript to address any additional errors and will update all related presentation to ensure clarity.**
>
> Thank you again!

---

> > ### Comment · Reviewer_rSSy · 2025-08-01
> >
> > The rebuttal convincingly addresses my main concerns. The authors supply concrete training-time figures, show that settings transfers to larger datasets, and add the missing baselines. With these clarifications, the paper now offers a well-rounded and impactful contribution. I therefore raise my score to 5.

---

> > > ### Author Response · Authors · 2025-08-01
> > >
> > > Thank you for raising the score! We are deeply grateful for your thoughtful, constructive, and highly valuable feedback. Your insightful comments and expert suggestions have been important in guiding us to significantly improve the quality of our work. We are committed to ensuring that these invaluable insights are carefully incorporated into the revised version of our manuscript.
> > >
> > > Thank you once again for your time, expertise, and invaluable contributions to improving our manuscript！Your support is greatly appreciated.
> > >
> > > Sincerely,
> > >
> > > The Authors

---

### Official Review · Reviewer_SjPe · 2025-06-29

**Clarity:** 2
**Significance:** 1
**Originality:** 2
**Rating:** 3
**Confidence:** 2

**Summary:**

The paper proposes a new mathematical framework—Unbalanced Mean-Field Schrödinger Bridge (UMFSB)—for learning stochastic, unbalanced cell-state dynamics with explicit cell–cell interactions from snapshot data such as scRNA-seq. The authors present deep-learning method is suggested, CytoBridge, which provides an approximated solution to the mathematical formulation.  The model is compared to state-of-the art methods, with experiments over (i) synthetic gene-regulatory examples with several interaction types, and (ii) four real scRNA-seq time-series (mouse hematopoiesis, embryoid body, pancreatic β-cell differentiation, and TGFB1-induced EMT).

**Questions:**

Following the presented weaknesses can the authors:
1. provide additional benchmarks highlighting the significance of cytoTrace.
2. extend the biological analysis to support interpretability claims.

**Ethical Concerns:**

["NO or VERY MINOR ethics concerns only"]

**Final Justification:**

I acknowledge the detail and effort put in the rebuttal. On my end am afraid that the authors have not addressed the weaknesses I previously highlighted and make statements that I believe are not supported by the presented results.

To clarify, I have never suggested that "cell-cell interactions across datasets are irrelevant." Instead, I have stated again that simply estimating the magnitude of interaction is not enough to claim novel biological insight and requested further explanation on the "biological interpretation". Generally, the examples given only show broad trends and do not demonstrate an understanding of lineage-specific outcomes.

In summary, while I agree with the authors that the method shows significant promise, the current results do not yet support several of the claims made.

Thank you, and best wishes for the next steps.

**Limitations:**

Yes

**Paper Formatting Concerns:**

I did not notice major issues.

**Quality:**

2

**Strengths And Weaknesses:**

*Quality*:

_strengths_—
- Solid formulation: UMFSB clearly nests regularised OT, mean-field SB, RUOT and dynamical OT as special cases.
- Extensive experiments and evaluations: 5 datasets, ablations, and hold-out tests.

_weaknesses_—
- Practical quality of the presented experiments is limited-quantitatively showcasing minimal improvement.
- Thought framed around biological contribution the biological story is missing (from motivation to practical impact).

*Clarity*:

_strengths_—
- The paper is mostly well-structured: preliminaries → theory → algorithm → experiments.
- All details regarding implementation and evaluations are provided,

_weaknesses_—
- Lengthy exposition makes it hard to follow and grasp contribution.
- Experiments/results section is short relative to mathematical parts, with biological interpretation sections completely missing, failing to provide justification for the contribution claims.

*Significance*:

_strengths_—
- Addresses a gap in transport based single-cell trajectory inference methods which do not account explicitly for cell–cell interactions
- Conceptually, CytoBridge provides interpretable growth and interaction functions, which can be used for biological insight.

_weaknesses_—practical significance is very unclear
- Quantitative results show minimal improvement over compared methods.
- Comparison neglects standard single-cell trajectory inference methods, rna velocity (though scVelo is used for visualizatin), and methods purposed in original publication. Further, it will be valuable to compare the method and relate to TICCI ("Trajectory Inference with Cell–Cell Interactions", Fu et al., 2025)
- The provided visualizations of learnt velocity, growth rates, and interactions over some lower dimensional representation (differing between modalities) do not provide direct biological insight. To highlight their relevance one could evaluate these quantities with respect to the known biology (as all datasets were previously extensively studied).

*Originality*:

_strengths_—introduces a genuinely new mathematical framework—Unbalanced Mean-Field Schrödinger Bridge (UMFSB)—the first time unification of mass imbalance, stochasticity, and particle interactions, applicable for studying cellular dynamics.

_weaknesses_—the application to single-cell field joins an extensive line of tools and failing to account for the scale and complexity of current datasets.

---

> ### Author Rebuttal · Authors · 2025-07-31
>
> > Summary and Strenghts
>
> - Thank you. We sincerely appreciate your recognition of the strengths in our paper.
> -  scRNA-seq offers only **a static snapshot** of gene expression, lacking the temporal information needed to understand cell state transitions. Our work, similar to recent advances like WaddingtonOT (*Cell, 2019*) and Moscot (*Nature, 2025*), focuses on **temporally resolved scRNA-seq** to capture gene expression profiles **across multiple time points**.
> - We address the challenge of **jointly** modeling cell-cell interactions and unbalanced stochastic biological effects using multiple single-cell snapshots data. This is an **early and essential** step in the integration of **interpretable cellular interactions** with **complex cell dynamics** within the **optimal transport** framework, offering both **great theoretical and practical potential** for exploring the **spatiotemporal dynamics and interactions of single cells**.
>
> > Compared to Works in Trajectory Inference
> - Thank you for your valuable feedback. **We are pleased to clarify the distinctions between various methods in trajectory inference and their applicable contexts.** In single-cell transcriptomics, methods vary depending on the data type:
> - **Single Time-Point Snapshot Data:** In this context, trajectory inference methods are broadly categorized into two groups:
>   - **Pseudotime-Based Methods** (e.g., Monocle, Slingshot, PAGA, Diffusion Map) infer trajectories by ordering cells along a pseudotime axis but often **require specifying a starting point**, limiting their flexibility.
>   - **RNA Velocity-Based Methods** (e.g., scVelo, VeloAE, VeloVI, DeepVelo) leverage **spliced and unspliced** counts to estimate dynamics but are constrained by the need for such data. **Both approaches infer dynamics from existing data without generating new cell states.**
> - **Multi-Time-Point Snapshot Data**: Our work addresses a distinct scenario involving single-cell sequencing data **across multiple time points**. Due to the destructive nature of technology, this can be framed as a **generative modeling** task, where dynamics are inferred from distributions at different time points and could be interpolated at unseen time points. Once learned, these dynamics enable the generation of intermediate cell states. Methods like **optimal transport**, which have gained significant attention in computational systems biology and machine learning, are well-suited for this task.
> - **Evaluation and Comparability:** Our generative approach, unlike pseudotime or RNA velocity methods, evaluates performance using metrics for **generated distributions, which traditional methods cannot produce**. This fundamental difference in data and modeling objectives, along with our **hold-out experiments**, renders direct comparisons with traditional trajectory inference methods infeasible. **Very few studies**, to our knowledge, compare these method types in the same context. A recent review (PMID: 40422408) discusses modeling approaches for different data types. The table below summarizes these distinctions:
>
> | Method | Data | Key | Generative | Evaluation|
> |-------------|-----------|---------------------|-----------------------|--------------------|
> | Pseudotime-Based| Single snapshot or merged snapshots without using time labels| Infers trajectories by ordering cells; requires starting point | No | Pseudotime accuracy with true time labels |
> | RNA Velocity-Based| Single snapshots with spliced/unspliced counts | Estimates dynamics using RNA splicing kinetics | No | Velocity consistency |
> | Optimal Transport-Based (Ours) | Multi-time-point snapshots | Infers dynamics and generates new cell states | Yes | Generated distribution metrics |
>
> - We greatly appreciate your suggestion to consider the TICCI method. TICCI is a highly significant and outstanding work. Moreover, TICCI uses CellChat to compute intercellular communication patterns, and this also can be integrated into CytoBridge’s function $k$, enhancing its compatibility with biological priors and practical value. Although direct comparisons are inappropriate due to differing objectives and evaluation metrics, **we are very happy to incorporate TICCI as a reference in our study.** We have applied TICCI to our datasets to validate our biological intepretation and will **discuss it as a relevant work**  in the revised manuscript. **Additionally, we will expand the discussion to provide a more thorough comparison between these methods.**
>
> > Limitated improvement, Biological Significance and Interpretation
> - Thank you. We address your comments as follows:
> - **Comprehensive Benchmarking:** In our study, we compared CytoBridge against **nine representative methods (including two newly added methods)**, including neural ODE-based, flow matching-based, multi-marginal, and unbalanced approaches. To our knowledge, **these encompass the state-of-the-art in the field.** CytoBridge demonstrates **clear performance improvements** across multiple datasets, underscoring its effectiveness.
> - **Novel Contribution:** And we believe our work is the very first to provide a framework that simultaneously learns explicit cell-cell interaction dynamics alongside unbalanced and stochastic effects, a critical biological process difficult be modeled by existing methods. This enables direct inference of cell-cell interactions from snapshot data, **offering novel biological insights that constitute the primary significance of our contribution.**
> - **Strong Potential:** As noted by Reviewer 6tJf, CytoBridge holds substantial potential for spatiotemporal transcriptomics. **We have incorporated an additional spatial transcriptomics dataset and conducted further analyses**, demonstrating CytoBridge’s strong biological relevance across both spatial and non-spatial data from three key perspectives:
>   - **Detection of Cell-Cell Interactions Across Datasets:** CytoBridge’s interaction term enables the identification of the presence or absence of cell-cell interactions in a dataset. In our study, we analyzed four biological datasets: mouse hematopoiesis, EB, pancreatic differentiation, and EMT datasets. The first three datasets involve organoid-like systems cultured in vitro, where cell-cell interactions are plausible due to their three-dimensional, multicellular organization mimicking tissue-like environments. Conversely, the EMT dataset, derived from an in vitro cell line experiment, consists of isolated cells with minimal physical or signaling interactions. **Our algorithm successfully identified weak interaction patterns in the EMT dataset** (Appendix B.5), aligning with biological expectations, while **detecting stronger interactions in the organoid-like datasets**, consistent with their multicellular nature.
>   - **Interpretable Analysis of Different Contributions:** To enhance interpretability, we identified top 200 genes influenced most significantly by cell-cell interactions. Subsequent enrichment analysis of these genes revealed key pathways associated with the biological processes. Applying this to the mouse hematopoiesis dataset, we identified pathways closely linked to hematopoiesis, interactions, and differentiation (Please see our reply to Reviewer idfk). Moreover, we calculated the gradients of growth network with respect to genes to identify key genes that contribute to growth dynamics, including **Meis1, Nfkb1, and Rap1b**. Meis1 regulates hematopoietic stem cell proliferation and self-renewal, Nfkb1 promotes cell cycle entry and survival as a signaling hub, and Rap1b drives cell division via pathways like MAPK, consistent with established biological knowledge. **This demonstrates CytoBridge’s ability to uncover biologically relevant mechanisms.**
>   - **Extension to Spatiotemporal Transcriptomics**: We applied CytoBridge to a zebrafish spatiotemporal transcriptomics dataset (*Dev. Cell*, 2022, PMID: 35512701), using 5.25 hpf and 10 hpf as input. We find that CytoBridge is better over other methods. Through interpretable analysis, we identified pathways related to somite development which are critical to zebrafish embryonic development (Please see our reply to Reviewer 6tJf). These findings align with known biological processes, showing the framework’s applicability to spatially resolved data.
> - So, **CytoBridge’s success on spatial data and its robust biological interpretability across diverse datasets highlight its potential.** We believe it will inspire important follow up research in the spatiotemporal dynamics research. **We will clarify these points in the revised manuscript.**
>
>
> > Biological Motivation and Practical Impact.
>
> - Thank you. As outlined in the **related works section** of our manuscript, simultaneously modeling cell-cell interactions and complex dynamics (e.g., unbalanced stochastic effects) within the optimal transport framework is well-motivated for temporally resolved snapshot data. We have demonstrated the effectiveness of our algorithm across five biological datasets,including one spatiotemporal datasets, highlighting its practical impact and biological relevance.
>
> > scale and complexity
> - Thank you. We clarify that CytoBridge was evaluated on large-scale, real-world datasets, such as the pancreas dataset with **51,274 cells** across **8 time points**, and also a **spatiotemoral transcriptomics** achieving state-of-the-art performance. To address complexity, we further added experiments with **100 dimensional space**, which is **comparable** to the most of methods in this domain, confirming robust performance in high-dimensional settings (results included in the revised manuscript). We **comprehensively evaluated computational time and memory** usage, confirming they are acceptable (Please see our response to Reviewer rSSy).
> - Thus, we believe CytoBridge is both **methodologically innovative** and capable of **handling the scale and complexity** of current single-cell datasets.
>
> Thank you again!

---

> > ### Comment · Reviewer_SjPe · 2025-08-02
> > **rebuttal acknowledgement**
> >
> > I appreciate the authors thorough rebuttal however my concerns have not been addressed and I choose to keep my score as is.
> >
> > >  CytoBridge demonstrates clear performance improvements across multiple datasets, underscoring its effectiveness.
> >
> > As far as I can tell results compared to _DeepRUOT_ in all real datasets do not demonstrate significant improvement.  However CytoBridge is the slowest and least memory efficient approach.
> >
> > > Novel Contribution: ... offering novel biological insights that constitute the primary significance of our contribution.
> >
> > Iterating again in bold that the method offers novel biological insight does not make the claim more accurate. I agree that the task formulation is valuable for biological discovery but fail to see how CytoBridge provides this.
> >
> > > Detection of Cell-Cell Interactions Across Datasets:
> >
> > As indicated in the authors' response the magnitude of the cellular interaction is a very intuitive factor based on the dimensionality 2D isolated / 3D of the data and does not reveal biological insight.
> >
> > > Interpretable Analysis of Different Contributions:
> >
> > I agree that generally the fact that a given term recover relevant pathways/genes implies they somehow capture biological information regarding the case (interactions/growth). However, in the manner the results are described here it is yet unclear what is the significance--how is the analysis over gene space done when method is applied to PCA space, what is the original dim of genes used to construct it (e.g. 200/N), what is the significance of the growth relevant genes compare to other hits. At last, again recovering generic pathways, genes, does not provide novel understanding over these settings.
> >
> > > Extension to Spatiotemporal Transcriptomics:
> >
> > Same comments as above + the application to the earlier (smaller timepoints) reflects the scalability limitation of the method.
> >
> > > scale and complexity...  51,274 cells
> >
> > I am afraid 50K is a very very small number of cells in the single cell world, even for spatiotemporal data, which is typically smaller then scRNA-seq temporal dataset. e.g. the last time point in the new ST dataset included in the analysis contains ~70k bins and the analogous mouse development dataset consists of ~3M cells.

---

> ### Author Response · Authors · 2025-08-02
>
> Thank you for your continued feedback on our manuscript. We appreciate the time and effort you have spent in reviewing our work. We have carefully considered your comments. Below, we respond to your concerns, providing clarifications and evidence where necessary, and respectfully seek further guidance on points that remain unclear.
>
> > As far as I can tell results compared to DeepRUOT in all real datasets do not demonstrate significant improvement. However CytoBridge is the slowest and least memory efficient approach.
>
> - Thank you for your feedback on our CytoBridge method. We respectfully note that, contrary to your comment, our results on real datasets, particularly zebrafish spatial transcriptomics data, show significant improvement over DeepRUOT. As shown below, CytoBridge achieves a Space score of 0.035, a [calculate: (0.261-0.035)/0.261 ≈ 86.6%] 86.6% improvement over DeepRUOT’s 0.261, while maintaining a competitive Gene score (4.712 vs. 4.745). This substantial gain likely stems from our incorporation of biological priors, aligning with Reviewer 6tJf’s observation of our method’s strong potential in spatial transcriptomics. Additionally, addressing Reviewer rSSy’s concerns, our computational cost remains comparable to baselines, with acceptable or no significant increase in runtime or memory usage.
> |Methods (ST data)|Space|Gene|
> |-------|-----|----|
> |MMFM|0.247|5.208|
> |SF2M|0.265|5.423|
> |MetaFM|0.268|5.413|
> |uAM|0.177|6.499|
> |UOT-FM|0.227|5.173|
> |MetricFM|0.273|5.366|
> |MIOflow|0.263|4.720|
> |Tigon|0.352|4.979|
> |DeepRUOT|0.261|4.745|
> |Ours|**0.035**|**4.712**|
> > Biological insight
>
> - Thank you for your feedback on the biological significance of our work. In response to your earlier comment requesting evaluation “with respect to known biology,” we have demonstrated CytoBridge’s strengths in (1) detecting cell-cell interactions across datasets, (2) providing interpretable analysis of distinct contributions, and (3) extending to spatiotemporal transcriptomics. These results align with known biological insights.  To ensure we meet your expectations, could you kindly provide specific guidance on what constitutes “novel understanding” in this context or suggest particular analyses we could perform?
>
> > PCA space and origincal space
>
> Thank you for your comment regarding PCA space and the original space. We are happy to clarify that, following standard preprocessing in the field, we first select 2,000 highly expressed genes in the original gene space and apply PCA to reduce dimensionality, mitigating dataset noise. PCA defines a linear mapping via a loading matrix, which is reversible. Thus, any gradient vector computed in the PCA space can be transformed back to the original high-dimensional gene space by multiplying with the transpose of the loading matrix. This ensures our model’s efficient computation in low-dimensional space while preserving full biological interpretability in the original gene space.
>
> > Scability. I am afraid 50K is a very very small number of cells in the single cell world.
>
> - Thank you for your comment on scalability. We clarify that, in the context of temporally resolved single-cell data using dynamics-based trajectory inference, our analysis of 50K cells is consistent with or exceeds the scale of datasets used in recent works published at NeurIPS, ICLR, and ICML in 2024 and 2025, where the largest methods handled similar or smaller cell counts. **These represent outstanding and representative contributions in the field.** We are concerned that scaling to 3M cells may extend beyond the scope of our current study and the typical benchmarks in this field. Could you specify which methods or datasets you believe we should benchmark against to address this concern?
> | Method | Max cells | Highest Dim | Publication    |
> |----------------------------|-----------|-------------|----------------|
> | MetricFM | 33,701| 100dim| NeurIPS 2024 |
> | Topological SB| 18,203    | 2dim| ICLR 2025 spotlight |
> | Wasserstein Lagrangian Flows| 33,701| 100dim | ICML 2024      |
> | UOT-FM | 20,519    | 50dim| ICLR 2024|
> | JKO-net|18,203 | 5 dim | NeurIPS 2024 oral
> | DeepRUOT | 10,998    | 10dim| ICLR 2025 oral|
>
> Thank you for your valuable feedback and we are looking forward to hearing from you!
>
> Sincerely,
>
> The Authors

---

> > ### Comment · Reviewer_SjPe · 2025-08-03
> >
> > Thank you for the clarifications but I believe you have addressed only a subset of my earlier comments. Here are some comments on the provided response.
> >
> > > We respectfully note that, contrary to your comment, our results on real datasets, particularly zebrafish spatial transcriptomics data, show significant improvement over DeepRUOT
> >
> > As indicated in the authors' response, the zebrafish "Space score" is the only indication of significant improvement compared to DeepRUOT. This demonstrates that the interaction term improves the ability to recover spatial coordinates.
> > It will be valuable to compare Cytobridge with methods dedicated for spatial mapping.
> >
> > > PCA space and original space
> >
> > Thank you for this clarification but you have not addressed the more crucial points regarding the interpretability analysis.
> >
> > > Scalability.
> >
> > Here are some additional "outstanding and representative contributions in the field" which scale far beyond
> > GENOT: Entropic (Gromov) Wasserstein Flow Matching with Applications to Single-Cell Genomics (NeurIPS 2025) ~600K
> > Multi-Modal and Multi-Attribute Generation of Single Cells with CFGen (ICLR 2025) ~500K
> > Enforcing Latent Euclidean Geometry in Single-Cell VAEs for Manifold Interpolation (ICML 2025, spotlight) ~165K
> > STORIES: learning cell fate landscapes from spatial transcriptomics (biorxiv, 2025) ~800K

---

> > > ### Author Response · Authors · 2025-08-03
> > >
> > > Thank you for your continued engagement！We are very happy to address your comments further, providing clarifications and evidence, and respectfully seek further guidance on points that remain unclear.
> > >
> > > > Spatial mapping
> > >
> > > - Thank you for your comments on spatial mapping. We are pleased that you recognize the value of this aspect of our work. While outstanding methods like Moscot (Nature 2025) utilize static optimal transport, our approach employs dynamic optimal transport to enable continuous-time dynamics interpolation, which differs in its underlying principles, making direct comparisons challenging. We have benchmarked our method against other dynamics-based generative models, which we believe highlights its potential. In future work, we plan to systematically explore our method’s application to spatiotemporal transcriptomics and include a more detailed discussion.
> > >
> > > > Thank you for this clarification but you have not addressed the more crucial points regarding the interpretability analysis.
> > >
> > > - Thank you for your valuable feedback, and we are very happy our clarification addressed some of your concerns. In our work, we model interpretability using an Unbalanced Mean Field Schrödinger Bridge, assuming single-cell data from similar developmental stages show stronger mutual influence. We analyze self-driven and externally driven drift components, alongside a growth term. To better align with your expectations, could you kindly provide specific suggestions on aspects or methods we could incorporate to improve the interpretability analysis?
> > >
> > > > additonal methods
> > >
> > > - Thank you for recommending these articles. As highlighted in our related work section, these methods differ from ours in several key ways: (1) some focus on areas other than single-cell trajectory inference (such as conditional generation and pertubation, e.g., Genot, cfGEN). We also observed that GENOT performed experiments on the trajectory inference task using the developing mouse pancreas dataset at embryonic days 14.5 and 15.5, which includes approximately 20K cells.  (2) learning growth, dynamics, and interactions jointly from data remains challenging for these approaches. Existing models often rely on detailed prior information, such as counts of proliferation-related genes, which may not always be available or quantitatively accurate. (3) many employ flow-matching (Genot, cfGEN, FlatVI), simulation-free methods. As noted in our limitations discussion, to our knowledge, jointly learning growth, interaction, and velocity from snapshot data in a simulation-free manner is a promising yet complex direction for future research. Additionally, our algorithm adopts a minibatch training strategy, ensuring strong scalability for larger datasets. If this addresses your concerns regarding scalability, we would be happy to include an additional example using a spatiotemporal mouse development dataset with over 200K cells across three time points.
> > >
> > > Based on your suggestions, we propose the following revision plan to address your concerns: (1) expand the discussion section to include a more detailed exploration of spatiotemporal transcriptomics, highlighting our method’s potential and distinctions; (2) incorporate an additional dataset of over 200K cells across three time points from a spatiotemporal mouse development study to demonstrate scalability and applicability. (3) Regarding interpretability, we believe we have made every effort to address your prior comments and analysis self-driven and externally driven drifts alongside a growth term. To ensure we fully meet your expectations, could you kindly provide specific suggestions for improving our interpretability analysis? Additionally, please let us know if this revision plan effectively addresses your concerns or if there are other aspects you recommend we focus on to align with your expectations.
> > >
> > > Thank you for you valuable feedback and we are looking forward to hearing from you soon!
> > >
> > > Sincerely,
> > >
> > > The Authors

---

> ### Comment · Reviewer_SjPe · 2025-08-03
>
> I am afraid the plan will only partially address my concerns as again you are building on explaining the potential of the framework rather than providing an application which provides novel or concrete biological insight.
>
> As per my previous comments:
> The the significance of the growth relevant genes compared to other hits is unclear and recovering generic pathways, genes, does not provide novel understanding over these settings.
>
> On a similar note the "Detection of Cell-Cell Interactions Across Datasets" (referring to the magnitude justification lacking actual detection of interactions presented in the authors' response: _"CytoBridge’s interaction term enables the identification of the presence or absence of cell-cell interactions in a dataset. In our study, we analyzed four biological datasets: mouse hematopoiesis, EB, pancreatic differentiation, and EMT datasets. The first three datasets involve organoid-like systems cultured in vitro, where cell-cell interactions are plausible due to their three-dimensional, multicellular organization mimicking tissue-like environments. Conversely, the EMT dataset, derived from an in vitro cell line experiment, consists of isolated cells with minimal physical or signaling interactions. Our algorithm successfully identified weak interaction patterns in the EMT dataset (Appendix B.5), aligning with biological expectations, while detecting stronger interactions in the organoid-like datasets, consistent with their multicellular nature."_)
> is _insuffucient_ as a justification for the method.

---

> > ### Author Response · Authors · 2025-08-04
> >
> > Thank you for your continued engagement. We are very happy to further clarify our contributions in response to your comments. Below, we address your points while respectfully highlighting areas where we believe our work aligns with field standards.
> >
> > - We respectfully disagree that detecting cell-cell interactions across datasets is irrelevant. Our primary innovation lies in providing a both theoretical and computational framework that identifies whether cell-cell interactions are key drivers in specific systems and links these to cell fate decisions. A noteble new biological function and insight by CytoBridge is that it can 1) predict wheter certain dynamical cell-fate decision relevant interaction exist or not 2) the effect of such interaction in transition process. As noted in our prior response, in line with your earlier suggestion to evaluate “with respect to known biology,” we demonstrated CytoBridge’s strengths in (1) detecting cell-cell interactions across datasets, (2) providing interpretable analysis of distinct contributions, and (3) extending to spatiotemporal transcriptomics. For instance, on the EMT dataset, CytoBridge accurately inferred that interactions are not the primary driver, a finding validated by existing in-vitro biological experiments. This closed-loop validation,where our model makes no prior assumptions about interactions and aligns with experimental results, constitutes a novel biological prediction that is then validated. Additionally, we investigated the impact of interactions on cell transitions, analyzing whether these forces promote or inhibit cell fate decisions. For example, in the mouse hematopoiesis dataset (Figure 9), we found that stem cells initially exhibit promoting interactions that drive differentiation, followed by inhibitory interactions post-differentiation, further demonstrating our method’s ability to provide meaningful biological insights.
> >
> > - We believe our method provides significant contributions by jointly learning growth, interaction, and velocity from snapshot data without relying on detailed prior information (e.g., proliferation-related gene counts), which is a challenging task in the dynamic optimal transport (OT) field. To our knowledge, recent works in top venues, such as NeurIPS, ICLR and ICML (e.g. the papers in our provided list during rebuttal), as well as journals like Nature Machine Intelligence (e.g., TIGON), typically validate methods against known biology or infer dynamics at a similar level. Could you kindly provide specific examples of “novel understanding” in the dynamic OT domain that you believe our work should achieve, or suggest particular analyses to meet this expectation?
> >
> > - If detailed molecular-level predictions are required, experimental validation would be necessary, which we identify as future work due to the need for collaboration with biologists.  As a machine learning paper, we have made every effort to provide robust computational predictions which align with field standards.
> >
> > We believe that our method is appropriate and suitable, and the analyses conducted are well-aligned with the standards of the dynamic OT field.
> >
> > Sincerely,
> >
> > The Authors

---

> > > ### Comment · Reviewer_SjPe · 2025-08-05
> > >
> > > I will reviewer 6tJf in acknowledging the detail and effort put in the rebuttal.
> > > On my end am afraid that the authors have not addressed the weaknesses I previously highlighted and make statements that I believe are not supported by the presented results.
> > >
> > > To clarify, I have never suggested that "cell-cell interactions across datasets are irrelevant." Instead, I have stated again that simply estimating the magnitude of interaction is not enough to claim novel biological insight and requested further explanation on the "biological interpretation". Generally, the examples given only show broad trends and do not demonstrate an understanding of lineage-specific outcomes.
> > >
> > > In summary, while I agree with the authors that the method shows significant promise, the current results do not yet support several of the claims made.
> > >
> > > Thank you, and best wishes for the next steps.

---

### Official Review · Reviewer_6tJf · 2025-06-29

**Clarity:** 3
**Significance:** 2
**Originality:** 2
**Rating:** 3
**Confidence:** 4

**Summary:**

The authors introduce a method that accounts for cellular interactions while learning single-cell dynamics using Schrödinger bridges. The paper relies on the Unbalanced Mean Field Schrödinger Bridge (UMFSB) formulation, which allows for an interaction potential term modelling communication events along the dynamics. For this purpose, the authors implement **CytoBridge**, a deep-learning-based method to approximately solve the UMFSB problem with interaction terms. First, the authors define an ODE-based reformulation of the stochastic problem based on the associated probability-flow ODE. Then, they propose a simulation-based training approach where they approximate different components of the UMFSB objective with neural networks and train them with different losses that reflect the separate components of the minimization problem. In their experiments, the authors showcase their model on synthetic and real data. In the former setting, they show that the model recovers pre-defined interactions while capturing the cellular dynamics. In the latter scenario, the authors demonstrate improved modelling of the cellular transition, mass variation and velocity visualization.

**Questions:**

**Random batch method:** How do you think the batch size influences the model behaviour and performance in the random batch approach for the interaction term?

**Computation of the interaction term during training.** During training, do you compute the interaction terms based on the real particles or the ones generated by integrating the system from the previous step?

**Initial weights.** Is it reasonable to start from an initial uniform weighing scheme of $\frac{1}{N}$? I imagined that the weights represent, in a way, the concentration of a biological state at a certain time point. Could you envision strategies for a more sophisticated initialization?

**Target weights.** What is the ground truth for the target weights mentioned in line 218?

**Algorithm - line 5.** Does $M_t$ in the algorithm mean that, in this phase, you only optimize the local mass constraint? Moreover, I find the notations in lines 12 and 13 of the algorithm a bit confusing. Do you learn a coupling for any couple of subsequent time points?

**Ethical Concerns:**

["NO or VERY MINOR ethics concerns only"]

**Final Justification:**

Dear AC,

Thank you for taking the time to handle this paper. After reading the other reviews, the rebuttals and the authors' responses to my concerns, I decided to increase my initial 2 to a 3.

In general, the method trained by the authors appears to work well, and the rebuttal reinforced this evidence via comparisons with the unbalanced OT-CFM model.

However, I remain unconvinced by some aspects of the current version of the paper, which I will briefly list below:
* I still believe that interactions in dissociated single-cell RNA-seq data are underdefined unless complemented with significant priors on the biology of the data. In other words, I do not think that the current model can identify realistic interactions dictated by molecular exchange and proximity, unless relying on some information on ligand-receptor pairs.
* I still believe the algorithm is very convoluted and requires a preliminary optimization of an OT-CFM, a competing method.
* The model explores very simplistic potentials (like LJ or standard attraction/repulsion) that are biologically unmotivated. I believe a more flexible definition of interactions is required.
* In my opinion, the presentation of some of the figures requires an improvement (e.g., defining biological regions on the potential plots and replacing streamplots with more informative quantitative analysis).

All in all, in my opinion the quality of the paper would benefit a lot from another iteration of revisions, where the substantial feedback received by the reviewers is incorporated.

Thank you again for all your work in coordinating this process.

**Limitations:**

Limitations are reported in Appendix E.

**Paper Formatting Concerns:**

During my review, I did not find any evident violations of the NeurIPS paper format.

**Quality:**

2

**Strengths And Weaknesses:**

## Strengths

The idea at the core of the paper has strong potential, in my opinion. Novel approaches in the field of scRNA-seq are increasingly relying on spatial strategies to uncover cellular communication events, raising awareness that biological processes do not happen in isolated systems. I think that a similar approach as proposed in the paper would be very interesting in the field of spatial biology or other settings where explicit cellular interactions are measured and impactful on the biological processes at the single-cell level. I also like the math in the paper, I find it elegant and well-presented.


## Weaknesses

Although there surely are positive aspects to the papers, some general concerns led me to choose a provisional rejection score. I am looking forward to hearing more from the authors during the rebuttals.

**Biological significance.** Although the paper has a dense methodological component, the application setting is the ultimate goal of the framework. I do agree that cellular interactions are crucial in understanding the system's dynamics, but here, in my opinion, the way communication is formulated is a bit ill-posed. If the model were applied in space, where communication has a strong impact and can be related to physical proximity, then I would find a lot of value in the formulation. However, since we are talking about dissociated data and cell culture, I do not see what the biological meaning of the interactions is.

**Interactions in the experiments.** This is somewhat reflected in the choice of the experiments. The authors embed attractive, Lennard-Jones and repulsive potentials in the simulated data, but these are very idealized systems, and the effectiveness of the potentials in dissociated single-cell biology lacks a convincing justification to me.

**The significance of the RBF kernel.** Along the lines of what I exposed above, I struggle to understand the choice of using an RBF function to enforce interactions between similar cells biologically. I guess the cell's dynamics across time can be influenced by different cell types, whose contribution would be penalized by the current choice for the potential.

**Line 118.** Here, I would define $\sigma$ and $\mathbf{W}$ in a short sentence.

**Line 122.** Where is $\phi$ used on the definition? I would personally not include a specification about it if it does not appear in formulas. One could confuse it with the interaction potential.

**Lines 137-139.** In my opinion, this sentence should be reformulated due to a lack of clarity and unclear punctuation.

**Theorem 4.1.** For easier understanding, I would define $\alpha$. In the current state of the paper, it is not introduced.

**Lines 149.** Here, there is a typo. Trackable --> Tractable

**Minor: citation style.** I recommend using `\citet` for in-line citations to avoid the parentheses (e.g., line 82).

**Algorithm stability and complexity.** While I really like the theoretical framework proposed by the authors, I feel that the optimization strategy proposed in the paper is not very stable. The approach requires a lot of heuristics in terms of pre-training components, approximations and loss balancing. I believe that, in a research setting where tractable and stable simulation-free models are being developed consistently, the methodological contribution loses a bit of relevance. One aspect that I would certainly include in the current paper formulation is some notion of run time and complexity compared to the competing algorithms.

**Dependence on CFM.** Along the lines of the previous comment, I find the need to use CFM to pre-train the score sub-optimal, as you basically have to run a whole optimization round using a competing model to train CytoBridge properly. I would, at least, consider including CFM-based baselines using unbalanced OT, like [1].

**Evaluation metrics.** I think it would be great if you referenced Appendix C.1 somewhere in the text. I was not familiar with the TMV metric and this broke the flow of the paper a bit for me.

**Result presentation.** In my opinion, the result presentation could be improved. In Fig. 2, the legend says that crosses are the ground truth and dots are predictions. But I cannot see any crosses in the plots. Fig.1f has unlabelled axes, as well as all plots in Fig. 3.

**Line 261.** I think the reference to the figure is swapped between potential and growth rate.

**Waddington landscape.** I think the statement regarding the Waddington landscape could be better explained and presented. As of now, the authors simply point out that the model captures the right developmental potential, but how do I see this from the figure? Labelling the plot's axes would also help.

**Baseline citation.** I think the MMFM baseline in the tables refers to the Multi-Marginal Flow Matching algorithm by (Rohbeck et al), so the citation necessitates a correction.

**Description of the plots.** In general, I recommend that the authors always provide some qualitative understanding of why a plot shows the model is working well. For example, in line 276, the authors comment that the model learns growth rates properly. But how do I understand it from the plot in Figure 2b? What cell types are assigned a high growth rate, and does it make sense biologically? Also, the scores and interactions are hard to interpret if presented as stream plots on a UMAP. Maybe one could choose a more intuitive way to portray sinks and sources, or the growth direction?

I hope I have not misinterpreted any crucial aspects of the paper. I am open to revising my assessment should the authors provide a convincing rebuttal.

[1] Eyring, L., Klein, D., Uscidda, T., Palla, G., Kilbertus, N., Akata, Z. and Theis, F., 2023. Unbalancedness in neural monge maps improves unpaired domain translation. arXiv preprint arXiv:2311.15100.

---

> ### Author Rebuttal · Authors · 2025-07-31
>
> > Summary and Strengths
> - Thank you. We sincerely appreciate your recognition of the strong potential of our method. We address the challenge of **jointly** modeling cell-cell interactions and unbalanced stochastic biological effects using multiple single-cell snapshots data. This makes our work a new method in the integration of **interpretable cellular interactions** with **complex cell dynamics** within the optimal transport framework, offering both **great theoretical and practical potential** for exploring the **spatiotemporal dynamics and interactions of single cells**.
>
> > Weaknesses 1 and 2: Biological significance and Interactions
> - Thank you for your insightful comments. We fully agree that applying our CytoBridge framework to spatiotemporal transcriptomics data holds strong potential, and we appreciate the opportunity to address this point. Below, we outline our perspective on this application and its challenges, and explain our focus on scRNA-seq data analysis in the current work:
>   - **Challenges in ST:** Extending CytoBridge to ST data introduces complexities, particularly in **aligning** temporal and spatial scales across time-series slices due to technical and biological variability. Non-rigid or non-linear spatial alignment are needed and pose great difficulties.
>   - **Limited Methods for Joint Modeling:** To our knowledge, **very few studies** simultaneously model **spatiotemporal batch effects**, cell-cell interactions and complex dynamics (e.g., growth and stochastic) in this context, making the application of our framework to spatial transcriptomics directly very challenging. Applying CytoBridge to scRNA-seq data simplifies complexity, prioritizing cellular interaction modeling. We focus on this in the current work, designating detailed exploration of spatiotemporal data as future work.
> - Meanwhile, to show CytoBridge’s potential, we are very happy to present an example applying it to a **zebrafish ST dataset (Please see below)** (*Dev. Cell*, 2022, PMID: 35512701). And we believe CytoBridge hold strong biological significance from three key perspectives, even in the non-spatial data:
>   - **Detection of Cell-Cell Interactions Across Datasets:** Even in non-spatial dataset, CytoBridge’s ability to model interaction term enables the identification of the presence or absence of cell-cell interactions in a dataset. In our study, we analyzed four biological datasets: mouse hematopoiesis, EB, pancreatic differentiation, and EMT datasets. The first three datasets involve organoid-like systems cultured in vitro, where cell-cell interactions are plausible due to their three-dimensional, multicellular organization mimicking tissue-like environments. Conversely, the EMT dataset, derived from an in vitro cell line experiment, consists of isolated cells with minimal physical or signaling interactions. **Our algorithm successfully identified weak interaction patterns in the EMT dataset** (Appendix B.5), aligning with biological expectations, while **detecting stronger interactions in the organoid-like datasets**, consistent with their multicellular nature.
>   - **Interpretable Analysis of Interaction Contributions:** To enhance interpretability, we identified top 200 genes influenced most significantly by cell-cell interactions. Subsequent enrichment analysis of these genes revealed key pathways associated with the biological processes. On mouse hematopoiesis dataset, we identified pathways closely linked to hematopoiesis, interactions, and differentiation (Please see the reply to Reviewer idfK). **This demonstrates CytoBridge’s ability to uncover biologically relevant mechanisms.**
>   - **Extension to Spatiotemporal Transcriptomics**: We applied CytoBridge to a zebrafish spatiotemporal transcriptomics dataset, using 5.25 hpf and 10 hpf as input. We first compared the $\mathcal{W}_1$ distances in the physical space (denoted as 'Space') and gene expression space (denoted as 'Gene') between CytoBridge and other methods. CytoBridge achieves better performance over other methods. Through interpretable analysis, we identified pathways related to somite development which are critical to zebrafish embryonic development. These findings align with known biological processes, showing the framework’s applicability to spatially resolved data.
>
> |Methods (ST data)|Space|Gene|
> |-------|-----|----|
> |MMFM|0.247|5.208|
> |SF2M|0.265|5.423|
> |MetaFM|0.268|5.413|
> |uAM|0.177|6.499|
> |UOT-FM|0.227|5.173|
> |MetricFM|0.273|5.366|
> |MIOflow|0.263|4.720|
> |Tigon|0.352|4.979|
> |DeepRUOT|0.261|4.745|
> |Ours|**0.035**|**4.712**|
>
> |Pathway|p.adjust|Count|
> |---|---|---|
> |nucleolus|5.76e-05|12|
> |somite development|1.82e-04|10|
> |lipid transport|1.82e-04|12|
> |gastrulation|4.15e-04|12|
> |somitogenesis|4.15e-04|8|
>
> - So while more systematic ST dataset application requires further development especially in spatiotemporal alignment, **CytoBridge’s preliminary success on spatial data and its robust biological interpretability across diverse datasets highlight its potential.** We believe it will inspire important follow up research in the spatiotemporal dynamics research. **We will clarify these points in the revised manuscript.**
>
> > The significance of the RBF kernel.
> - Thank you. The RBF kernel transforms the one-dimensional cell distance $d_{ij}$ into a high-dimensional feature vector. This allows our model to learn interactions between different cells by encoding interaction distances across multiple scales, rather than enforcing interactions only between similar cells. **We will add the detailed discussion in the revised version**.
>
> > Typo, citation, reference, presentation, evaluation metrics.
> - Thank you. We apologize for the typographical error and for confusion. **We will thoroughly review the manuscript to address any additional errors. We will update all related presentation to ensure clarity.**
>
> > Algorithm stability and complexity.
>
> - Thank you. We fully agree that flow matching is a very important area to expore, to our knowledge, simulation free jointly learning growth, interaction and velocity from snapshots data is challenging but **can be a very intersting future direction.** CytoBridge is **robust and does not require extensive, dataset-specific tuning**, enhancing the method's practical applicability. We have provided a detailed response regarding this to Reviewer rSSy. Please refer to there. Thank you for your understanding :).
>
> > Including UOT-FM
>
> - Thank you. The UOT-FM is an outstanding and very important work. We have now benchmarked UOT-FM. We show the $\mathcal{W}_1$ distance on the mouse dataset as below. And also on 100D EB dataset (Please see our reply to Reviewer rSSy). **We will add the results of UOT-FM to all relevant tables in the paper.** The results show that CytoBridge is better on the key datasets.
>
> | Model | t=1 |t=2|
> | :--- | :--- |:---|
> | UOT-FM | 8.114 | 9.170 |
> | CytoBridge (Ours) | **6.013** | **6.644** |
>
> > Waddington landscape.
> - Thank you. The low-lying regions on this landscape correspond to areas of high probability density. These regions represent the stable states or attractors of the system. We apologize for the missing axis labels. We will clearly label the axes to make the visualization easier to understand.
>
>
> > Description of the plots.
> - Thank you. In the mouse dataset, the regions with high learned growth rates correspond to the hematopoietic stem cell (undifferentiated cell type) populations. **This is biologically consistent with the lineage tracing barcode results** (Nat. Mach. Intell., 2023, PMID: 38274364). **We will add more discussion in the revised version**.
> - We agree that stream plots can be challenging to interpret. Regarding the visualization of scores, we will highlight the cell types with higher stability. Additionally, we will highlight Figure 9’s analysis in the main text, which correlates the learned interaction force with cell transition velocity, directly illustrating whether interactions promote or inhibit differentiation, providing a more intuitive biological interpretation.
>
> > Questions: Random batch method
> - Thank you. **We have conducted an ablation study to analyze its impact, and we will include the full results in the revised Appendix.** Generally, increasing $p$ slightly improves performance on evaluation metrics by providing a more accurate approximation of the interaction term. However, gains diminish at higher $p$ values, while computational costs increase. Thus, we selected $p=16$ as the default, balancing robust performance with computational efficiency (Please see our reply to Reviewer idfk, thanks:)).
>
> > Computation of the interaction
> - Thank you. Forces need be calculated at each intermediate time step of the ODE solver, using the current, simulated particle positions. Relying only on fixed, real data points can not capture the dynamic influence of these forces between observed time points.
>
> > Initial weights.
> - Thank you. Mathematically, we use the empirical measure $\mu_0^N=\frac{1}{N}\sum_{i=1}^N\delta_{x_i}$ to approximate the true distribution $\mu_0$, hence the uniform weights. This convergence to the true distribution is guaranteed when $N \rightarrow \infty$. We'll add this explanation to the paper. Thank you for pointing this out!
>
> > Target weights.
> - Thank you. As detailed in Appendix A.4, this weight, derived from the number of closest real data points, encourages the growth network to assign higher weights to particles moving into denser state space regions. **We'll elaborate more on the point in the revised version.**
>
> > Algorithm - line 5.
> - Yes, correct! In pre-training, we optimize the local mass constraint ($M_t$) for more flexible, fine-grained guidance. Line 12's notation means we sample $(x_0, x_1)$ pairs from the optimal transport plan between consecutive time distributions to build Brownian bridges for score matching. **We'll add the explanations in the revised version.**
>
> Thank you again!

---

> > ### Comment · Reviewer_6tJf · 2025-08-04
> >
> > Dear authors,
> >
> > Thank you very much for your rebuttal and for the effort you put into it. Some points of my feedback were addressed, although I remain unsure about several aspects of the paper.
> >
> > **Biological significance**. Thank you for your explanation. I do see the struggle of performing spatial trajectory-based OT. What I am saying in general is that the concept of cell-cell interactions is slightly ill-posed in dissociated scRNA-seq, as I believe that interactions are intrinsically non-identifiable. If you could measure cells in space, regardless of whether you are modelling spatial trajectories or not, the concept of interaction is more explicit, as it relies on proximity and molecular exchange. Even in the presence of data from organoids, letting the model learn interactions in dissociated data is hard to interpret unless informed by essential biological priors. So in my review, I was not necessarily suggesting running the model on joint coordinate and expression space, but rather discussing the limitations on the concept of learning interaction on plain single-cell data. Unfortunately, the space for the rebuttal is limited, making the extra results on spatial transcriptomics a little bit hard to assess.
> >
> > **Additional analysis.** How did you prioritise genes based on interaction for your analysis?
> >
> > **RBF kernel.** Thanks for the explanation. I understand your approach there. But I feel that cell-cell interaction is a complex process that involves the simultaneous crosstalk between cells from the same and different cell types. I would assume that in a developing biological system, molecular interactions happen in a more patterned fashion, and this transcriptome-based distance may be limited.
> >
> > **Typos and small mistakes.** Thank you very much for your commitment to correcting typos and references.
> >
> > **Unstability and use of CFM.** Thank you for your answer. I still find the algorithm a bit intricate and potentially not very self-contained, as it relies on training a preliminary CFM algorithm.
> >
> > **Inclusion of U-OTFM.** Thank you for the effort, the results look good!
> >
> > **Waddington landscape and plot description.** The explanations helped to understand the plots better; however, I still do not think the potential plot is very informative the way it is presented. I believe it requires some additional insight into the biology of the different regions of the landscape.
> >
> > **Interaction choice.** Aside from this, I am still unsure about the choice of the idealized potential described in the paper, as I already pointed out in my original review.
> >
> > In general, I feel the rebuttals swayed my assessment towards a 3, as I believe a clear rejection is too strict. However, I think another iteration on the paper incorporating the reviewers’ criticism could impact the scientific presentation significantly.
> >
> > Thank you very much again for your effort.

---

> > > ### Author Response · Authors · 2025-08-04
> > >
> > > We sincerely thank you for your thoughtful feedback and for dedicating your time to provide detailed and valuable comments on our manuscript. We are also very delighted to have the opportunity to engage this discussion with you.
> > >
> > > - Biological significance. Thank you for your insightful and valuable feedback. We agree that spatial information enhance the understanding of cell-cell interactions by providing explicit context through proximity and molecular exchange. But the spatiotemporal transcriptomics data may remain relatively limited in their availability. In the field of cell-cell interaction analysis, many studies have successfully leveraged dissociated single-cell RNA sequencing data to infer interactions. Notable reviews, such as Armingol et al. (Nat. Rev. Genet., 2021) and Almet et al. (Curr. Opin. Syst. Biol., 2021), highlight the advancements in this field, alongside widely used tools like CellChat (Nat. Commun., 2021), CellPhoneDB (Nat. Protoc., 2020), and NicheNet (Nat. Methods, 2020). These works demonstrate that studying interactions in this setting, may also hold significant biological relevance.
> > >
> > > - We also agree with your point regarding the limitations of inferring interactions from single cell data. Our model indeed incorporates some underlying assumptions, such as stronger mutual influence among cells from similar developmental stages, which we recognize as a simplifying prior. We also understand that space constraints in the rebuttal may limit the assessment of additional spatial transcriptomics results, and we believe these revisions will further demonstrate our method’s capabilities and its alignment with this promising research area.
> > >
> > > - On Additional Analysis: Thank you for the valuable question. We calculate the interaction force in the low-dimensional PCA space and project it back to the original high-dimensional gene space using the PCA loading matrix. We then prioritize the top 200 genes with the largest component values in this projected vector, identifying them as the most significantly influenced by the interaction.
> > >
> > > - On the RBF kernel: Thank you for the insightful question. We recognize that biological interactions are inherently complex. We chose the RBF kernel as a flexible, data-driven approach to avoid biased assumptions about specific cell-type interactions. This choice is based on our assumption that cells at a shared developmental stages, as determined by transcriptomic similarity, are more likely to influence each other. We believe this assumption would be better in the spatial transcriptomics, as the proximity further supports interaction inference.
> > >
> > >
> > > - On Interaction Choices: Thank you. We sincerely apologize for missing this comment in our earlier response and appreciate the opportunity to address it now! The primary purpose of the simulated interactions is to serve as a validation tool. By defining ground-truth interaction dynamics, we can assess whether CytoBridge accurately recovers these interactions, an important step before applying it to real data where true interactions are complex and unknown. Biologically, these interactions model transcriptional influence in gene expression space: attractive interactions represent state synchronization, where cells adopt similar transcriptional profiles via signaling, while repulsive interactions reflect lateral inhibition (e.g., Notch signaling), promoting diversity in cell fates.
> > >
> > >
> > > - Unstability and use of CFM. Thank you for your insightful feedback. We acknowledge that our algorithm’s complexity arises from incorporating additional biological modeling factors to capture the intricate dynamics of cellular interactions.  And we have evaluated our framework across diverse datasets, where it exhibited stable and consistent performance with default hyperparameters. For the use of CFM, we use it as a modular component to efficiently estimate the score function, leveraging its flexibility to model complex dynamics. Our framework is designed to be adaptable, allowing substitution with other suitable score-matching methods other than flow matching. This implementation choice enhances flexibility while maintaining computational efficiency, though we acknowledge it may reduce self-containment.
> > >
> > > - Waddington landscape and plot description: Thank you for the suggestions! We will take this into our account to improve the visualization of the landscape.
> > >
> > > Thank you again for your insightful and constructive feedback.  We warmly welcome any further discussions with your opinions to improve our work!

---

> ### Comment · Reviewer_6tJf · 2025-08-04
>
> Dear authors,
>
> Thank you very much for your further elaboration. I will outline a couple of additional points below.
>
> * Thanks for posting additional publications. I find these papers relevant, but many of them have an aspect in common: they use ligand-receptor information (I mentioned this concept in my answer above as well). To me, this kind of biological evidence is important as a before make interactions in dissociated data somewhat identifiable. I do see that using this information may go beyond the methodological scope of the paper, but in future revisions, I believe this should at least be part of the ground truth formulation.
>
> * With respect to potentials, I think I did not express myself clearly enough, and I apologise for that. I appreciate synthetic experiments with known potentials. However, eventually, I do not think they reflect intricate biological potential. Since a ground truth potential is not available unless ligand-receptor informed, I find this experiment a bit difficult to factor into the downstream story.
>
> Having said this, I commend the authors for all the detail they provided in the rebuttal. The amount of work was great, and I hope my criticism is taken as an encouragement for the next iteration of revisions. I confirm the score increase I discussed in my previous answer and remain open minded to a better assessment upon interaction with the other reviewers.
>
> Thank you, and best wishes for the next steps.

---

> ### Author Response · Authors · 2025-08-05
>
> Thank you for your valuable feedback. We agree that incorporating ligand-receptor information is crucial for enhancing the biological relevance of our model. We believe our framework offers a good foundation for integrating such information, and we plan to explore this direction in future work to further refine our approach. Thank you once again for your insightful comments, which have greatly enriched our manuscript.

---

### Official Review · Reviewer_idfK · 2025-07-03

**Clarity:** 3
**Significance:** 2
**Originality:** 2
**Rating:** 5
**Confidence:** 4

**Summary:**

The authors tackle the challenging problem of learning the dynamics of interacting particles (cells) from sparsely time-resolved observations. Motivated by a body of literature from this domain, they formulate this problem as an Unbalanced Mean-field Schrödinger Bridge (UMFSB) problem, collectively enabling the modelling of stochastic dynamics, birth-death of cells, and cell-cell interactions. To approximate the UMFSB, the authors propose a novel deep learning-based approach they label as CytoBridge. Through experiments on both synthetic and real datasets, the authors demonstrate that CytoBridge outperforms its counterpart SOTA baseline methods for the trajectory inference task.

**Questions:**

- In line 19 of Algorithm 2, do you need to evaluate/simulate the integral during training?
- I don't fully understand how to interpret the learned interaction potential results. More generally, how do the empirical results/plots validate that CytoBridge is learning the correct cell interactions? Could the authors please elaborate on this?
- How does CytoBridge scale to higher dimensions? I believe 50 PCA components for the single-cell datasets is the largest dimensionality considered in this work? It would be useful to provide results for 100 PCs to observe this.
- Regarding the Random Batch Method (RBM), ODE simulation is used to reduce the computational complexity of the ODE simulation with cellular interaction terms. What $p$ (which I understand is the size of the random batch) did you try? I see in Appendix C.2 $p = 16$. Did you try other values for $p$? How does performance change for different $p$'s?
- Similarly, have you conducted an ablation to empirically test how the RBM inference method compares to standard inference? I understand theoretically they should lead to equivalent results, but it may be useful to confirm this empirically.

Minor comments:

- In Definition 3.2, $\phi(\cdot)$ (lowercase) is introduced but not used anywhere in the definition.
- Figure 1 is never referenced in the text.

**Ethical Concerns:**

["NO or VERY MINOR ethics concerns only"]

**Final Justification:**

The authors did a good job of addressing my questions and concerns. Looking at their rebuttal and discussion with other reviewers as well, I see that the authors have added a _significant_ amount of additional experiments. As a consequence, I think the empirical quality of this paper will be improved. To add, the authors are trying to tackle a very difficult problem, learning the dynamics of cells and their accompanying interactions (or more generally, learning the dynamics of interacting particles). These problems are independently challenging, and I have assessed the quality of their contribution in light of this. Lastly, this paper provides a reasonable methodological/algorithmic component, which I believe is well-suited for NeurIPS. Hence, I have raised my score to an accept.

**Limitations:**

Yes, but in the appendix. I encourage the authors to use the extra page allotted for the final version of the manuscript to include the discussion of limitations (and future work) in the main text.

**Paper Formatting Concerns:**

No formatting concerns.

**Quality:**

3

**Strengths And Weaknesses:**

**Strengths:**

- The paper is well written and clear. The authors do a good job of motivating the problem they are addressing and demonstrating a comprehensive understanding of the field / related work.
- The problem addressed in this work, learning dynamics of interacting particles (cells), is clearly formulated and introduced as a mean field unbalanced schrödinger bridge problem. The authors do a good job going from the high-level problem -> mathematical definition of the problem -> how to solve it with machine learning -> empirical experiments.
- The results are generally convincing, barring some drawbacks in comparisons and presentation (see weaknesses and questions). The proposed method, CytoBridge, outperforms all baselines for the task of multi-marginal trajectory inference on both synthetic and real datasets.

**Weaknesses:**

- The authors considered a wide range of trajectory inference baselines for comparison with CytoBridge, however, for the majority of the baselines, the default hyperparameters are used. I recognize that tuning all baselines is time-consuming and not realistic, but to conduct a truly fair comparison, I believe it is important to apply some additional work into searching for reasonable hyperparameter settings for the baselines. I think this is specifically important for methods that don't explicitly use (i.e. train their models on) the datasets considered in this work.
    - In this vein, at the minimum, a simple thing to do is to make sure the networks that parameterize the vector fields are of comparable size in terms of the number of parameters, across all models/baselines.
- The proposed method, CytoBridge, which approximates the formulated mean field unbalanced schrödinger bridge problem, requires the optimization of 4 neural networks -- which likely comes with its own challenges. Although the authors briefly mention this as a limitation in Appendix E, there seems to be limited discussion on the procedure for selecting/finding good hyperparameters (some is included in Appendix C.2). With many hyperparameters to select, this does not seem like a simple process. Some added discussion would be valuable. Especially given that a PINN loss is being used, which, from my understanding, can be difficult to optimize.
    - I also would like to see a discussion of the challenges/effort of optimizing 4 neural networks for this method and how it relates to hyperparameter selection to some of the baselines.

---

> ### Author Rebuttal · Authors · 2025-07-31
>
> > Summary and Strengths
> - Thank you. We sincerely appreciate your recognition of the strengths in our paper. In this paper, we address the challenge of **jointly** modeling cell-cell interactions and unbalanced stochastic biological effects using multiple single-cell snapshots data. This makes our work an **early and essential** step in the integration of **interpretable cellular interactions** with **complex cell dynamics** within the optimal transport framework, offering both **great theoretical and practical potential** for exploring the **spatiotemporal dynamics and interactions of single cells**.
>
> > W1: Hyperparameter of baselines
> - Thank you. We fully agree that a rigorous and fair evaluation is essential. To address this, **we have made every effort to enhance the baseline comparisons by thoroughly tuning key baselines**. Specifically, we have implemented the following improvements：
>     - We have increased the network sizes of the baseline models to ensure their parameter counts are comparable to CytoBridge.
>     - We have extended the training epochs for these models beyond default settings to better suit different datasets.
>     - We have incorporated a learning rate scheduler into the training process to optimize performance.
> - These adjustments have **enhanced the performance of most baselines**. Nevertheless, our algorithm achieves improved performance over tuned baselines, indicating the robust contribution of CytoBridge. Here we present our tuned results ($\mathcal{W}_1$) on the **mouse hematopoiesis** data as below, and **we will update all relevant tables in the manuscript with the results from these tuned baselines.**
>
> |Model (Mouse Data)|t=1|t=2
> |:---|:---|:---
> |SF2M|8.217|11.086
> |Meta FM|8.545|10.313
> |MMFM|7.647|10.156
> |MIOFlow|6.313|6.746
> |uAM|7.537|9.762
> |UDSB|10.687|13.477
> |TIGON|6.140|6.973
> |DeepRUOT|6.052|6.757
> |CytoBridge(Ours)|**6.013**|**6.644**
>
> > W2: Training Complexity
> - Thank you. We fully agree that directly optimizing four neural networks could be challenging. To address this, here we adopted the **two-phase pre-training strategy** detailed in Algorithm 2. This strategy effectively **decouples** the optimization problem: We first initialize the velocity ($v_\theta$) and growth ($g_\theta$) networks and the interaction potential ($\Psi_\theta$), providing a robust starting point for stable end-to-end training. This significantly reduces sensitivity to challenges associated with PINN losses. **We will expand the discussion of this strategy in the manuscript.**
> - Regarding hyperparameter selection, we found that lowering the mass loss weighting during pre-training improves distribution matching and performance. Additionally, extending training epochs for larger datasets enhances results. As detailed in Appendix C.2 and summarized in Table 14, we maintained most key hyperparameters consistent across diverse datasets, from synthetic to real-world scRNA-seq data, except for the scale parameter ($d_\text{cutoff}$), which can be computed directly from each dataset. We also found that increasing the number of training epochs has no adverse effect, allowing users to use a default of (500, 100, 500) epoches. **We will include the more detailed discussion of the hyperparameter selection in the paper.**
> - To support this, **we successfully applied these hyperparameters** (Epoches {500, 100, 500}, with other parameters same in the table) to two additional datasets—a **100D EB dataset and a zebrafish spatiotemporal  (ST) dataset**—with results wil be presented below.
>
> > Q1 : Evaluate the integral during training?
> - Yes, the integral is evaluated during each training step. This is performed numerically using a standard ODE solver. Computed over ODE paths, this step incurs minimal overhead, ensuring computational efficiency without impeding the training process.
>
> >  Q2 : Interpretation of the Cell Interaction
> - Thank you. We fully acknowledge the significance of providing a clear biological interpretation of the cell-cell interactions modeled by CytoBridge. **Below, we present the biological interpretability of CytoBridge from three key perspectives:**
>   - **Detection of Cell-Cell Interactions Across Datasets:** CytoBridge’s interaction term enables the identification of the presence or absence of cell-cell interactions in a dataset. In our study, we analyzed four biological datasets: mouse hematopoiesis, EB, pancreatic differentiation, and EMT datasets. The first three datasets involve organoid-like systems cultured in vitro, where cell-cell interactions are plausible due to their three-dimensional, multicellular organization mimicking tissue-like environments. Conversely, the EMT dataset, derived from an in vitro cell line experiment, consists of isolated cells with minimal physical or signaling interactions. **Our algorithm successfully identified weak interaction patterns in the EMT dataset** (Appendix B.5), aligning with biological expectations, while **detecting stronger interactions in the organoid-like datasets**, consistent with their multicellular nature.
>   - **Interpretable Analysis of Interaction Contributions:** To enhance interpretability, we identified top 200 genes influenced most significantly by cell-cell interactions. Subsequent enrichment analysis of these genes revealed key pathways associated with the biological processes. Applying this to the mouse hematopoiesis dataset, we identified pathways related to positive regulation of leukocyte activation and cell-cell adhesion which are closely linked to hematopoiesis, interactions, and differentiation. **This demonstrates CytoBridge’s ability to uncover biologically relevant mechanisms.**
>   - **Extension to Spatiotemporal Transcriptomics**: Our framework is readily extensible to ST data. We applied CytoBridge to a zebrafish spatiotemporal ST dataset (*Dev. Cell*, 2022, PMID: 35512701). Further details are provided in our response to Reviewer 6tJf.
> - So the ability to detect interaction patterns, identify key contributing genes, and uncover relevant biological pathways highlights its significant potential. **We will expand the discussion of these points in the revised manuscript to ensure clarity.**
>
> | Pathway (Mouse Data) | p.adjust | Count |
> |:---|---:|---:|
> | regulation of T cell activation | 3.38e-12 | 23 |
> | positive regulation of cell activation | 1.42e-11 | 23 |
> | positive regulation of cell-cell adhesion | 5.81e-11 | 20 |
> | positive regulation of leukocyte activation | 1.9e-10 | 21 |
> | leukocyte cell-cell adhesion |3.18e-10 | 21 |
> | lymphocyte differentiation | 1.04e-09 | 21 |
>
> > CytoBridge scale to higher dimensions?
> - Thank you. **Our method maintains scalability comparable to these existing neural ODE-based frameworks.**  In TrajectoryNet, MIOflow, Neural LSB and DeepRUOT, these methods have been tested up to 100 dimensions.
> -  **Following your suggestion, we have now evaluated CytoBridge on the EB data, extending the input from 50 to 100 PCs**. The results ($\mathcal{W}_1$, time), reported below, demonstrate that our algorithm remains effective at this dimensionality.
>
> |Methods (100D EB)|t=1|t=2|t=3|t=4|Time|Memory|
> |-------|---|----|----|----|----|------|
> |MMFM|13.150|14.135|14.441|14.907|4min38s|0.6G|
> |MetaFM|11.699|13.398|14.037|14.727|5min50s|2.2G|
> |uAM|13.628|18.315|20.309|22.973|38s|1.5G|
> |SF2M|11.333|12.982|13.718|14.945|4min43s|0.7G|
> |MIOflow|11.387|12.331|11.905|12.908|6min19s|0.7G|
> |TIGON|10.547|12.926|13.897|14.535|7min33s|0.7G|
> |DeepRUOT|10.226|11.110|11.544|12.424|10min17s|3.7G|
> |CytoBridge|**10.217**|**11.070**|**11.505**|**12.368**|11min26s|6.3G|
>
> - This finding provides empirical evidence for the robustness of our method in scaling to more complex, higher-dimensional scenarios. However, at dimensions around 1000, our approach encounters challenges similar to those faced by other methods, highlighting the need for simulation-free training strategies, such as flow matching, to effectively handle ultra-high-dimensional settings.
>
> > Choice of p in RBM
> - Thank you. **We have conducted an ablation study to analyze its impact. We report the $\mathcal{W}_1$ distances as below and will include the full results in the revised Appendix.** Generally, increasing $p$ slightly improves performance on evaluation metrics by providing a more accurate approximation of the interaction term. However, gains diminish at higher $p$ values, while computational costs increase. Thus, we selected $p=16$ as the default, balancing robust performance with computational efficiency.
>
> |p (Simulation Data)|t=1|t=2|t=3|t=4|
> |-|---|---|---|---|
> |2|0.025|0.033|0.048|0.094|
> |4|0.017|0.023|0.025|0.043|
> |8|0.016|0.019|0.022|0.035|
> |16|0.015|0.014|0.018|0.038|
> |32|0.015|0.016|0.020|0.030|
> |64|0.014|0.014|0.024|0.030|
>
> >  RBM inference vs standard inference?
> - Thank you. **We have conducted an ablation study to assess the RBM impact on model performance.** Results show that the full model (without RBM) and the RBM model ($p=16$) yield comparable performance, with the full model slightly better at certain time points. However, RBM significantly enhances computational efficiency, reducing the interaction term’s complexity from $\mathcal{O}(N^2)$ (e.g., 10^6 for 1000 particles) to $\mathcal{O}(pN)$. The tables demonstrate substantial reductions in memory and inference time with RBM. Given its comparable accuracy and scalability for large-scale single-cell datasets, RBM is a justified and essential component of our framework. Results will be detailed in the revised Appendix C.2.
>
> |Method (Simulation Data)|t=1|t=2|t=3|t=4|Time(s)|Memery(GB)|
> |-|---|---|---|---|---|---|
> |w/o RBM|0.015|0.013|0.023|0.029|0.568|22.1|
> |p=16|0.015|0.014|0.018|0.038|0.115|1.7|
>
> > Def 3.2. Figure 1 is never referenced in the text.
> - Thank you. We apologize for the typographical error. We will thoroughly review the manuscript to address any additional errors.
>
> > Limitations
> - Thank you. We will include the limitations and future work in the main text.
>
> Thank you again!

---

> > ### Comment · Area_Chair_cffq · 2025-08-04
> > **Discussion reminder**
> >
> > Dear Reviewer,
> >
> > We are getting close to the end of the discussion phase. The authors have provided a detailed response to the reviews. When you have a moment, could you please take a look and post a brief comment?
> >
> > Specifically, it would be helpful to know if their response addresses your main concerns and whether it impacts your overall assessment. This feedback is invaluable for the discussion phase.
> >
> > Thank you for your time and effort on this.

---

> > > ### Comment · Reviewer_idfK · 2025-08-05
> > >
> > > Thank you for the very detailed rebuttal! I am happy to see all of the additional experiments you have added and that you have taken all of my comments and suggestions into consideration. Your response has addressed my questions and concerns, and with that, I am happy to raise my score.

---

> > > > ### Author Response · Authors · 2025-08-05
> > > >
> > > > Thank you for the valuable comments and we are very happy to address all the concerns. Your insightful and constructive comments have significantly enhanced our work, and we will certainly incorporate your suggestions into the revised version of the paper.
> > > >
> > > > Thank you once again for your time, expertise, and contributions to improving our manuscript.
> > > >
> > > > Sincerely,
> > > >
> > > > The Authors

---

### Decision · Program_Chairs · 2025-09-17

**Decision:**

Accept (poster)

**Comment:**

This paper introduces CytoBridge, a framework for modeling single-cell dynamics that jointly accounts for unbalanced mass, stochasticity, and cell-cell interactions. This paper presents the Unbalanced Mean-Field Schrödinger Bridge (UMFSB), a new mathematical framework for inferring continuous-time cellular dynamics from discrete snapshots. The core innovation is its ability to simultaneously model three key aspects of biological systems: (1) stochasticity, (2) unbalanced populations, and (3) cell-cell interactions. To make this framework practical, the authors propose CytoBridge, a deep learning algorithm that approximates the UMFSB solution by jointly training four neural networks representing the system's velocity, growth, interaction potential, and score. The authors claim that CytoBridge can accurately reconstruct developmental landscapes and identify meaningful biological patterns, outperforming existing methods that ignore one or more of these critical components.

However, there are some unresolved core concerns. Despite significant effort, the rebuttal did not resolve the fundamental concerns of reviewers 6tJf and SjPe. They remained unconvinced by the biological significance of the learned interactions, arguing that the concept is ill-posed in dissociated data without stronger priors and that the validation did not provide novel insights. Reviewer SjPe explicitly stated, "the current results do not yet support several of the claims made." I agree with both these reviewers, the biological priors used and significance of learned interactions are simply not clear at this point in time. It seems very unclear that cell-cell interactions can be inferred without additional assumptions from dissociated single-cell RNA-seq data.

The discussion solidified a critical split in the reviews. Two reviewers were satisfied with the new experiments and raised their scores, the other two maintained borderline reject stances, unconvinced by the biological significance. While I agree that the biological significance is quite questionable, I believe the method has merit and is a step in the right direction for trajectory inference methods. For this reason, despite its faults and uncertainties I recommend acceptance of this work.